# SciRE-Solver: Accelerating Diffusion Models Sampling by Score-integrand Solver with Recursive Difference

## Abstract

One downside of Diffusion models (DMs) is their slow iterative process. Recent algorithms for fast sampling are designed from the differential equations. However, in the fast algorithms, estimating the derivative of the score function evaluations becomes intractable due to the complexity of large-scale, well-trained neural networks. In this work, we introduce the recursive difference method to calculate the derivative of the score function networks. Building upon, we propose *SciRE-Solver* with the convergence order guarantee for accelerating DMs sampling. Our proposed sampling algorithms attain SOTA FIDs in comparison to existing training-free sampling algorithms, under various number of score function evaluations (NFE). Such as, we achieve 3.48 FID with 12 NFE, and 2.42 FID with 20 NFE for continuous-time model on CIFAR-10. Moreover, we also test the pretrained model of EDM on CIFAR-10 and achieve 2.29 FID with 12 NFE, as well as 1.76 FID with 100 NFE. Empirically, SciRE-Solver with multi-step methods can achieve high-quality samples on the text-to-image generation tasks with only 6~20 NFEs.

## 1 Introduction

Diffusion models (DMs) (Sohl-Dickstein et al., 2015; Ho et al., 2020; Song et al., 2021c) have recently gained significant progress on various tasks, including image generation (Dhariwal & Nichol, 2021; Meng et al., 2022), text-to-image generation (Ramesh et al., 2022), video synthesis (Ho et al., 2022), and voice synthesis (Chen et al., 2021; Liu et al., 2022a). DMs are composed of two diffusion stages. The forward stage of DMs is to add randomness with Gaussian noise in order to slowly disrupt the data distribution, without any training. The reverse stage of DMs is tasked with recovering the original input data from the diffused data by learning to reverse the forward diffusion process, step by step. DMs learn models by emulating the ground-truth inverse process of a fixed forward process.

One key downside of DMs is their slow iterative sampling process (Song et al., 2021a; Karras et al., 2022). Two distinct categories of methods have arisen to tackle this challenge: training-based and training-free methods. Training-based methods require additional training, such as knowledge distillation (Salimans & Ho, 2021; Meng et al., 2023) and consistency models (Song et al., 2023), noise level learning (Nichol & Dhariwal, 2021), or models combined with other generative models (Xiao et al., 2022; Vahdat et al., 2021a; Zhang & Chen, 2021). Training-free methods strive to accelerate the sampling process through numerical algorithms without requiring extra training. Recent training-free fast sampling methods can be attributed to the design of numerical algorithms for solving diffusion ODEs, benefiting from the fact that the sampling process of DMs can be reformulated as solving the corresponding diffusion ODE, as confirmed by DDIM (Song et al., 2021a) and Score-based models (Song et al., 2021c). Following this framework, several fast numerical algorithms with impressive results on DMs have been suggested, including PNDM (Liu et al., 2022b), DPM-Solver (Lu et al., 2022b), DEIS (Zhang & Chen, 2023), UniPC (Zhao et al., 2023), and ERA-Solver (Li et al., 2023). The core differences of these algorithms can be attributed to various derivative estimation or discretization methods, which imply that employing different methods to estimate the derivative of the score function will result in varying sampling performance.

In this work, we introduce a new derivative estimation method, called the Recursive Difference (RD), to calculate the derivative of the score function networks. The FID-measured ablation experiments demonstrate the effectiveness of using the RD method. Based on the RD method and the truncated Taylor expansion of score-integrand, we propose *SciRE-Solver* with the convergence order guarantee

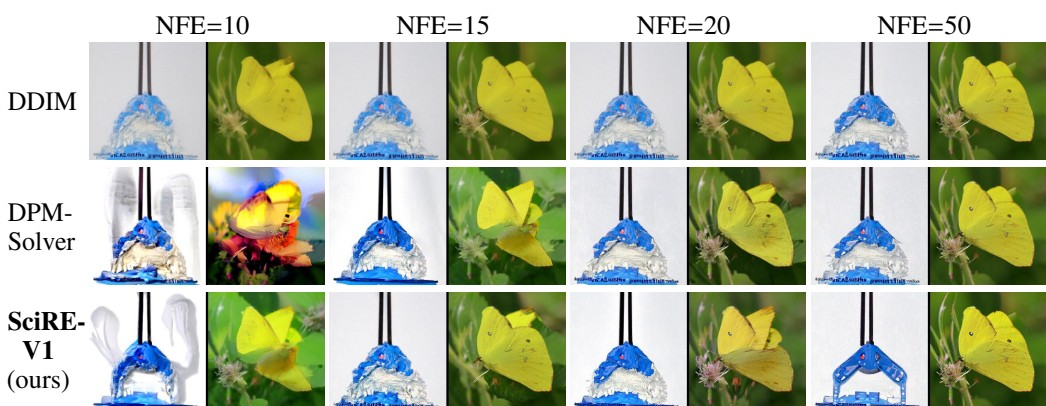

Figure 1: Generated samples of the pre-trained DM on ImageNet 256×256 (classifier scale: 2.5) using 10-50 sampling steps from different sampling methods with the same random seed and codebase. Our algorithm, SciRE-V1 Solver, generates high-quality results in a fewer number of steps.

for accelerating DMs sampling. Our proposed sampling algorithms with RD method advance the sampling efficiency of the training-free sampling method to a new level. Such as, we achieve 3.48 FID with 12 NFE and 2.42 FID with 20 NFE for continuous-time DMs on CIFAR-10, respectively. Furthermore, we observe that SciRE-V1 with a small NFEs demonstrates the promising potential to surpass the FIDs achieved in the original papers of some pre-trained models, distinguishing itself from other samplers. For example, we reach SOTA value of 2.40 FID with 100 NFE for continuous-time DM and of 3.15 FID with 84 NFE for discrete-time DM on CIFAR-10, as well as of 2.17 (2.02) FID with 18 (50) NFE for discrete-time DM on CelebA 64×64. Experiments demonstrate that SciRE-Solver (V1 and V2) exhibit also the ability to generate high-quality results with fewer iterations when applied to high-resolution image datasets, as shown in Figures 1, 6, 8.

## 2 BACKGROUND

### 2.1 DIFFUSION ODEs

A Markov sequence $\{\mathbf{x}_t\}_{t\in[0,T]}$ with $T > 0$ starting with $\mathbf{x}_0$, in the forward diffusion of DMs for $D$-dimensional data, is defined by the following transition kernel:

$$q\left(\mathbf{x}_t \mid \mathbf{x}_0\right) = \mathcal{N}\left(\mathbf{x}_t; \alpha_t\mathbf{x}_0, \sigma_t^2\mathbf{I}\right). \tag{2.1}$$

This transition kernel is equivalent to the stochastic differential equation (SDE) (Kingma et al., 2021):

$$d\mathbf{x}_t = f(t)\mathbf{x}_t\,dt + g(t)d\boldsymbol{\omega}_t, \quad \mathbf{x}_0 \sim q_0\left(\mathbf{x}_0\right), \tag{2.2}$$

where $\boldsymbol{\omega}_t \in \mathbb{R}^D$ denotes a standard Wiener process, and $f(t) = \frac{d\log\alpha_t}{dt}$, $g^2(t) = \frac{d\sigma_t^2}{dt} - 2\frac{d\log\alpha_t}{dt}\sigma_t^2$. This forward diffusion has the following equivalent reverse diffusion from time $T$ to 0 (Song et al., 2021c):

$$d\mathbf{x}_t = \left[f(t)\mathbf{x}_t - g^2(t)\nabla_\mathbf{x}\log q_t\left(\mathbf{x}_t\right)\right]dt + g(t)d\overline{\boldsymbol{\omega}}_t, \quad \mathbf{x}_T \sim q_T\left(\mathbf{x}_T\right), \tag{2.3}$$

where $\overline{\boldsymbol{\omega}}_t$ represents a standard Wiener process. In score-based models, Song et al. (2021c) derived the following ordinary differential equation (ODE):

$$\frac{d\mathbf{x}_t}{dt} = f(t)\mathbf{x}_t - \frac{1}{2}g^2(t)\nabla_\mathbf{x}\log q_t\left(\mathbf{x}_t\right), \quad \mathbf{x}_T \sim q_T\left(\mathbf{x}_T\right), \tag{2.4}$$

where the marginal distribution $q_t\left(\mathbf{x}_t\right)$ of $\mathbf{x}_t$ is equivalent to the marginal distribution of $\mathbf{x}_t$ of the SDE in Eq. (2.3). By substituting the trained noise prediction model $\boldsymbol{\epsilon}_\theta\left(\mathbf{x}_t, t\right)$ for the scaled score function: $-\sigma_t\nabla_\mathbf{x}\log q_t\left(\mathbf{x}_t\right)$, Song et al. (2021c) defined the *diffusion ODE* for DMs:

$$\frac{d\mathbf{x}_t}{dt} = f(t)\mathbf{x}_t + \frac{g^2(t)}{2\sigma_t}\boldsymbol{\epsilon}_\theta\left(\mathbf{x}_t, t\right), \quad \mathbf{x}_T \sim \mathcal{N}\left(\mathbf{0}, \hat{\sigma}^2\boldsymbol{I}\right). \tag{2.5}$$

Since the data prediction model $\boldsymbol{x}_\theta\left(\mathbf{x}_t, t\right)$ and the noise prediction model $\boldsymbol{\epsilon}_\theta\left(\mathbf{x}_t, t\right)$ satisfying: $\boldsymbol{x}_\theta\left(\mathbf{x}_t, t\right) = (\mathbf{x}_t - \sigma_t\boldsymbol{\epsilon}_\theta\left(\mathbf{x}_t, t\right))/\alpha_t$ (Kingma et al., 2021), there exists an equivalent diffusion ODE:

$$\frac{d\mathbf{x}_t}{dt} = \left(f(t) + \frac{g^2(t)}{2\sigma_t^2}\right)\mathbf{x}_t - \alpha_t\frac{g^2(t)}{2\sigma_t^2}\boldsymbol{x}_\theta\left(\mathbf{x}_t, t\right), \quad \mathbf{x}_T \sim \mathcal{N}\left(\mathbf{0}, \hat{\sigma}^2\boldsymbol{I}\right). \tag{2.6}$$

## 2.2 NUMERICAL METHODS OF DIFFUSION ODES

Traditional numerical techniques for solving ODEs find their roots in concepts like Taylor expansions, the trapezoidal rule, and Simpson's rule. These foundational ideas have paved the way for the development of well-known approaches such as **Euler's method**, **Runge-Kutta methods**, and **linear multi-step methods** (Süli, 2010). In the realm of diffusion ODEs, a similar lineage of inspiration from these classical methods can be observed in the construction of various numerical approaches.

DDIM (Song et al., 2021a) can be accurately interpreted as the forward **Euler** method from the perspective of the diffusion ODE in Eq. 2.5. Song et al. (2021c) tested the **Runge-Kutta** Fehlberg method for diffusion ODEs. Liu et al. (2022b) investigated the **Runge-Kutta** methods and **linear multi-step** methods, and based on this, further proposed the PNDM. Lu et al. (2022b) introduced the exponential integrator with the semi-linear structure from the ODE literature (Atkinson et al., 2011), and employed **Taylor expansion** techniques to handle the remaining integration, resulting in the proposed DPM-Solver. Zhang & Chen (2023) proposed DEIS by introducing the exponential integrator and further leveraging the assistance of both **Runge-Kutta** methods and **linear multi-step** (Adams-Bashforth) methods. Li et al. (2023) explored the use of **linear multi-step** (implicit Adams) methods with Lagrange interpolation function, and further proposed ERA-Solver.

In this work, our main focus is on algorithms based on Taylor expansions. We introduce sampling algorithms that are predicated on the recursive difference method, which stands out as one of the distinctions between our algorithm and the DPM-Solver.

## 3 SAMPLING ALGORITHMS BASED ON RECURSIVE DIFFERENCE FOR DIFFUSION MODELS

This section introduces the recursive difference (RD) method, which is employed to compute the derivative of score function within sampling algorithms for DMs based on Taylor expansion. Based on the RD method and the truncated Taylor expansion of the score-integrand, we propose the *SciRE-Solver* with the convergence order guarantee to accelerating sampling of DMs.

### 3.1 RECURSIVE DIFFERENCE METHOD FOR DIFFUSION ODES

Since samples can generated by solving the diffusion ODEs numerically from $T$ to $0$, sampling algorithms can be designed from the numerical solutions of differential equations. By applying the *variation-of-constants* formula (Hale & Lunel, 2013) to ODEs (2.5) and (2.6), we have

$$\mathbf{x}_t = e^{\int_s^t f(r)\mathrm{d}r} \left( \int_s^t h_1(r)\boldsymbol{\epsilon}_\theta(\mathbf{x}_r, r) \, \mathrm{d}r + \mathbf{x}_s \right), \quad \mathbf{x}_t = e^{h_2(t)} \left( -\int_s^t e^{-h_2(r)} \frac{\alpha_r g^2(r)}{2\sigma_r^2} \boldsymbol{x}_\theta(\mathbf{x}_r, r) \, \mathrm{d}r + \mathbf{x}_s \right), \quad (3.1)$$

where $h_1(r) := e^{-\int_s^r f(z)\mathrm{d}z} \frac{g^2(r)}{2\sigma_r}$, $h_2(r) := \int_s^r f(z) + \frac{g^2(z)}{2\sigma_z^2}\mathrm{d}z$, and $\mathbf{x}_s$ represents the given initial value. Then, the most simplified solution formulas for the diffusion ODEs can be obtained, as follows.

**Proposition 3.1** *Let $\mathbf{x}_s$ be a given initial value at time $s > 0$. Then, the diffusion ODEs in Eq. (2.5) and Eq. (2.6) has the following solution formulas, respectively:*

$$\frac{\mathbf{x}_t}{\alpha_t} - \frac{\mathbf{x}_s}{\alpha_s} = \int_{\mathrm{NSR}(s)}^{\mathrm{NSR}(t)} \boldsymbol{\epsilon}_\theta\left(\mathbf{x}_{\mathrm{rNSR}(\tau)}, \mathrm{rNSR}(\tau)\right) \mathrm{d}\tau, \tag{3.2}$$

$$\frac{\mathbf{x}_t}{\sigma_t} - \frac{\mathbf{x}_s}{\sigma_s} = \int_{1/\mathrm{NSR}(s)}^{1/\mathrm{NSR}(t)} \boldsymbol{x}_\theta\left(\mathbf{x}_{\mathrm{rNSR}(1/\tau)}, \mathrm{rNSR}(1/\tau)\right) \mathrm{d}\tau, \tag{3.3}$$

*where $\mathrm{NSR}(\gamma) := \frac{\sigma_\gamma}{\alpha_\gamma}$, we refer to it as the time-dependent **noise-to-signal-ratio (NSR)** function; $\mathrm{rNSR}(\cdot)$ is the inverse function of $\mathrm{NSR}(\cdot)$, satisfying $\gamma = \mathrm{rNSR}\left(\mathrm{NSR}(\gamma)\right)$ for any diffusion time $\gamma$. We provide the detailed derivation in Appendix B for two solution formulas.*

As the integral term in the r.h.s. of (3.2) is solely dependent on the evaluation network $\boldsymbol{\epsilon}_\theta(\mathbf{x}_s, s)$ of scaled score function, we refer to such a concise solution formula as "*score-integrand form*" of diffusion ODEs. Compared to the exponential-product-score-based solution formula in (Lu et al., 2022b), empirically, the algorithm based on the score-integrand form generates more stable samples when using a few NFEs ($\leq 10$), as shown in Figures 1 and 5. In score-integrand form, we can solve

the diffusion ODE by directly integrating the $\epsilon_\theta\left(\mathbf{x}_{\mathrm{rNSR}(\tau)}, \mathrm{rNSR}(\tau)\right)$. In theory, directly tackling this problem is very challenging because $\epsilon_\theta\left(\mathbf{x}_t, t\right)$ is a large-scale, well-trained complex neural network. Nevertheless, we can solve it using numerical methods. For example, we can perform a Taylor expansion on the score-integrand to obtain a rough iterative scheme.

Denote $h_{t_i} := \mathrm{NSR}(t_{i-1}) - \mathrm{NSR}(t_i)$, $\tau_{t_i} := \mathrm{NSR}(t_i)$, $\psi(\tau) := \mathrm{rNSR}(\tau)$, and $\epsilon_\theta^{(k)}\left(\mathbf{x}_{\psi(\tau)}, \psi(\tau)\right) := \frac{\mathrm{d}^k \epsilon_\theta\left(\mathbf{x}_{\psi(\tau)}, \psi(\tau)\right)}{\mathrm{d}\tau^k}$ as $k$-th order total derivative of $\epsilon_\theta\left(\mathbf{x}_{\psi(\tau)}, \psi(\tau)\right)$ w.r.t. $\tau$. For $n \geq 1$, the $n$-th order Taylor expansion of $\epsilon_\theta(\mathbf{x}_{\psi(\tau_{t_{i-1}})}, \psi(\tau_{t_{i-1}}))$ w.r.t. $\tau$ at $\tau_{t_i}$ is

$$\epsilon_\theta\left(\mathbf{x}_{\psi(\tau_{t_{i-1}})}, \psi(\tau_{t_{i-1}})\right) = \sum_{k=0}^{n} \frac{h_{t_i}^k}{k!} \epsilon_\theta^{(k)}\left(\mathbf{x}_{\psi(\tau_{t_i})}, \psi(\tau_{t_i})\right) + O(h_{t_i}^{n+1}). \tag{3.4}$$

By substituting this Taylor expansion into Eq. (3.2), we get

$$\mathbf{x}_{t_{i-1}} = \frac{\alpha_{t_{i-1}}}{\alpha_{t_i}} \mathbf{x}_{t_i} + \alpha_{t_{i-1}} \sum_{k=0}^{n} \frac{h_{t_i}^{k+1}}{(k+1)!} \epsilon_\theta^{(k)}\left(\mathbf{x}_{\psi(\tau_{t_i})}, \psi(\tau_{t_i})\right) + O(h_{t_i}^{n+2}). \tag{3.5}$$

Consequently, Eq. (3.5) provides an iterative scheme for solving the diffusion ODE. By following the classical thought path, we can develop an $n$-th order solver for diffusion ODEs by omitting the error term $O(h_{t_i}^{n+1})$ and approximating the first $(n-1)$-order derivatives $\epsilon_\theta^{(k)}(\mathbf{x}_{\psi(\tau_{t_i})}, \psi(\tau_{t_i}))$ for $k \leq n-1$ in turn (Atkinson et al., 2011). Such as, we can obtain the *first-order iterative algorithm* when $n = 1$:

$$\tilde{\mathbf{x}}_{t_{i-1}} = \frac{\alpha_{t_{i-1}}}{\alpha_{t_i}} \tilde{\mathbf{x}}_{t_i} + \alpha_{t_{i-1}} h_i \epsilon_\theta\left(\tilde{\mathbf{x}}_{\psi(\tau_{t_i})}, \psi(\tau_{t_i})\right), \tag{3.6}$$

where $\tilde{\mathbf{x}}$ is an approximation of the true value $\mathbf{x}$, and $\tilde{\mathbf{x}}_{t_N} = \mathbf{x}_T$ is the given initial value.

Beneath the surface of smooth operations, a pivotal challenge emerges: how to assess derivatives in Taylor expansions when dealing with $n \geq 2$. When it comes to estimating derivatives, one preferred choice is the *finite difference (FD) method*. Clearly, the FD method truncates all challenging higher-order derivative terms ($k \geq 2$) and possesses a truncation error of $O(h_{t_i})$. Some indications suggest that the FD method often lacks outstanding numerical performance in practice. For example, in the pursuit of enhanced numerical performance, it is common to replace $(e^h - h - 1)/h^2$ with $(e^h - 1)/h$ as the new FD coefficient within the framework of exponential integrators (Hochbruck & Ostermann, 2005; Lu et al., 2022b; Zhang & Chen, 2023), guided by the concept of equivalent infinitesimal w.r.t $h$. In light of such indication, we speculate that utilizing the conventional FD method directly to evaluate the derivative of the score function may be a suboptimal choice. Our experiments have further substantiated this conjecture, as illustrated in Figure 3.

To improve the FD method while avoiding the intricacies of higher-order derivatives, we recursively apply the principles of FD to handle terms involving higher-order derivatives at the evaluation point. For example, when dealing with third-order derivative terms, our approach is outlined as follows:

$$\begin{aligned}
\Gamma^{(3)}(\tau_{t_i}) &= \frac{\Gamma^{(2)}(\tau_{t_{i-1}}) - \Gamma^{(2)}(\tau_{t_i})}{h_{t_i}} + O(h_{t_i}) = \frac{\Gamma^{(2)}(\tau_{t_{i-1}})}{h_{t_i}} - \frac{\frac{\Gamma^{(1)}(\tau_{t_{i-1}}) - \Gamma^{(1)}(\tau_{t_i})}{h_{t_i}}}{h_{t_i}} + O(h_{t_i}) \\
&= \frac{\Gamma^{(1)}(\tau_{t_i})}{h_{t_i}^2} - \frac{\Gamma^{(1)}(\tau_{t_{i-1}})}{h_{t_i}^2} + \frac{\Gamma^{(2)}(\tau_{t_{i-1}})}{h_{t_i}} + O(h_{t_i}),
\end{aligned} \tag{3.7}$$

where $\tau_{t_i}$ represents the evaluation point and $\Gamma^{(k)}(\tau)$ is used to denote $\epsilon_\theta^{(k)}\left(\mathbf{x}_{\psi(\tau)}, \psi(\tau)\right)$ for simplicity. Under such recursive rule, each high-order derivative term in Eq. (3.4) can be rewritten as the sum of a scaled first-order derivative function at $\tau_{t_i}$ and a function w.r.t. $\tau_{t_{i-1}}$, while this representation incurs a truncation error of $O(h_{t_i})$. Subsequently, by merging the resulting series of scaled first-order derivatives, we can obtain a new derivative estimate for the score function. We refer to such structured estimation method as the *recursive difference (RD) method*. In Appendix D, we present a detailed derivation of the RD method, with the results as stated in Theorem 3.1 and Corollary 3.1.

Denote $\mathrm{NSR}_{\min} := \min_i\{\mathrm{NSR}(t_i)\}$, $\mathrm{NSR}_{\max} := \max_i\{\mathrm{NSR}(t_i)\}$. We derive the following recursive results for the derivative at the evaluation point.

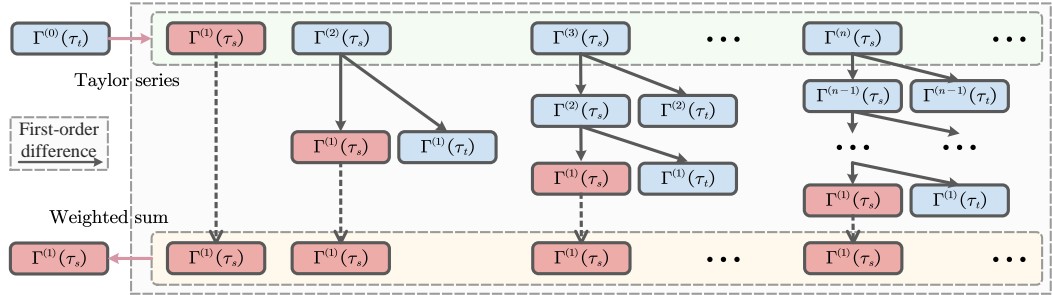

Figure 2: Schematic diagram of the *recursive difference method* tailored for sampling algorithms of diffusion models. The diagram exhibits the derivative process of $\Gamma^{(1)}(\tau_s)$ with $\Gamma^{(0)}(\tau_t)$ given as input. Similarly, we can obtain the $\Gamma^{(k)}(\tau_s)$, $\forall k \in \mathbb{Z}_+$ with $\Gamma^{(0)}(\tau_t)$ as input using analogous procedures.

**Theorem 3.1** *Let $\mathbf{x}_s$ be a given initial value at time $s > 0$, $\mathbf{x}_t$ be the estimated value at time $t$ obtained by the first-order iterative algorithm in Eq. (3.6). Assume that $\epsilon_\theta\left(\mathbf{x}_{\psi(\tau)}, \psi(\tau)\right) \in \mathbb{C}^\infty[\mathrm{NSR}_{\min}, \mathrm{NSR}_{\max}]$. Then, we have*

$$
\begin{aligned}
\epsilon_\theta^{(1)}\left(\tilde{\mathbf{x}}_{\psi(\tau_s)}, \psi(\tau_s)\right) &= \frac{e}{e-1} \frac{\epsilon_\theta\left(\tilde{\mathbf{x}}_{\psi(\tau_t)}, \psi(\tau_t)\right) - \epsilon_\theta\left(\tilde{\mathbf{x}}_{\psi(\tau_s)}, \psi(\tau_s)\right)}{h_s} \\
&- \frac{\epsilon_\theta^{(1)}\left(\tilde{\mathbf{x}}_{\psi(\tau_t)}, \psi(\tau_t)\right)}{e-1} - \frac{(e-2)h_s}{2(e-1)} \epsilon_\theta^{(2)}\left(\tilde{\mathbf{x}}_{\psi(\tau_t)}, \psi(\tau_t)\right) + O(h_s^2),
\end{aligned}
\tag{3.8}
$$

*where $\mathbb{C}^\infty[\mathrm{NSR}_{\min}, \mathrm{NSR}_{\max}]$ denotes $\epsilon_\theta\left(\mathbf{x}_{\psi(\tau)}, \psi(\tau)\right)$ is an infinitely continuously differentiable function w.r.t. $\tau$ over the interval $[\mathrm{NSR}_{\min}, \mathrm{NSR}_{\max}]$.*

We observe that the differentiability constraint imposed by Theorem 3.1 appears to be rather restrictive. To enhance its broad applicability, we further derive the recursive result with limited differentiability.

**Corollary 3.1** *Let $\mathbf{x}_s$ be a given initial value at time $s > 0$, $\mathbf{x}_t$ be the estimated value at time $t$ obtained by the first-order iterative algorithm in Eq. (3.6). Assume that $\epsilon_\theta\left(\mathbf{x}_{\psi(\tau)}, \psi(\tau)\right) \in \mathbb{C}^m[\mathrm{NSR}_{\min}, \mathrm{NSR}_{\max}]$, i.e., $m$ times continuously differentiable, where $m \geq 3$. Then, we have*

$$
\begin{aligned}
\epsilon_\theta^{(1)}\left(\tilde{\mathbf{x}}_{\psi(\tau_s)}, \psi(\tau_s)\right) &= \frac{1}{\phi_1(m)} \frac{\epsilon_\theta\left(\tilde{\mathbf{x}}_{\psi(\tau_t)}, \psi(\tau_t)\right) - \epsilon_\theta\left(\tilde{\mathbf{x}}_{\psi(\tau_s)}, \psi(\tau_s)\right)}{h_s} \\
&- \frac{\phi_2(m)}{\phi_1(m)} \epsilon_\theta^{(1)}\left(\tilde{\mathbf{x}}_{\psi(\tau_t)}, \psi(\tau_t)\right) - \frac{\phi_3(m)h_s}{\phi_1(m)} \epsilon_\theta^{(2)}\left(\tilde{\mathbf{x}}_{\psi(\tau_t)}, \psi(\tau_t)\right) + O(h_s^2),
\end{aligned}
\tag{3.9}
$$

*where $\phi_1(m) = \sum\limits_{k=1}^{m} \frac{(-1)^{k-1}}{k!}$, $\phi_2(m) = \sum\limits_{k=2}^{m} \frac{(-1)^k}{k!}$, and $\phi_3(m) = \sum\limits_{k=3}^{m} \frac{(-1)^{k+1}}{k!}$.*

A simplified truncation form for the RD method is given by $\Gamma^{(1)}(\tau_{t_i}) = \frac{\Gamma(\tau_{t_{i-1}}) - \Gamma(\tau_{t_i})}{\phi_1(m)h_{t_i}}$, obtained by substituting the RD estimation formula from Eq. (3.9) into Eq. (3.5). The complete recursive process of such simplified form is illustrated in Figure 2. Further details are provided in Appendix E.1. Since other truncated forms of the RD method necessitate derivative evaluation at point $\tau_t$ beyond the evaluation point $\tau_s$, we leave this exploration for future research. The main characteristic of such simplified RD version is that *this novel estimation incorporates low-order derivative information hidden in the higher-order derivative terms of the Taylor expansion*. Compared to FD method, such RD method incorporates additional information $\frac{1-\phi_1(m)}{\phi_1(m)} \frac{\Gamma(\tau_{t_{i-1}}) - \Gamma(\tau_{t_i})}{h_{t_i}}$ from other higher-order derivative terms, which may counterbalance with these higher-order terms to a certain level. In Appendix E.2, we provide essential analyses concerning the simplified RD method.

### 3.2 SAMPLING ALGORITHMS BASED ON RECURSIVE DIFFERENCE METHOD

Since Proposition 3.1 involves distinct differential equations that result in different discretization results, we propose two solver versions called *SciRE-V1* and *SciRE-V2* for the diffusion ODE

corresponding to the noise prediction model and the data prediction model, respectively. Now, based on the Eq. (3.5) and the RD methods stated by Corollary 3.1 and Theorem 3.1, we propose two algorithms named *SciRE-V1-2* and *SciRE-V1-3* for $n = 2$ and $n = 3$, respectively. Under mild assumptions, we provide the convergence order for SciRE-V1-$k$ ($k = 2, 3$), as stated in the following theorem. The proof is given in Appendix F. Due to the typically increased complexity of higher-order algorithms, the treatment of $k \geq 4$ will be left for future research. The iteration schemes of SciRE-V1 using multi-step methods are provided in Appendix G.1. In Appendix G.2, we propose *SciRE-V2* for the ODE Eq. (3.3) using the mentioned-above thought process and the RD method.

---

**Algorithm 1** SciRE-V1-2

---

**Require:** initial value $\mathbf{x}_T$, time trajectory $\{t_i\}_{i=0}^N$, model $\epsilon_\theta, m \geq 3$

1: $\tilde{\mathbf{x}}_{t_N} \leftarrow \mathbf{x}_T, r_1 \leftarrow \frac{1}{2}$
2: **for** $i \leftarrow N$ to 1 **do**
3: $\quad h_i \quad \leftarrow \mathrm{NSR}(t_{i-1}) - \mathrm{NSR}(t_i)$
4: $\quad s_i \quad \leftarrow \mathrm{rNSR}\,(\mathrm{NSR}(t_i) + r_1 h_i)$
5: $\quad \tilde{\mathbf{x}}_{s_i} \quad \leftarrow \frac{\alpha_{s_i}}{\alpha_{t_i}}\tilde{\mathbf{x}}_{t_i} + \alpha_{s_i} r_1 h_i \epsilon_\theta\left(\tilde{\mathbf{x}}_{t_i}, t_i\right)$
6: $\quad \tilde{\mathbf{x}}_{t_{i-1}} \leftarrow \frac{\alpha_{t_{i-1}}}{\alpha_{t_i}}\tilde{\mathbf{x}}_{t_i} + \alpha_{t_{i-1}} h_i \epsilon_\theta\left(\tilde{\mathbf{x}}_{t_i}, t_i\right) + \alpha_{t_{i-1}}\frac{h_i}{2\phi_1(m)r_1}\left(\epsilon_\theta\left(\tilde{\mathbf{x}}_{s_i}, s_i\right) - \epsilon_\theta\left(\tilde{\mathbf{x}}_{t_i}, t_i\right)\right)$
7: **end for**
**Return:** $\tilde{\mathbf{x}}_0$.

---

**Algorithm 2** SciRE-V1-3

---

**Require:** initial value $\mathbf{x}_T$, time trajectory $\{t_i\}_{i=0}^N$, model $\epsilon_\theta, m \geq 3$

1: $\tilde{\mathbf{x}}_{t_N} \leftarrow \mathbf{x}_T, r_1 \leftarrow \frac{1}{3}, r_2 \leftarrow \frac{2}{3}$
2: **for** $i \leftarrow N$ to 1 **do**
3: $\quad h_i \leftarrow \mathrm{NSR}(t_{i-1}) - \mathrm{NSR}(t_i)$
4: $\quad s_{i_1}, s_{i_2} \leftarrow \mathrm{rNSR}\,(\mathrm{NSR}(t_i) + r_1 h_i)\,, \mathrm{rNSR}\,(\mathrm{NSR}(t_i) + r_2 h_i)$
5: $\quad \tilde{\mathbf{x}}_{s_{i_1}} \leftarrow \frac{\alpha_{s_{i_1}}}{\alpha_{t_i}}\tilde{\mathbf{x}}_{t_i} + \alpha_{s_{i_1}} r_1 h_i \epsilon_\theta\left(\tilde{\mathbf{x}}_{t_i}, t_i\right)$
6: $\quad \tilde{\mathbf{x}}_{s_{i_2}} \leftarrow \frac{\alpha_{s_{i_2}}}{\alpha_{t_i}}\tilde{\mathbf{x}}_{t_i} + \alpha_{s_{i_2}} r_2 h_i \epsilon_\theta\left(\tilde{\mathbf{x}}_{t_i}, t_i\right) + \alpha_{s_{i_2}}\frac{h_i}{\phi_1(m)}\left(\epsilon_\theta\left(\tilde{\mathbf{x}}_{s_{i_1}}, s_{i_1}\right) - \epsilon_\theta\left(\tilde{\mathbf{x}}_{t_i}, t_i\right)\right)$
7: $\quad \tilde{\mathbf{x}}_{t_{i-1}} \leftarrow \frac{\alpha_{t_{i-1}}}{\alpha_{t_i}}\tilde{\mathbf{x}}_{t_i} + \alpha_{t_{i-1}} h_i \epsilon_\theta\left(\tilde{\mathbf{x}}_{t_i}, t_i\right) + \alpha_{t_{i-1}}\frac{h_i}{2\phi_1(m)r_2}\left(\epsilon_\theta\left(\tilde{\mathbf{x}}_{s_{i_2}}, s_{i_2}\right) - \epsilon_\theta\left(\tilde{\mathbf{x}}_{t_i}, t_i\right)\right)$
8: **end for**
**Return:** $\tilde{\mathbf{x}}_0$.

---

**Theorem 3.2** *Assume that* $\epsilon_\theta\left(\mathbf{x}_{\psi(\tau)}, \psi(\tau)\right) \in \mathbb{C}^m[\mathrm{NSR}_{\min}, \mathrm{NSR}_{\max}]$. *Then, for* $k = 2, 3$, *the global convergence order of SciRE-V1-k is no less than* $k - 1$.

## 4 Assessing the Efficacy of the RD Method through Ablation Studies

This section demonstrates the effectiveness of the RD method from two perspectives: 1. Comparing it with traditional finite difference (FD) method; 2. Introducing the RD method into the exponential-based calculation formula and comparing it with its counterpart algorithm, DPM-Solver-2.

In Corollary 3.1, the RD method degenerates into the FD method, if we set $\phi_1(m) = 1$ and drop other terms. Thus, we set $\phi_1(m) = 1$ in our SciRE-V1 codebase to represent the sampling algorithm based on FD method. Comparative experiments are presented in Figure 3 under identical settings.

To further investigate the RD method, we introduce *SciREI-Solver* (n=2), a variant combining the RD method and the exponential-based calculation formula from DPM-Solver. Refer to Appendix C for the details of SciREI-Solver. We compare the generative performance of SciREI-Solver-2 and DPM-Solver-2 with the identical settings on the CIFAR-10 and CelebA 64 datasets using various time trajectories and termination times, the experiment results are presented in Figure 4. More generally, we also provide the sampling comparison between the RD-based sampling algorithms (SciRE-V1-2 and SciREI-Solver-2) and the baseline algorithm (DPM-Solver-2) on high-resolution image datasets, as shown in Figure 5. More comparisons are provided in Appendix C.

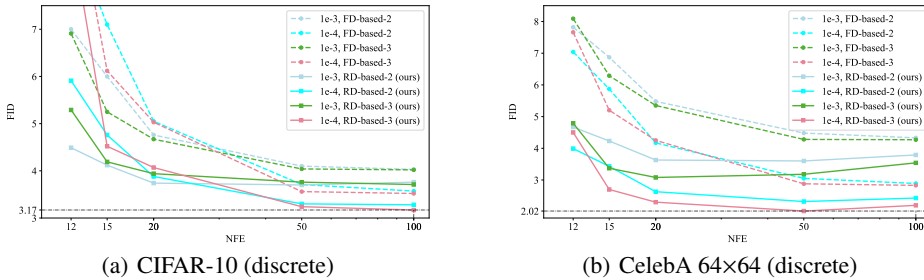

(a) CIFAR-10 (discrete)  (b) CelebA 64×64 (discrete)

Figure 3: Comparisons of FID ↓ obtained by employing RD and FD in SciRE-V1 codebase. The RD-based method is consistently superior to the FD-based method across different cases.

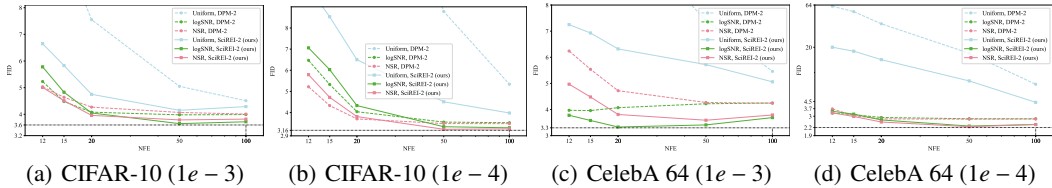

(a) CIFAR-10 ($1e-3$)  (b) CIFAR-10 ($1e-4$)  (c) CelebA 64 ($1e-3$)  (d) CelebA 64 ($1e-4$)

Figure 4: Comparisons of FID ↓ obtained by SciREI-2 and DPM-2 solvers across different trajectories. SciREI-2 is more robust than DPM-2 across different time trajectories under the same sampling step.

## 5 EXPERIMENTS

This section show that SciRE-Solver can improve the sampling efficiency of pre-trained DPM models, including continuous-time and discrete-time DMs. We conduct sampling experiments using individual SciRE-V1-2 and SciRE-V1-3 on the pre-trained models of DM. When comparing with existing fast sampling algorithms, we will compare the best FID values reported by these algorithms in the relevant literature with the FID obtained by our proposed SciRE-V1 under the same NFE, as shown in Table 1. Moreover, we also investigate the SciRE-Solver (V1 and V2) algorithms on image-text generation tasks, as shown in Figures 6 and 8. More details and experiments can be found in Appendix H.

### 5.1 EXPERIMENT SETTING AND ABLATION STUDY

When running our proposed SciRE-V1-$k$ in Algorithms 1 and 2, it is necessary to assign a value $m$ to $\phi_1(m)$. As stated in Corollary 3.1, when assigning $m$, we need to ensure that $m \geq 3$. Considering that the limit of $\phi_1(m)$ is $\frac{e-1}{e}$, then our experiments only consider these two extreme cases, i.e., we only choose to allocate $m$ as 3 or directly set $\phi_1(m) = \frac{e-1}{e}$. We provide ablation experiments for these two cases in Appendix H. The earlier experiments were all run on TITAN-V GPUs.

### 5.2 COMPARISONS OF SAMPLING METHODS USING DISCRETE-TIME AND CONTINUOUS-TIME MODELS

We compare SciRE-V1 proposed in Section 3.2 with existing discrete-time training-free methods in Table 1. Specifically, we use the discrete-time model trained by $L_{\text{simple}}$ in (Ho et al., 2020) on CIFAR-10 and CelebA 64×64 datasets with linear noise schedule, and assign $m = 3$ to $\phi_1(m)$. Under this setting, we use the same NSR-type time trajectory with fixed parameter for both SciRE-V1-2 and SciRE-V1-3, the details are available in Appendix H. SciRE-V1 almost reaches convergence at around 66 NFE and 18 NFE, achieving the new SOTA values of 3.15 FID with 84 NFE, and of 2.17 FID with 18 NFE on CIFAR-10 and CelebA 64×64, respectively.

We compare SciRE-V1-$k$ and SciRE-V1-agile with DPM-Solver-$k$ (Lu et al., 2022b), DPM-Solver-fast and DEIS (Zhang & Chen, 2023), where $k = 2, 3$. On CIFAR-10, we use "VP deep" model (Song et al., 2021c) with the linear noise schedule. When NFE ≥ 15, we employ the identical NSR-type time trajectories with consistent parametric functions for SciRE-V1-2 and SciRE-V1-3, respectively, the details are available in Appendix H. Meanwhile, we consider using the sigmoid-type time trajectory only when NFE is less than 15. The superior of SciRE-V1 is particularly evident in its ability to generate high-quality samples with 2.42 FID in just 20 NFE, as shown in (b) of Figure 7. Furthermore, supported by several experimental validations, SciRE-V1 achieves 2.40 FID in just 100 NFE, which attains a new SOTA value under the VP-deep model (Song et al., 2021c) that we used.

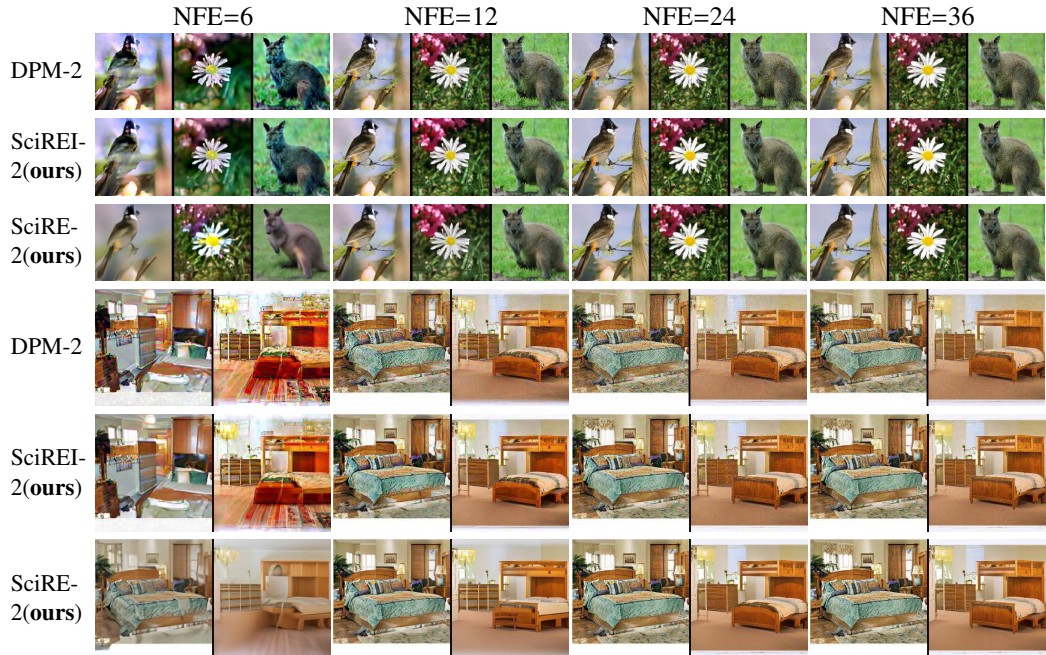

Figure 5: Compare the generation results of the RD-based methods (Solvers: SciRE-V1-2, SciREI-2) and the baseline method (Solver: DPM-2) using 6-36 sampling steps with the uniform time trajectory and identical settings, on pre-trained models with ImageNet 128×128 and LSUN bedroom 256×256.

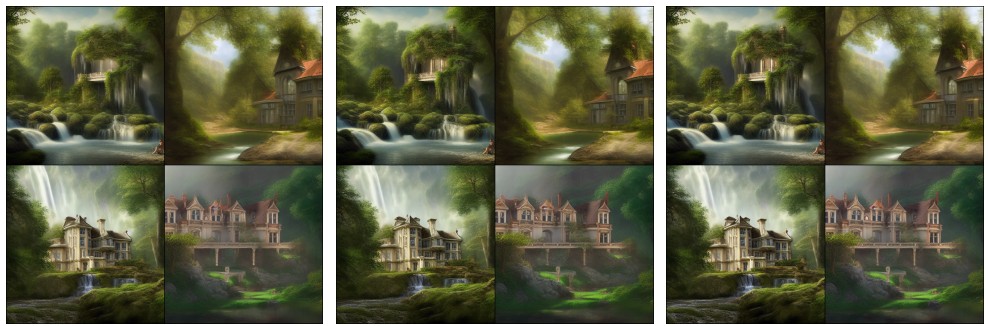

(a) DPM-Solver++ (multistep). (b) SciRE-V1-2m, Algorithm 4. (c) SciRE-V2-2m, Algorithm 7.

Figure 6: Random samples of Stable-Diffusion, using only 6 NFE and text prompt "A beautiful mansion beside a waterfall in the woods, by josef thoma, matte painting, trending on artstation HQ".

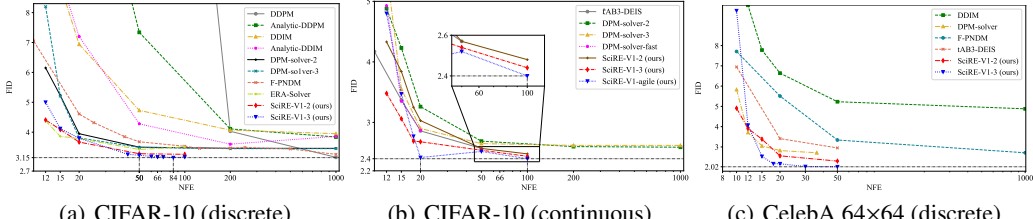

(a) CIFAR-10 (discrete)  (b) CIFAR-10 (continuous)  (c) CelebA 64×64 (discrete)

Figure 7: The comparative diagram of FID ↓ of different training-free sampling methods on the CIFAR-10 and CelebA 64×64 datasets. In these three cases, our samplers reach SOTA.

# 6 CONCLUSIONS

In this work, we introduce the recursive difference (RD) method to calculate the derivative of the score function evaluations in the realm of diffusion models. By applying the RD method to the truncated Taylor expansion of the score-integrand, we propose the *SciRE-Solver* with the convergence order guarantee to accelerate the sampling process of DMs. The effectiveness of the RD method in evaluating the derivative of the score function in regular diffusion modes has been confirmed

| Sampling method \NFE | 12 | 15 | 20 | 50 | 200 | 1000 |
|---|---|---|---|---|---|---|
| CIFAR-10 (discrete-time model (Ho et al., 2020), linear noise schedule) | | | | | | |
| DDPM | 246.3 | 197.6 | 137.3 | 32.6 | 4.03 | 3.16 |
| DDIM | 11.02 | 8.92 | 6.94 | 4.73 | 4.07 | 3.95 |
| Analytic-DDIM | 11.68 | 9.16 | 7.20 | 4.28 | 3.60 | 3.86 |
| $t$AB3-DEIS | 7.12(10NFE) | | 4.53 | 3.78 | \ | \ |
| DPM-Solver-2 | 6.15 | $^\dagger$5.23 | 3.95 | 3.50 | 3.46 | 3.46 |
| DPM-Solver-3 | 8.20 | 5.21 | $^\dagger$3.81 | $^\dagger$3.49 | $^\dagger$3.45 | $^\dagger$3.45 |
| F-PNDM | 7.03(10NFE) | | 4.61 | 3.68 | 3.47 | 3.26 |
| ERA-Solver | **4.38** | **3.86** | 3.79 | 3.42 | 3.51(100NFE) | |
| SciRE-V1-2 (ours) | 4.41 | $^\dagger$4.09 | **3.67** | 3.28 | 3.26(100NFE) | |
| SciRE-V1-3 (ours) | 5.00 | 4.12 | $^\dagger$3.80 | $^\dagger$**3.23** | **3.15** (84NFE) | |
| CIFAR-10 (VP deep continuous-time model (Song et al., 2021c)) | | | | | | |
| DPM-Solver-2 | 4.88 | $^\dagger$4.23 | 3.26 | 2.69 | 2.60 | 2.59 |
| DPM-Solver-3 | 5.53 | 3.55 | $^\dagger$2.90 | $^\dagger$2.65 | $^\dagger$2.62 | $^\dagger$2.62 |
| DPM-Solver-fast | 4.93 | 3.35 | 2.87 | \ | \ | \ |
| $t$AB3-DEIS | \ | 3.37 | 2.86 | 2.57 | \ | \ |
| SciRE-V1-2 (ours) | 4.33 | $^\dagger$3.84 | 3.03 | 2.57 | 2.48 (100NFE) | |
| SciRE-V1-3 (ours) | **3.48** | **3.06** | $^\dagger$2.68 | $^\dagger$2.54 | $^\dagger$2.44 (100NFE) | |
| SciRE-V1-agile (ours) | 4.80 | 3.47 | **2.42** | **2.52** | **2.40** (100NFE) | |
| CIFAR-10 (edm (Karras et al., 2022)) | | | | | | |
| EDM-Heun | 7.28 | $^\dagger$4.47 | 2.38 | 1.83 | 1.84 (100NFE) | |
| SciRE-V1-2 | **2.29** | $^\dagger$**2.16** | **1.94** | **1.79** | **1.76** (100NFE) | |

| CelebA 64×64 (discrete-time model (Song et al., 2021a), linear noise schedule) | | | | | | |
|---|---|---|---|---|---|---|
| Sampling method \NFE | 10 | 12 | 15 | 20 | 50 | 1000 |
| DDIM | 10.85 | 9.99 | 7.78 | 6.64 | 5.23 | 4.88 |
| DPM-Solver | 5.83 | **3.71** | 3.05 | 2.82 | 2.71 | (36NFE) |
| F-PNDM | 7.71 | \ | \ | 5.51 | 3.34 | 2.71 |
| $t$AB3-DEIS | 6.95 | \ | \ | 3.41 | 2.95 | \ |
| SciRE-V1-2 (ours) | **4.91** | 3.91 | $^\dagger$3.38 | 2.56 | 2.30 | – |
| SciRE-V1-3 (ours) | $^\dagger$9.72 | 4.07 | **2.53** | $^\dagger$**2.17** | $^\dagger$**2.02** | – |

Table 1: Generation quality measured by FID ↓ of different sampling methods for DMs on CIFAR-10 and CelebA 64×64. In this Table, we compare the best FID reported in existing literature with the FID achieved by our proposed SciRE-V1 at the same NFE. The bold black represents the best result obtained under the same NFE (column). The results with $^\dagger$ means the actual NFE is smaller than the given NFE because the given NFE cannot be divided by 2 or 3. Some results are missing in their original papers, which are replaced by " \ ". Here, we used the same time trajectory scheme to evaluate the results of SciRE-V1 on CIFAR-10 and CelebA 64×64 datasets with discrete models. The setting of continuous-time on CIFAR-10 are described in Section 5.2. More comparisons and additional details are shown in Appendix H.

through comparative experiments involving FID and generated samples. These experiments were conducted for ablation comparisons with both the finite-based difference algorithm and the popular DPM-Solver-2 algorithm. SciRE-Solver (versions: V1 and V2 ) is a new type of algorithm that provides an alternative sampling scheme for accelerating diffusion models. Numerical experiments indicate that SciRE-Solver not only can generates high-quality samples across various datasets using fewer-steps but also, using a small NFEs demonstrates promising potential to surpass the FID achieved by some pre-trained models in their original papers using no fewer than 1000 NFEs.

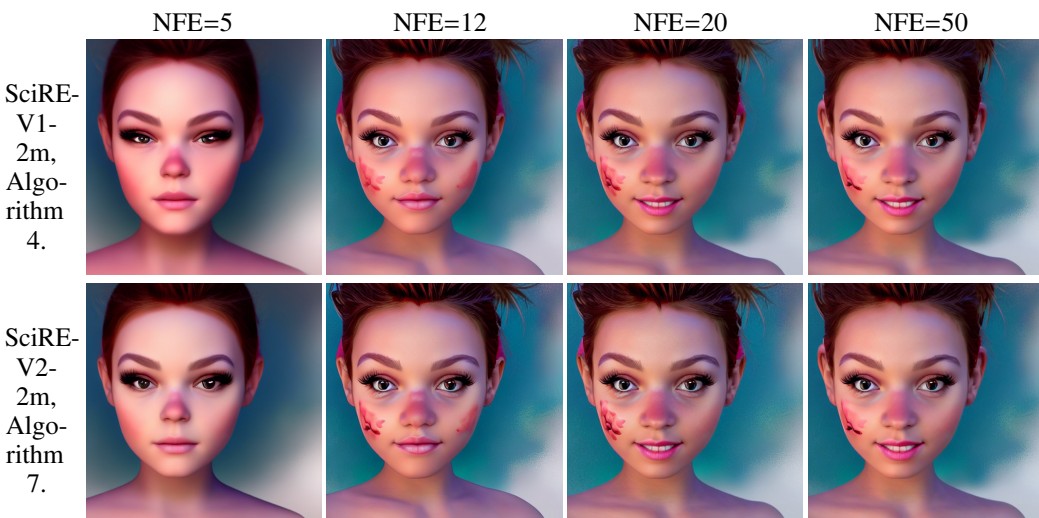

Figure 8: Random samples of Stable-Diffusion by SciRE-V1 and SciRE-V2, using varying NFEs and the text prompt "a girl face in Disney style, physically-based rendering, ultimate painting, UHD".

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

# Appendix

## Table of Contents

# A    PRELIMINARIES

**Related Work:**

Diffusion models (DMs) (Sohl-Dickstein et al., 2015; Ho et al., 2020; Song et al., 2021c) have recently gained remarkable generation ability on image generation (Dhariwal & Nichol, 2021; Karras et al., 2022), yielding extensive applications such as speech, singing and video synthesis (Chen et al., 2021; Liu et al., 2022a; Ho et al., 2022), controllable image generation, translation and editing (Nichol et al., 2022; Ramesh et al., 2022; Rombach et al., 2022; Meng et al., 2022; Zhao et al., 2022; Couairon et al., 2023; Zhang & Agrawala, 2023), likelihood estimation (Song et al., 2021b; Kingma et al., 2021; Lu et al., 2022a; Zheng et al., 2023). As diffusion models gradually achieve success in various fields, the downside of slow diffusion sampling is also gaining more attention. Therefore, improving the output quality and/or reducing the computational cost of sampling is an important topic in diffusion models research, e.g., (Dockhorn et al., 2022; Jolicoeur-Martineau et al., 2021; Liu et al., 2022b; Lu et al., 2022b;c; Luhman & Luhman, 2021; Nichol & Dhariwal, 2021; Salimans & Ho, 2022; Vahdat et al., 2021b; Karras et al., 2022; Zhang & Chen, 2023).

The forward stage of DMs is to add randomness with Gaussian noise in order to slowly disrupt the data distribution, without any training. The reverse stage of DMs is tasked with recovering the original input data from the diffused (noisy) data by learning to reverse the forward diffusion process, step by step. Both the forward and reverse diffusion stages can be explained using differential equations. Next, we provide an introduction to diffusion models from the perspective of differential equations.

## A.1    DIFFUSION SDEs

In the forward diffusion process of DMs for $D$-dimensional data, a Markov sequence $\{\mathbf{x}_t\}_{t\in[0,T]}$ with $T > 0$ starting with $\mathbf{x}_0$ is defined by the transition distribution

$$q\left(\mathbf{x}_t \mid \mathbf{x}_{t-1}\right) := \mathcal{N}\left(\mathbf{x}_t; \beta_t \mathbf{x}_{t-1}, \left(1 - \beta_t^2\right)\mathbf{I}\right), \tag{A.1}$$

where $\beta_t \in \mathbb{R}^+$ is the variance schedule function, which is differentiable w.r.t $t$ and possesses a bounded derivative. Given a transition distribution, one can formulate the transition kernel function for noisy data $\mathbf{x}_t$ conditioned on clean data $\mathbf{x}_0$ through the superposition of Gaussian distributions:

$$q\left(\mathbf{x}_t \mid \mathbf{x}_0\right) = \mathcal{N}\left(\mathbf{x}_t; \alpha_t \mathbf{x}_0, \sigma_t^2 \mathbf{I}\right). \tag{A.2}$$

In DDPM (Ho et al., 2020), $\alpha_t$ is defined as $\prod_{i=1}^t \beta_i$, and $\sigma_t^2 + \alpha_t^2 = 1$ is referred to as the variance-preserving (VP) setting. DMs choose *noise schedules* for $\alpha_t$ and $\sigma_t$ to ensure that the *marginal distribution* $q_T\left(\mathbf{x}_T\right)$ approximates $\mathcal{N}\left(\mathbf{x}_T; \mathbf{0}, \hat{\sigma}^2\mathbf{I}\right)$ for $\hat{\sigma} > 0$. Kingma et al. (2021) generalized this fixed schedule of Eq. (A.1) through the parameterization

$$\sigma_t^2 = \text{sigmoid}\left(\gamma_{\boldsymbol{\eta}}(t)\right), \tag{A.3}$$

where $\gamma_{\boldsymbol{\eta}}(t)$ is a monotonic neural network with parameters $\boldsymbol{\eta}$. Under the schedule defined in Eq. (A.3) and the VP setting, $\alpha_t^2/\sigma_t^2$ is strictly decreasing w.r.t. $t$ and is referred to as the signal-to-noise-ratio (SNR). With the schedule of Eq. (A.3), Kingma et al. (2021) established the equivalence between the transition kernel of the following SDE and the one in Eq. (A.2) for $\forall t \in [0, T]$:

$$d\mathbf{x}_t = f(t)\mathbf{x}_t\, dt + g(t)d\boldsymbol{\omega}_t, \quad \mathbf{x}_0 \sim q_0\left(\mathbf{x}_0\right), \tag{A.4}$$

where $\boldsymbol{\omega}_t \in \mathbb{R}^D$ denotes a standard Wiener process, and

$$f(t) = \frac{d\log\alpha_t}{dt}, \quad g^2(t) = \frac{d\sigma_t^2}{dt} - 2\frac{d\log\alpha_t}{dt}\sigma_t^2. \tag{A.5}$$

Further, Song et al. (2021c) demonstrated with some regularity conditions that the forward process in Eq. (A.2) has the following equivalent reverse process (*reverse SDE*) from time $T$ to 0:

$$d\mathbf{x}_t = \left[f(t)\mathbf{x}_t - g^2(t)\nabla_{\mathbf{x}}\log q_t\left(\mathbf{x}_t\right)\right]dt + g(t)d\overline{\boldsymbol{\omega}}_t, \quad \mathbf{x}_T \sim q_T\left(\mathbf{x}_T\right), \tag{A.6}$$

where $\overline{\boldsymbol{\omega}}_t$ represents a standard Wiener process in the reverse time. Since $f(t)$ and $g(t)$ are determined by the noise schedule $(\alpha_t, \sigma_t)$ in the reverse SDE (A.6), the sole term that remains unknown is the score function $\nabla_{\mathbf{x}}\log q_t\left(\mathbf{x}_t\right)$ at each time $t$. Therefore, DMs train the model by using a neural network

$\epsilon_\theta(\mathbf{x}_t, t)$ parameterized by $\theta$ to approximate the scaled score function: $-\sigma_t \nabla_\mathbf{x} \log q_t(\mathbf{x}_t)$, where the parameter $\theta$ is trained by a re-weighted variant of the evidence lower bound (ELBO) (Ho et al., 2020; Song et al., 2021c):

$$
\begin{aligned}
\mathcal{L}(\theta; \lambda(t)) &= \frac{1}{2} \int_0^T \lambda(t) \mathbb{E}_{q_t(\mathbf{x}_t)} \left[ \left\| \epsilon_\theta(\mathbf{x}_t, t) + \sigma_t \nabla_\mathbf{x} \log q_t(\mathbf{x}_t) \right\|_2^2 \right] dt \\
&= \frac{1}{2} \int_0^T \lambda(t) \mathbb{E}_{q_0(\mathbf{x}_0)} \mathbb{E}_{q(\epsilon)} \left[ \| \epsilon_\theta(\mathbf{x}_t, t) - \epsilon \|_2^2 \right] dt + C,
\end{aligned}
\tag{A.7}
$$

where $\epsilon \sim q(\epsilon) = \mathcal{N}(\epsilon; \mathbf{0}, \mathbf{I})$, $\mathbf{x}_t = \alpha_t \mathbf{x}_0 + \sigma_t \epsilon$, $\lambda(t)$ represents a weighting function, and $C$ is a $\theta$-independent constant. After the model is trained, DMs replace the score function in Eq. (A.4) with $-\epsilon_\theta(\mathbf{x}_t, t)/\sigma_t$ and define the following *diffusion SDE*:

$$
d\mathbf{x}_t = \left[ f(t)\mathbf{x}_t + \frac{g^2(t)}{\sigma_t} \epsilon_\theta(\mathbf{x}_t, t) \right] dt + g(t) d\overline{\omega}_t, \quad \mathbf{x}_T \sim \mathcal{N}\left(\mathbf{0}, \hat{\sigma}^2 \mathbf{I}\right).
\tag{A.8}
$$

DMs can generate samples by numerically solving the diffusion SDE stated in Eq. (A.8) using discretization methods that span from $T$ to 0.

## A.2 DIFFUSION ODEs

Based on the reverse SDE in Eq. (A.6), Song et al. (2021c) derived a Liouville equation by investigating the evolution equation (Fokker-Planck Equation) of the probability density function of the variable $\mathbf{x}_t$. This Liouville equation has the same probability density function w.r.t. the variable $\mathbf{x}_t$ as that of the reverse SDE. As a result, the reverse SDE can be transformed into the following ODE:

$$
\frac{d\mathbf{x}_t}{dt} = f(t)\mathbf{x}_t - \frac{1}{2} g^2(t) \nabla_\mathbf{x} \log q_t(\mathbf{x}_t), \quad \mathbf{x}_T \sim q_T(\mathbf{x}_T),
\tag{A.9}
$$

where $\mathbf{x}_t$ has a marginal distribution $q_t(\mathbf{x}_t)$, which is equivalent to the marginal distribution of $\mathbf{x}_t$ of the reverse SDE in Eq. (A.6). Since the $\epsilon_\theta(\mathbf{x}_t, t)$ trained in Eq. (A.7) can also be thought of as predicting the Gaussian noise added to $\mathbf{x}_t$, it is commonly referred to as the *noise prediction model*. By substituting the trained noise prediction model for the score function in Eq. (A.9), Song et al. (2021c) defined the following *diffusion ODE* for DMs:

$$
\frac{d\mathbf{x}_t}{dt} = f(t)\mathbf{x}_t + \frac{g^2(t)}{2\sigma_t} \epsilon_\theta(\mathbf{x}_t, t), \quad \mathbf{x}_T \sim \mathcal{N}\left(\mathbf{0}, \hat{\sigma}^2 \mathbf{I}\right).
\tag{A.10}
$$

Therefore, one can also generate samples by solving the diffusion ODE from $T$ to 0. Alternatively, the data prediction model $\boldsymbol{x}_\theta(\mathbf{x}_t, t)$ and the noise prediction model $\epsilon_\theta(\mathbf{x}_t, t)$ satisfying: $\boldsymbol{x}_\theta(\mathbf{x}_t, t) = (\mathbf{x}_t - \sigma_t \epsilon_\theta(\mathbf{x}_t, t))/\sigma_t$ (Kingma et al., 2021), there exists an equivalent diffusion ODE w.r.t. $\boldsymbol{x}_\theta(\mathbf{x}_t, t)$:

$$
\frac{d\mathbf{x}_t}{dt} = \left( f(t) + \frac{g^2(t)}{2\sigma_t^2} \right) \mathbf{x}_t - \alpha_t \frac{g^2(t)}{2\sigma_t^2} \boldsymbol{x}_\theta(\mathbf{x}_t, t), \quad \mathbf{x}_T \sim \mathcal{N}\left(\mathbf{0}, \hat{\sigma}^2 \mathbf{I}\right).
\tag{A.11}
$$

# B PROOF OF PROPOSITION 3.1

We first demonstrate the simplified solution formula based on the noise prediction model.

The solution formula of the semi-linear ODE (A.10) can be formulated by the *variation-of-constants* formula (Hale & Lunel, 2013):

$$
\mathbf{x}_t = e^{\int_s^t f(\gamma) d\gamma} \left( \int_s^t h_1(\gamma) \epsilon_\theta(\mathbf{x}_\gamma, \gamma) d\gamma + \mathbf{x}_s \right),
\tag{B.1}
$$

where $h_1(\gamma) := e^{-\int_s^\gamma f(z) dz} \frac{g^2(\gamma)}{2\sigma_\gamma}$, and $\mathbf{x}_s$ represents the given initial value. Since $f(\gamma) = \frac{d \log(\alpha_\gamma)}{d\gamma}$, thus $h_1(\gamma) = \frac{\alpha_s}{\alpha_\gamma} \frac{g^2(\gamma)}{2\sigma_\gamma}$. We observe that $h_1(\gamma)$ can be rewritten as

$$
h_1(\gamma) = \frac{\alpha_s}{2\alpha_\gamma \sigma_\gamma} \left( \frac{d\sigma_\gamma^2}{d\gamma} - 2\frac{d \log \alpha_\gamma}{d\gamma} \sigma_\gamma^2 \right) = \frac{\alpha_s}{\alpha_\gamma} \left( \frac{d\sigma_\gamma}{d\gamma} - \frac{\sigma_\gamma}{\alpha_\gamma} \frac{d\alpha_\gamma}{d\gamma} \right) = \alpha_s \frac{d\text{NSR}(\gamma)}{d\gamma},
\tag{B.2}
$$

where $\text{NSR}(\gamma) := \frac{\sigma_\gamma}{\alpha_\gamma}$, and we refer to it as the time-dependent *noise-to-signal-ratio (NSR)* function. Note that the NSR function defined above differs from the signal-to-noise-ratio (SNR) function defined in (Kingma et al., 2021), but there is a relationship between them: $\text{SNR} = \frac{1}{\text{NSR}^2}$. Then, based on Eq. (B.2), we can rewrite Eq. (B.1) as

$$\mathbf{x}_t = \frac{\alpha_t}{\alpha_s}\mathbf{x}_s + \alpha_t \int_s^t \frac{\mathrm{d}\text{NSR}(\gamma)}{\mathrm{d}\gamma}\epsilon_\theta(\mathbf{x}_\gamma, \gamma)\,\mathrm{d}\gamma. \tag{B.3}$$

Since $\text{NSR}(\cdot)$ is a monotonically function w.r.t. time, we can define its reverse function as $\text{rNSR}(\cdot)$, such that $\gamma = \text{rNSR}(\text{NSR}(\gamma))$ for any diffusion time $\gamma$. Thus, using the *change-of-variables* for $\text{NSR}(\gamma)$ to Eq. (B.3), we can obtain

$$\begin{aligned}
\mathbf{x}_t &= \frac{\alpha_t}{\alpha_s}\mathbf{x}_s + \alpha_t \int_s^t \frac{\mathrm{d}\text{NSR}(\gamma)}{\mathrm{d}\gamma}\epsilon_\theta(\mathbf{x}_\gamma, \gamma)\,\mathrm{d}\gamma \\
&= \frac{\alpha_t}{\alpha_s}\mathbf{x}_s + \alpha_t \int_s^t \epsilon_\theta(\mathbf{x}_\gamma, \gamma)\,\mathrm{d}\text{NSR}(\gamma) \\
&= \frac{\alpha_t}{\alpha_s}\mathbf{x}_s + \alpha_t \int_{\text{NSR}(s)}^{\text{NSR}(t)} \epsilon_\theta(\mathbf{x}_{\text{rNSR}(\tau)}, \text{rNSR}(\tau))\,\mathrm{d}\tau.
\end{aligned} \tag{B.4}$$

Thus, we complete the proof of the simplified solution formula in the Proposition 3.1 based on the noise prediction model.

Next, we demonstrate the simplified solution formula based on the data prediction model. By applying the variation-of-constants formula to Eq. (A.11), we have

$$\mathbf{x}_t = e^{h_2(t)}\left(-\int_s^t e^{-h_2(r)}\frac{\alpha_r g^2(r)}{2\sigma_r^2}\mathbf{x}_\theta(\mathbf{x}_r, r)\,\mathrm{d}r + \mathbf{x}_s\right), \tag{B.5}$$

where $h_2(r) := \int_s^r f(z) + \frac{g^2(z)}{2\sigma_z^2}\mathrm{d}z$, and $\mathbf{x}_s$ represents the given initial value. $h_2(r)$ can be rewritten as

$$h_2(r) = \int_s^r \frac{\mathrm{d}\log\alpha_\gamma}{\mathrm{d}\gamma} + \frac{1}{2\sigma_\gamma^2}\frac{\mathrm{d}\sigma_\gamma^2}{\mathrm{d}\gamma} - \frac{\mathrm{d}\log\alpha_\gamma}{\mathrm{d}\gamma}\mathrm{d}\gamma = \int_s^r \frac{1}{\sigma_\gamma}\frac{\mathrm{d}\sigma_\gamma}{\mathrm{d}\gamma}\mathrm{d}\gamma = \log\frac{\sigma_r}{\sigma_s}. \tag{B.6}$$

Eq. (B.5) can then be rewritten as

$$\begin{aligned}
\mathbf{x}_t &= \frac{\sigma_t}{\sigma_s}\left(-\int_s^t \frac{\sigma_s}{\sigma_r}\frac{\alpha_r g^2(r)}{2\sigma_r^2}\mathbf{x}_\theta(\mathbf{x}_r, r)\,\mathrm{d}r + \mathbf{x}_s\right) \\
&= \frac{\sigma_t}{\sigma_s}\mathbf{x}_s - \sigma_t \int_s^t \frac{1}{\sigma_r}\frac{\alpha_r g^2(r)}{2\sigma_r^2}\mathbf{x}_\theta(\mathbf{x}_r, r)\,\mathrm{d}r \\
&= \frac{\sigma_t}{\sigma_s}\mathbf{x}_s - \sigma_t \int_s^t \frac{\alpha_r}{\sigma_r}\frac{1}{2\sigma_r^2}\left(\frac{\mathrm{d}\sigma_r^2}{\mathrm{d}r} - 2\frac{\mathrm{d}\log\alpha_r}{\mathrm{d}r}\sigma_r^2\right)\mathbf{x}_\theta(\mathbf{x}_r, r)\,\mathrm{d}r \\
&= \frac{\sigma_t}{\sigma_s}\mathbf{x}_s + \sigma_t \int_s^t \frac{\alpha_r}{\sigma_r}\left(\frac{\mathrm{d}\log\alpha_r}{\mathrm{d}r} - \frac{1}{2\sigma_r^2}\frac{\mathrm{d}\sigma_r^2}{\mathrm{d}r}\right)\mathbf{x}_\theta(\mathbf{x}_r, r)\,\mathrm{d}r \\
&= \frac{\sigma_t}{\sigma_s}\mathbf{x}_s + \sigma_t \int_s^t \frac{\alpha_r}{\sigma_r}\left(\frac{1}{\alpha_r}\frac{\mathrm{d}\alpha_r}{\mathrm{d}r} - \frac{1}{\sigma_r}\frac{\mathrm{d}\sigma_r}{\mathrm{d}r}\right)\mathbf{x}_\theta(\mathbf{x}_r, r)\,\mathrm{d}r \\
&= \frac{\sigma_t}{\sigma_s}\mathbf{x}_s + \sigma_t \int_s^t \left(\frac{1}{\sigma_r}\frac{\mathrm{d}\alpha_r}{\mathrm{d}r} - \frac{\alpha_r}{\sigma_r^2}\frac{\mathrm{d}\sigma_r}{\mathrm{d}r}\right)\mathbf{x}_\theta(\mathbf{x}_r, r)\,\mathrm{d}r \\
&= \frac{\sigma_t}{\sigma_s}\mathbf{x}_s + \sigma_t \int_s^t \mathbf{x}_\theta(\mathbf{x}_r, r)\frac{\mathrm{d}}{\mathrm{d}r}\left(\frac{\alpha_r}{\sigma_r}\right)\mathrm{d}r \\
&= \frac{\sigma_t}{\sigma_s}\mathbf{x}_s + \sigma_t \int_s^t \mathbf{x}_\theta(\mathbf{x}_r, r)\,\mathrm{d}\frac{\alpha_r}{\sigma_r},
\end{aligned} \tag{B.7}$$

by change-of-variables,

$$\mathbf{x}_t = \frac{\sigma_t}{\sigma_s}\mathbf{x}_s + \sigma_t \int_{\frac{\alpha_s}{\sigma_s}}^{\frac{\alpha_t}{\sigma_t}} \mathbf{x}_\theta(\mathbf{x}_{h_3(\tau)}, h_3(\tau))\,\mathrm{d}\tau, \tag{B.8}$$

where $h_3(\frac{\alpha_\gamma}{\sigma_\gamma}) := \gamma$ for any diffusion time $\gamma$. Note that $\text{NSR}(\gamma) = \frac{\sigma_\gamma}{\alpha_\gamma}$, then $\frac{\alpha_\gamma}{\sigma_\gamma} = \frac{1}{\text{NSR}(\gamma)}$ and $h_3(\tau) = \text{rNSR}\left(\frac{1}{\tau}\right)$. Therefore, we complete the proof of the simplified solution formula in the Proposition 3.1 based on the data prediction model.

**Hints** Eq. (B.4) implies that the solution of the diffusion ODE can be decomposed into a linear part and a nonlinear part, and this structure arises from the use of the *variation-of-constants* formula (Hale & Lunel, 2013). The linear part can be computed analytically, while the remaining nonlinear part is an integral involving the neural network of score function evaluations. Compared with directly numerically solving the diffusion ODE, such decomposition method can reduce numerical errors and improve calculation accuracy because the linear part can be analytically computed, as demonstrated by DPM-Solver (Lu et al., 2022b). We observe that the integral term on the r.h.s. of Eq. (B.4) appears to be a traditional integration problem, involving the score function as the integrand solely. Thus, we could use conventional numerical methods for solving integrals to evaluate it. However, caution must be exercised when employing these methods, as the integrand is merely an approximation of the scaled score function, and its explicit expression remains unknown, while the integrand involves some large-scale neural networks. Therefore, using traditional techniques to accelerate the sampling process of diffusion models may amplify the numerical error in such scenarios. Nonetheless, in the realm of diffusion models, we can draw inspiration from traditional numerical techniques to develop fast sampling algorithms suitable for diffusion models.

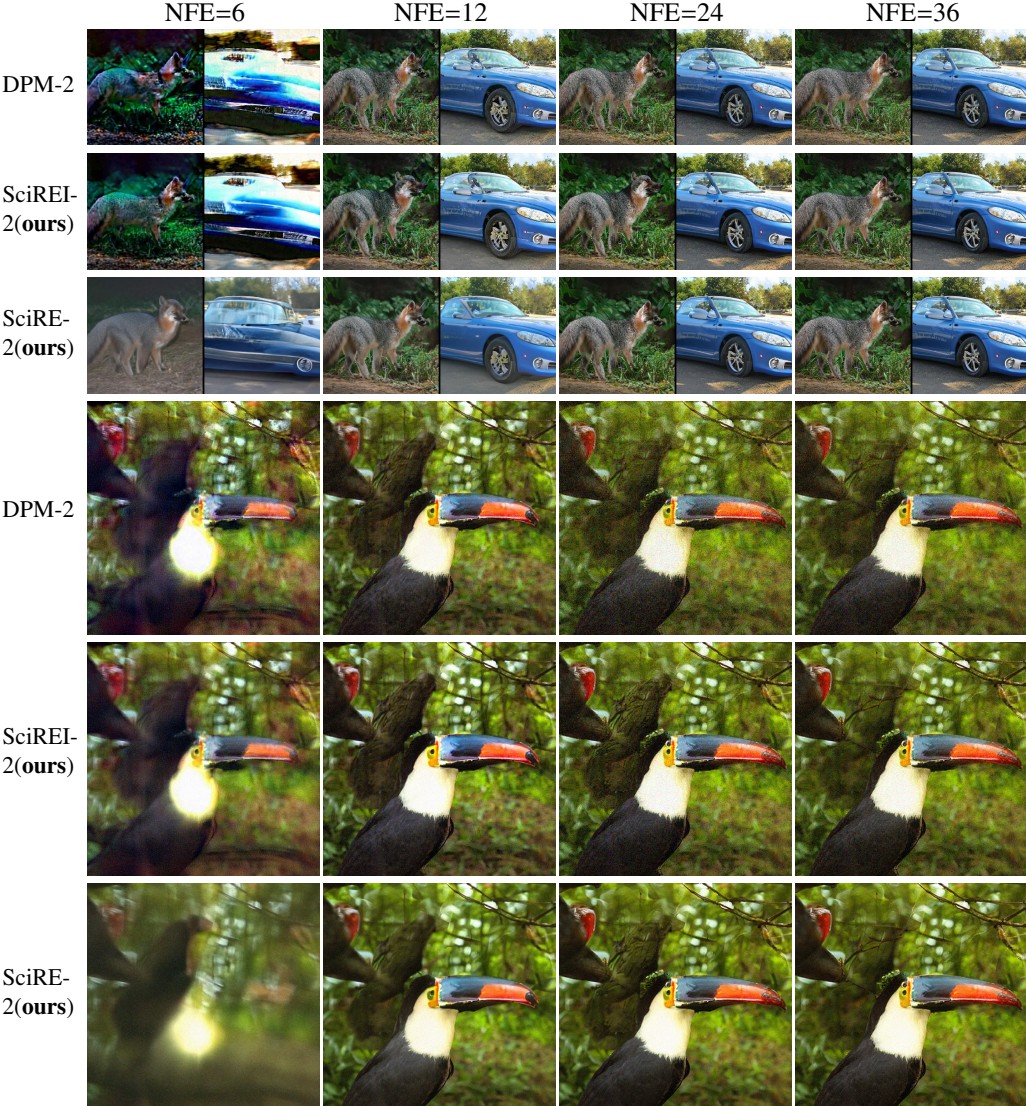

Figure 9: Compare the generation results of the RD-based methods (Solvers: SciRE-V1-2, SciREI-2) and the baseline method (Solver: DPM-2) using 6-36 sampling steps with the uniform time trajectory and identical settings, on pre-trained models with ImageNet 256×256 and 512×512 datasets.

Table 2: Comparison of Quality Generation between FD-based and RD-based Algorithms. We use consistent *NSR* trajectory (**k=2**) and the same codebase.

| FD or RD | Initial time\ NFE | 12 | 15 | 20 | 50 | 100 |
|---|---|---|---|---|---|---|
| The discrete-time model of CIFAR-10 dataset (Ho et al., 2020) | | | | | | |
| FD (Solver-2) | $1e-3$ | 7.00 | †6.00 | 4.76 | 4.10 | 4.03 |
| | $1e-4$ | 9.03 | †7.10 | 5.05 | 3.71 | 3.57 |
| RD (SciRE-V1-2) | $1e-3$ | **4.49** | †**4.12** | **3.74** | 3.70 | 3.76 |
| | $1e-4$ | 5.91 | †4.76 | 3.88 | **3.30** | **3.28** |
| FD (Solver-3) | $1e-3$ | 6.91 | 5.25 | †4.67 | †4.04 | †4.02 |
| | $1e-4$ | 10.19 | 6.12 | †5.03 | †3.56 | †3.52 |
| RD (SciRE-V1-3) | $1e-3$ | **5.29** | 4.19 | †**3.94** | †3.76 | †3.71 |
| | $1e-4$ | 9.10 | 4.52 | †4.07 | †**3.24** | †**3.17** |
| The discrete-time model of CelebA 64×64 dataset (Ho et al., 2020) | | | | | | |
| FD (Solver-2) | $1e-3$ | 7.82 | †6.87 | 5.48 | 4.48 | 4.33 |
| | $1e-4$ | 7.04 | †5.87 | 4.17 | 3.05 | 2.89 |
| RD (SciRE-V1-2) | $1e-3$ | 4.67 | †4.23 | 3.63 | 3.60 | 3.79 |
| | $1e-4$ | **3.99** | †**3.43** | **2.63** | **2.32** | **2.43** |
| FD (Solver-3) | $1e-3$ | 8.09 | 6.29 | †5.35 | †4.28 | †4.27 |
| | $1e-4$ | 7.66 | 5.20 | †4.25 | †2.88 | †2.83 |
| RD (SciRE-V1-3) | $1e-3$ | 4.79 | 3.37 | †3.08 | †3.18 | †3.54 |
| | $1e-4$ | **4.50** | **2.70** | †**2.30** | †**2.02** | **2.20** |

## C   SciREI-Solver, and Compared to DPM-Solver-2 for Assessing the Benefits with RD

In order to further explore the effectiveness of the RD method, we propose a variant named *SciREI-Solver*, which incorporates the RD method and the exponential-based calculation formula provided by DPM-Solver. We provide numerical experiments to demonstrate the benefits of the RD method.

### C.1   SciREI-Solver

In this work, we introduced the RD method to evaluating the derivative of the scaled score function, in light of these results, we proposed the SciRE-V1 with the truncated Taylor expansion of the score-integrand. To further investigate the effectiveness the RD method in the realm of sampling for diffusion models, we apply the RD method to the exponential-based contextualisation provided by the DPM-Solver (Lu et al., 2022b), and proposed the *SciREI-Solver*. We specifically investigate the RD method in the context of "the generalized version of DPM-Solver-2, i.e., the Algorithm 4 in the Appendix of the DPM-Solver paper" (referred to as DPM-Solver-2 throughout this paper for simplicity). Since both SciREI-Solver and DPM-Solver-2 are derived when $n = 2$, we occasionally refer to *SciREI-Solver* as *SciREI-Solver-2* for to enhance clarity in comparisons.

Formally, Lu et al. (2022b) provides an exponential contextualized solution formula for the diffusion ODE:

$$\mathbf{x}_t = \frac{\alpha_t}{\alpha_s}\mathbf{x}_s - \alpha_t \int_{\lambda_s}^{\lambda_t} e^{-\lambda}\hat{\epsilon}_\theta(\hat{\mathbf{x}}_\lambda, \lambda)\,\mathrm{d}\lambda \tag{C.1}$$

where $\lambda_t = \log\frac{\alpha_t}{\sigma_t}$, $\hat{\epsilon}_\theta(\hat{\mathbf{x}}_\lambda, \lambda) = \epsilon_\theta(\mathbf{x}_{t(\lambda)}, t(\lambda))$, and $t(\lambda)$ is the inverse function of $\lambda_t$ w.r.t. time $t$. Under this solution formula of exponential-based contextualisation, the formula below is obtained by Taylor expansion around $\lambda$:

$$\mathbf{x}_t = \frac{\alpha_t}{\alpha_s}\mathbf{x}_s - \sigma_t \sum_{k=0}^{n} h^{k+1}\varphi_{k+1}(h)\hat{\epsilon}_\theta^{(k)}(\hat{\mathbf{x}}_{\lambda_s}, \lambda_s) + O(h^{n+2}), \tag{C.2}$$

where $\varphi_k(h) = \int_0^1 e^{(1-\delta)h}\frac{\delta^{k-1}}{(k-1)!}\mathrm{d}\delta$, $\varphi_0(h) = e^h$.

---

**Algorithm 3** SciREI-Solver (or SciREI-Solver-2)

---

**Require:** initial value $\mathbf{x}_T$, time trajectory $\{t_i\}_{i=0}^N$, model $\epsilon_\theta, m \geq 3$

1: $\tilde{\mathbf{x}}_{t_N} \leftarrow \mathbf{x}_T, r_1 \leftarrow \frac{1}{2}$
2: **for** $i \leftarrow N$ to $0$ **do**
3: $\quad h_i \leftarrow \lambda_{t_{i-1}} - \lambda_{t_i}$
4: $\quad s_i \leftarrow t_\lambda (\lambda_{t_i} + r_1 h_i)$
5: $\quad \tilde{\mathbf{x}}_{s_i} \leftarrow \frac{\alpha_{s_i}}{\alpha_{t_i}} \tilde{\mathbf{x}}_{t_i} - \sigma_{s_i} \left( e^{r_1 h_i} - 1 \right) \epsilon_\theta (\tilde{\mathbf{x}}_{t_i}, t_i)$
6: $\quad \tilde{\mathbf{x}}_{t_{i-1}} \leftarrow \frac{\alpha_{t_{i-1}}}{\alpha_{t_i}} \tilde{\mathbf{x}}_{t_i} - \sigma_{t_{i-1}}(e^{h_i} - 1)\epsilon_\theta(\tilde{\mathbf{x}}_{t_i}, t_i) - \frac{\sigma_{t_{i-1}}}{\phi_1(m) r_1 h_i}(e^{h_i} - h_i - 1) \left( \epsilon_\theta(\tilde{\mathbf{x}}_{s_i}, s_i) - \epsilon_\theta(\tilde{\mathbf{x}}_{t_i}, t_i) \right)$
7: **end for**
**Return:** $\tilde{\mathbf{x}}_0$.

---

When $n = 1$ in Eq. (C.2), we have then

$$\mathbf{x}_t = \frac{\alpha_t}{\alpha_s} \mathbf{x}_s - \sigma_t h \varphi_1(h) \hat{\epsilon}_\theta(\hat{\mathbf{x}}_{\lambda_s}, \lambda_s) - \sigma_t h^2 \varphi_2(h) \hat{\epsilon}_\theta^{(1)}(\hat{\mathbf{x}}_{\lambda_s}, \lambda_s) + O(h^3) \tag{C.3}$$

where

$$\varphi_1(h) = \frac{e^h - 1}{h}, \quad \varphi_2(h) = \frac{e^h - h - 1}{h^2}. \tag{C.4}$$

The following iteration is obtained by DPM-Solver-2:

$$\mathbf{x}_t = \frac{\alpha_t}{\alpha_s} \mathbf{x}_s - \sigma_t(e^h - 1)\hat{\epsilon}_\theta(\hat{\mathbf{x}}_{\lambda_s}, \lambda_s) - \textcolor{blue}{\frac{\sigma_t}{2r_1}(e^h - 1)} \left( \hat{\epsilon}_\theta(\hat{\mathbf{x}}_{\lambda_{s_1}}, \lambda_{s_1}) - \hat{\epsilon}_\theta(\hat{\mathbf{x}}_{\lambda_s}, \lambda_s) \right). \tag{C.5}$$

With our proposed the recursive difference (RD) method to evaluate $\hat{\epsilon}_\theta^{(1)}(\hat{\mathbf{x}}_{\lambda_s}, \lambda_s)$, we get the following new iteration:

$$\mathbf{x}_t = \frac{\alpha_t}{\alpha_s} \mathbf{x}_s - \sigma_t(e^h - 1)\hat{\epsilon}_\theta(\hat{\mathbf{x}}_{\lambda_s}, \lambda_s) - \textcolor{orange}{\frac{\sigma_t}{\phi_1(m) r_1 h}(e^h - h - 1)} \left( \hat{\epsilon}_\theta(\hat{\mathbf{x}}_{\lambda_{s_1}}, \lambda_{s_1}) - \hat{\epsilon}_\theta(\hat{\mathbf{x}}_{\lambda_s}, \lambda_s) \right), \tag{C.6}$$

where the definition of $\phi_1(m)$ is referred to in Corollary 3.1. Thus, we will refer this new iteration algorithm to as *SciREI-Solver* shown in Algorithm 3, which incorporates the RD method and the exponential-based calculation formula recommended by DPM-Solver.

## C.2 DIFFERENCES WITH DPM-SOLVER

Clearly, there are differences between SciREI-Solver and DPM-Solver-2, as indicated by the **blue** and **orange** labels in Eq. (C.5) and Eq. (C.6).

In the following, we present a straightforward comparison between SciRE-V1, proposed by us in the main content of this paper, and DPM-Solver. Firstly, the score-integrand form in Eq. (3.2) is different the solution formula of exponential-based contextualisation in Eq. (C.1). Secondly, different solution forms of diffusion ODE result in different integrands and distinct Taylor series expansions around different function spaces. Specifically, we expand $\epsilon_\theta(\mathbf{x}_{\psi(\tau)}, \psi(\tau))$ in a Taylor series around $\tau$, which is distinct from DPM-Solver where $\epsilon_\theta(\mathbf{x}_{t(\lambda)}, t(\lambda))$ is expanded w.r.t. $\lambda$, as $\tau \neq \lambda$. Such differences lead to different results, as expressed in Eq. 3.5 and Eq. C.2. The differences of Eq. 3.5 and Eq. C.2 illustrate that, despite reparameterizing the diffusion ODE in both cases, different changes-of-variable have resulted in distinct algorithmic sources based on Taylor expansion concept. Finally, and most importantly, SciRE-V1 is a numerical algorithm based on the RD method we introduced, which fundamentally distinguishes it from the DPM-Solver, much like the difference (the blue and orange labels) between SciREI-Solver and DPM-Solver-2.

## C.3 THE BENEFITS WITH RD: EFFECTIVENESS AND ROBUSTNESS

In order to validate the benefits of the RD method, we compare the FID scores obtained for generated samples from the RD-based methods and other methods with the same settings and codebase on the CIFAR-10 and CelebA 64×64 datasets. Specifically, we compare the RD method with the traditional finite difference (FD) method, and we also compare the RD-based SciREI-Solver-2 with its counterpart algorithm, DPM-Solver-2. For high-resolution image datasets, we conduct sampling

Table 3: Comparison of Quality Generation between DPM-Solver-2 and SciREI-Solver-2 (our) Algorithms. We use consistent *NSR* trajectory (**k=3.1**) and the same codebase.

| Trajectory | Initial Time | Sampling method \NFE | 12 | 15 | 20 | 50 | 100 |
|---|---|---|---|---|---|---|---|
| CIFAR-10 (discrete-time model (Ho et al., 2020), linear noise schedule) | | | | | | | |
| Uniform time | $\epsilon = 10^{-3}$ | DPM-Solver-2 | 11.81 | $^\dagger$10.16 | 7.55 | 5.05 | 4.51 |
| | | SciREI-Solver-2 (ours) | **6.65** | $^\dagger$**5.83** | **4.75** | **4.15** | 4.29 |
| Uniform time | $\epsilon = 10^{-4}$ | DPM-Solver-2 | 33.67 | $^\dagger$29.16 | 20.50 | 8.78 | 5.34 |
| | | SciREI-Solver-2 (ours) | 9.93 | $^\dagger$8.53 | 6.50 | 4.51 | **3.98** |
| logSNR | $\epsilon = 10^{-3}$ | DPM-Solver-2 | **5.23** | $^\dagger$**4.48** | 4.08 | 3.98 | 3.99 |
| | | SciREI-Solver-2 (ours) | 5.78 | $^\dagger$4.83 | 4.05 | 3.65 | 3.72 |
| logSNR | $\epsilon = 10^{-4}$ | DPM-Solver-2 | 6.47 | $^\dagger$5.33 | **4.04** | 3.55 | 3.52 |
| | | SciREI-Solver-2 (ours) | 7.06 | $^\dagger$6.03 | 4.33 | **3.33** | **3.28** |
| NSR ($k = 3.1$) | $\epsilon = 10^{-3}$ | DPM-Solver-2 | 5.03 | $^\dagger$4.64 | 4.27 | 4.07 | 4.01 |
| | | SciREI-Solver-2 (ours) | **5.01** | $^\dagger$**4.51** | 3.96 | 3.79 | 3.82 |
| NSR ($k = 3.1$) | $\epsilon = 10^{-4}$ | DPM-Solver-2 | 5.22 | $^\dagger$**4.33** | **3.70** | 3.48 | 3.476 |
| | | SciREI-Solver-2 (ours) | 5.79 | $^\dagger$4.72 | 3.81 | **3.21** | **3.23** |
| CelebA 64×64 (discrete-time model (Song et al., 2021a), linear noise schedule) | | | | | | | |
| Uniform time | $\epsilon = 10^{-3}$ | DPM-Solver-2 | 15.23 | $^\dagger$13.63 | 10.99 | 7.41 | 5.46 |
| | | SciREI-Solver-2 (ours) | **7.25** | $^\dagger$**6.93** | **6.32** | **5.72** | 5.06 |
| Uniform time | $\epsilon = 10^{-4}$ | DPM-Solver-2 | 61.90 | $^\dagger$53.40 | 38.19 | 16.98 | 7.24 |
| | | SciREI-Solver-2 (ours) | 20.05 | $^\dagger$17.98 | 14.24 | 7.93 | **4.40** |
| logSNR | $\epsilon = 10^{-3}$ | DPM-Solver-2 | 3.97 | $^\dagger$3.96 | 4.07 | 4.22 | 4.25 |
| | | SciREI-Solver-2 (ours) | 3.78 | $^\dagger$3.58 | 3.33 | 3.41 | 3.69 |
| logSNR | $\epsilon = 10^{-4}$ | DPM-Solver-2 | **3.27** | $^\dagger$**3.13** | 2.90 | 2.80 | 2.799 |
| | | SciREI-Solver-2 (ours) | 3.44 | $^\dagger$3.18 | **2.71** | **2.29** | **2.39** |
| NSR ($k = 3.1$) | $\epsilon = 10^{-3}$ | DPM-Solver-2 | 6.24 | $^\dagger$5.54 | 4.72 | 4.27 | 4.24 |
| | | SciREI-Solver-2 (ours) | 4.97 | $^\dagger$4.48 | 3.81 | 3.59 | 3.79 |
| NSR ($k = 3.1$) | $\epsilon = 10^{-4}$ | DPM-Solver-2 | 3.67 | $^\dagger$3.04 | 2.79 | 2.77 | 2.78 |
| | | SciREI-Solver-2 (ours) | **3.30** | $^\dagger$**2.98** | **2.56** | **2.25** | **2.39** |

comparisons under the same settings and codebase for ImageNet at resolutions of 128×128, 256×256, and 512×512, as well as for the LSUN bedroom dataset at a resolution of 256×256, due to server limitations. To ensure fairness in our experiments, we maintain the same settings and codebase for each sampling algorithm to evaluate the various methods of RD-based and RD-none.

Firstly, we use FID to measure the sampling performance of the sampling algorithms when estimating derivatives using finite difference (*FD*) method and recursive derivative (*RD*) method, respectively. Here, we set $\phi_1(m) = 1$ in SciRE-V1 to represent the sampling algorithm based on FD method. Based on the FID metric of generated samples, with the same codebase, we assess the performance of these two derivative estimation methods using discrete diffusion models trained on the CIFAR-10 and CelebA 64×64 datasets. Without loss of generality, we use a consistent NSR trajectory with $k = 2$, because SciRE-V1 can achieve the better quality of generated samples for $k \in [2, 7]$, as mentioned in Section H.2. Our numerical experiments demonstrate that, across different initial times, the quality of generated samples achieved by the SciRE-V1 using RD method consistently outperforms that of the solver using FD method, as shown in Table 2. These numerical experiments measured by FID demonstrate that in the domain of diffusion ODEs, the use of the RD method for estimating derivatives of score function evaluation networks consistently outperforms the traditional FD method.

Secondly, for further investigation of the RD method, we introduce SciREI-Solver ($n = 2$) in section C.1, referring to it as SciREI-Solver-2 to align it in form with its counterpart, DPM-Solver-2. We compare the generative performance of SciREI-Solver-2 and DPM-Solver-2 with the identical settings on the CIFAR-10 and CelebA 64×64 datasets using various time trajectories and termination times, as illustrated in Table 3. Table 3 demonstrates that SciREI-Solver-2 based on the RD method exhibits greater robustness than the DPM-Solver-2 across different time trajectories, especially on the CelebA 64×64 dataset. These experiments measured by FID also simultaneously demonstrate that as NFE increases, SciREI-Solver-2 based on the RD method consistently outperforms its counterpart DPM-

Solver-2. Next, we will conduct some sampling comparison experiments for both of SciREI-Solver-2 and the DPM-Solver-2.

Thirdly, we provide the sampling comparisons between the RD-based sampling algorithms (including SciRE-V1-2 and SciREI-Solver-2) and the baseline algorithm (DPM-Solver-2) on high-resolution image datasets. In Figure 5, we compare the generation results of the RD-based methods (Solvers: SciRE-2, SciREI-2) and the baseline method (Solver: DPM-2) using 6-36 sampling steps with the uniform time trajectory and identical settings, on pre-trained models with ImageNet 128×128 and LSUN bedroom 256×256. Here, we further compare the generation results of the RD-based methods and the baseline method on pre-trained models with ImageNet 256×256 and 512×512 datasets, using 6-36 sampling steps with the uniform time trajectory and identical settings. In these experiments, we can observe that when using 36 NFEs, samples generated by the popular DPM-Solver-2 still exhibit more noise compared to our proposed SciREI-Solver-2 and SciRE-V1-2 based on the RD method. Therefore, all these sampling experiments on high-resolution image datasets also demonstrate the effectiveness of the RD method.

In summary, all experiments-above under the same settings and codebase indicate that the RD method brings benefits to Taylor-based numerical algorithms in the realm of diffusion ODEs. Therefore, we strongly recommend using the RDE method, if the sampling algorithms require evaluating the derivative of the score function evaluation networks.

## D  PROOF OF THEOREM 3.1 AND COROLLARY 1

### D.1  PRELIMINARIES

Throughout this section, we denote $\text{NSR}_{\min} := \min_i \{\text{NSR}(t_i)\}$, $\text{NSR}_{\max} := \max_i \{\text{NSR}(t_i)\}$, and assume that $\epsilon_\theta\left(\mathbf{x}_{\psi(\tau)}, \psi(\tau)\right) \in \mathbb{C}^\infty[\text{NSR}_{\min}, \text{NSR}_{\max}]$, which means that the total derivatives $\frac{\mathrm{d}^k \epsilon_\theta(\mathbf{x}_{\psi(\tau)}, \psi(\tau))}{\mathrm{d}\tau^k}$ exist and are continuous for $k \in \mathbb{Z}_+$. Notice that $\tau := \text{NSR}(t)$, $\psi(\tau) := \text{rNSR}(\tau)$ and the reverse function of NSR, i.e. rNSR, satisfying $t = \text{rNSR}(\text{NSR}(t)) = \psi(\tau)$. Denote $h_s := \text{NSR}(t) - \text{NSR}(s) = \tau_t - \tau_s$, and $\epsilon_\theta^{(k)}\left(\mathbf{x}_{\psi(\tau)}, \psi(\tau)\right) := \frac{\mathrm{d}^k \epsilon_\theta(\mathbf{x}_{\psi(\tau)}, \psi(\tau))}{\mathrm{d}\tau^k}$ as $k$-th order total derivative of $\epsilon_\theta\left(\mathbf{x}_{\psi(\tau)}, \psi(\tau)\right)$ w.r.t. $\tau$. For $n \geq 1$, the $n$-th order Taylor expansion of $\epsilon_\theta\left(\mathbf{x}_{\psi(\tau_t)}, \psi(\tau_t)\right)$ w.r.t. $\tau$ at $\tau_s$ is

$$\epsilon_\theta\left(\mathbf{x}_{\psi(\tau_t)}, \psi(\tau_t)\right) = \sum_{k=0}^{n} \frac{h_s^k}{k!} \epsilon_\theta^{(k)}\left(\mathbf{x}_{\psi(\tau_s)}, \psi(\tau_s)\right) + O(h_s^{n+1}). \tag{D.1}$$

For any $k \geq 0$, we can approximate the $k$-th order total derivative term $\epsilon_\theta^{(k)}\left(\mathbf{x}_{\psi(\tau_s)}, \psi(\tau_s)\right)$ in Eq. (D.1) by using the first-order difference formula:

$$\epsilon_\theta^{(k)}\left(\mathbf{x}_{\psi(\tau_s)}, \psi(\tau_s)\right) = \frac{\epsilon_\theta^{(k-1)}\left(\mathbf{x}_{\psi(\tau_t)}, \psi(\tau_t)\right) - \epsilon_\theta^{(k-1)}\left(\mathbf{x}_{\psi(\tau_s)}, \psi(\tau_s)\right)}{h_s} - O(h_s). \tag{D.2}$$

For ease of notation, we denote $\epsilon_\theta^{(k)}\left(\mathbf{x}_{\psi(\tau)}, \psi(\tau)\right)$ as $\Gamma^{(k)}(\tau)$. Notice that $\epsilon_\theta\left(\mathbf{x}_{\psi(\tau)}, \psi(\tau)\right) = \Gamma^{(0)}(\tau)$. Then Eq. (D.2) can be represented as:

$$\Gamma^{(k)}(\tau_s) = \frac{\Gamma^{(k-1)}(\tau_t) - \Gamma^{(k-1)}(\tau_s)}{h_s} - O(h_s). \tag{D.3}$$

### D.2  PROOF OF THEOREM 3.1

*Proof.* While $n \to \infty$, Eq. (D.1) becomes:

$$\begin{aligned}
\Gamma^{(0)}(\tau_t) &= \sum_{k=0}^{\infty} \frac{h_s^k}{k!} \Gamma^{(k)}(\tau_s) \\
&= \Gamma^{(0)}(\tau_s) + \sum_{k=1}^{\infty} \frac{h_s^k}{k!} \Gamma^{(k)}(\tau_s).
\end{aligned} \tag{D.4}$$

Moving $\Gamma^{(0)}(\tau_s)$ from the right-hand side of the above equation to the left-hand side and then dividing both sides of the equation by $h_s$, we can obtain:

$$
\begin{aligned}
&\frac{\Gamma^{(0)}(\tau_t) - \Gamma^{(0)}(\tau_s)}{h_s} \\
&= \sum_{k=1}^{\infty} \frac{h_s^{k-1}}{k!} \Gamma^{(k)}(\tau_s) \\
&= \Gamma^{(1)}(\tau_s) + \sum_{k=2}^{\infty} \frac{h_s^{k-1}}{k!} \Gamma^{(k)}(\tau_s) \\
&= \Gamma^{(1)}(\tau_s) + \sum_{k=2}^{\infty} \frac{h_s^{k-1}}{k!} \left( \frac{\Gamma^{(k-1)}(\tau_t) - \Gamma^{(k-1)}(\tau_s)}{h_s} - O(h_s) \right) \\
&= \Gamma^{(1)}(\tau_s) + \sum_{k=2}^{\infty} \frac{h_s^{k-2}}{k!} \left( \Gamma^{(k-1)}(\tau_t) - \Gamma^{(k-1)}(\tau_s) \right) - \underbrace{\sum_{k=2}^{\infty} \frac{h_s^{k-1}}{k!} O(h_s)}_{\text{remainder } Q} \\
&= \Gamma^{(1)}(\tau_s) - \sum_{k=2}^{\infty} \frac{h_s^{k-2}}{k!} \Gamma^{(k-1)}(\tau_s) + \underbrace{\sum_{k=2}^{\infty} \frac{h_s^{k-2}}{k!} \Gamma^{(k-1)}(\tau_t)}_{R_1} - Q \\
&= \left( 1 - \frac{1}{2} \right) \Gamma^{(1)}(\tau_s) - \sum_{k=3}^{\infty} \frac{h_s^{k-2}}{k!} \Gamma^{(k-1)}(\tau_s) + R_1 - Q \\
&= \left( 1 - \frac{1}{2} \right) \Gamma^{(1)}(\tau_s) - \sum_{k=3}^{\infty} \frac{h_s^{k-3}}{k!} \left( \Gamma^{(k-2)}(\tau_t) - \Gamma^{(k-2)}(\tau_s) - O(h_s^2) \right) + R_1 - Q \\
&= \left( 1 - \frac{1}{2} \right) \Gamma^{(1)}(\tau_s) + \sum_{k=3}^{\infty} \frac{h_s^{k-3}}{k!} \Gamma^{(k-2)}(\tau_s) - \underbrace{\sum_{k=3}^{\infty} \frac{h_s^{k-3}}{k!} \Gamma^{(k-2)}(\tau_t)}_{R_2} - O(h_s^2) + R_1 \\
&= \left( 1 - \frac{1}{2} + \frac{1}{6} \right) \Gamma^{(1)}(\tau_s) + \sum_{k=4}^{\infty} \frac{h_s^{k-3}}{k!} \Gamma^{(k-2)}(\tau_s) + R_2 + R_1 - O(h_s^2) \\
&= \left( 1 - \frac{1}{2!} + \frac{1}{3!} - \frac{1}{4!} \right) \Gamma^{(1)}(\tau_s) - \sum_{k=5}^{\infty} \frac{h_s^{k-4}}{k!} \Gamma^{(k-3)}(\tau_s) + R_1 + R_2 + R_3 - O(h_s^2) \\
&\qquad \cdots \\
&= \sum_{k=1}^{\infty} \frac{(-1)^{k-1}}{k!} \Gamma^{(1)}(\tau_s) + \sum_{i=1}^{\infty} R_i - O(h_s^2),
\end{aligned}
\tag{D.5}
$$

where

$$
\begin{aligned}
R_1 &= \sum_{k=2}^{\infty} \frac{h_s^{k-2}}{k!} \Gamma^{(k-1)}(\tau_t) \\
R_2 &= -\sum_{k=3}^{\infty} \frac{h_s^{k-3}}{k!} \Gamma^{(k-2)}(\tau_t) \\
R_3 &= \sum_{k=4}^{\infty} \frac{h_s^{k-4}}{k!} \Gamma^{(k-3)}(\tau_t) \\
&\qquad \cdots \\
R_i &= (-1)^{i+1} \sum_{k=i+1}^{\infty} \frac{h_s^{k-i-1}}{k!} \Gamma^{(k-i)}(\tau_t), \quad i \in \mathbb{Z}_+.
\end{aligned}
\tag{D.6}
$$

Adding the first term and second term of $R_i, i \in \mathbb{Z}_+$ separately, we can derive

$$\sum_{i=1}^{\infty} R_i = \sum_{k=2}^{\infty} \frac{(-1)^k}{k!} \Gamma^{(1)}(\tau_t) + \sum_{k=3}^{\infty} \frac{(-1)^{k+1} h_s}{k!} \Gamma^{(2)}(\tau_t) + \underbrace{\sum_{i=3}^{\infty} \sum_{k=i+1}^{\infty} \frac{(-1)^{k+i-1} h_s^{i-1}}{k!} \Gamma^{(i)}(\tau_t)}_{O(h_s^2)}. \tag{D.7}$$

Notice that

$$\sum_{i=3}^{\infty} \sum_{k=i+1}^{\infty} \frac{(-1)^{k+i-1} h_s^{i-1}}{k!} \Gamma^{(i)}(\tau_t) = \sum_{i=3}^{\infty} (-1)^{i-1} h_s^{i-1} \Gamma^{(i)}(\tau_t) \sum_{k=i+1}^{\infty} \frac{(-1)^k}{k!} = O(h_s^2), \tag{D.8}$$

because $\sum_{k=i+1}^{\infty} \frac{(-1)^k}{k!}, \forall i \in \mathbb{Z}_+$ are all convergent alternating series which can be easily proved with Leibniz's test. Then Eq. (D.5) can be shown as:

$$\frac{\Gamma^{(0)}(\tau_t) - \Gamma^{(0)}(\tau_s)}{h_s} = \sum_{k=1}^{\infty} \frac{(-1)^{k-1}}{k!} \Gamma^{(1)}(\tau_s) + \sum_{i=1}^{2} \sum_{k=i+1}^{\infty} \frac{(-1)^{k+i-1} h_s^{i-1}}{k!} \Gamma^{(i)}(\tau_t) - O(h_s^2). \tag{D.9}$$

Consequently, we can get $\Gamma^{(1)}(\tau_s)$ by simple manipulation of rearranging and affine transformation applied to above equation:

$$\begin{aligned}
\Gamma^{(1)}(\tau_s) &= \frac{e}{e-1} \frac{\Gamma^{(0)}(\tau_t) - \Gamma^{(0)}(\tau_s)}{h_s} - \frac{e}{e-1} \sum_{i=1}^{2} \sum_{k=i+1}^{\infty} \frac{(-1)^{k+i-1} h_s^{i-1}}{k!} \Gamma^{(i)}(\tau_t) + O(h_s^2) \\
&= \frac{e}{e-1} \frac{\Gamma^{(0)}(\tau_t) - \Gamma^{(0)}(\tau_s)}{h_s} - \frac{e}{e-1} \left( \frac{1}{e} \Gamma^{(1)}(\tau_t) + \frac{e-2}{2e} h_s \Gamma^{(2)}(\tau_t) \right) + O(h_s^2) \\
&= \frac{e}{e-1} \frac{\Gamma^{(0)}(\tau_t) - \Gamma^{(0)}(\tau_s)}{h_s} - \frac{\Gamma^{(1)}(\tau_t)}{e-1} - \frac{(e-2)h_s}{2(e-1)} \Gamma^{(2)}(\tau_t) + O(h_s^2),
\end{aligned} \tag{D.10}$$

where

$$\begin{aligned}
\sum_{k=1}^{\infty} \frac{(-1)^{k-1}}{k!} &= 1 - e^{-1} \\
\sum_{k=2}^{\infty} \frac{(-1)^k}{k!} &= e^{-1} \\
\sum_{k=3}^{\infty} \frac{(-1)^{k+1}}{k!} &= -e^{-1} + \frac{1}{2} = \frac{e-2}{2e},
\end{aligned} \tag{D.11}$$

for $e^x = \sum_{k=0}^{\infty} \frac{x^k}{k!}$ with $x = -1$.

Notice that we denote $\epsilon_\theta^{(k)} \left( \mathbf{x}_{\psi(\tau)}, \psi(\tau) \right)$ as $\Gamma^{(k)}(\tau)$. While using $\tilde{\mathbf{x}}$ to approximate $\mathbf{x}$ and replacing the terms like $\Gamma^{(k)}(\tau)$ in Eq. (D.10) with terms like $\epsilon_\theta^{(k)} \left( \mathbf{x}_{\psi(\tau)}, \psi(\tau) \right)$, we can get the results shown in Theorem 3.1. □

### D.3 Proof of Corollary 1

We observe that the differentiability constraint imposed by Theorem 3.1 appears to be rather restrictive. In order to enhance its broad applicability, we further propose a recursive derivative estimation method under the assumption of limited differentiability. The corresponding proof process is as follows:

*Proof.* Assume that $\epsilon_\theta \left( \mathbf{x}_{\psi(\tau)}, \psi(\tau) \right) \in \mathbb{C}^n [\text{NSR}_{\min}, \text{NSR}_{\max}]$. While $n$ is a finite positive integer, Eq. (D.1) becomes:

$$\begin{aligned}
\Gamma^{(0)}(\tau_t) &= \sum_{k=0}^{n} \frac{h_s^k}{k!} \Gamma^{(k)}(\tau_s) + O(h_s^{n+1}) \\
&= \Gamma^{(0)}(\tau_s) + \sum_{k=1}^{n} \frac{h_s^k}{k!} \Gamma^{(k)}(\tau_s) + O(h_s^{n+1}).
\end{aligned} \tag{D.12}$$

Same as the derivation process in (D.5), we can obtain

$$
\begin{aligned}
&\frac{\Gamma^{(0)}(\tau_t) - \Gamma^{(0)}(\tau_s)}{h_s} \\
&= \sum_{k=1}^{n} \frac{h_s^{k-1}}{k!}\Gamma^{(k)}(\tau_s) + O(h_s^n) \\
&= \Gamma^{(1)}(\tau_s) + \sum_{k=2}^{n} \frac{h_s^{k-1}}{k!}\Gamma^{(k)}(\tau_s) + O(h_s^n) \\
&= \Gamma^{(1)}(\tau_s) - \sum_{k=2}^{n} \frac{h_s^{k-2}}{k!}\Gamma^{(k-1)}(\tau_s) + \underbrace{\sum_{k=2}^{n} \frac{h_s^{k-2}}{k!}\Gamma^{(k-1)}(\tau_t)}_{R_1} - \underbrace{O(h_s^2) + O(h_s^n)}_{O(h_s^2)-O(h_s^n)=O(h_s^2)} \\
&= \left(1 - \frac{1}{2}\right)\Gamma^{(1)}(\tau_s) + \sum_{k=3}^{n} \frac{h_s^{k-3}}{k!}\Gamma^{(k-2)}(\tau_s) - \underbrace{\sum_{k=3}^{n} \frac{h_s^{k-3}}{k!}\Gamma^{(k-2)}(\tau_t)}_{R_2} - O(h_s^2) + R_1 \\
&= \left(1 - \frac{1}{2} + \frac{1}{6}\right)\Gamma^{(1)}(\tau_s) - \sum_{k=4}^{n} \frac{h_s^{k-4}}{k!}\Gamma^{(k-3)}(\tau_s) + \underbrace{\sum_{k=4}^{n} \frac{h_s^{k-4}}{k!}\Gamma^{(k-3)}(\tau_t)}_{R_3} - O(h_s^2) + R_1 + R_2 \\
&\qquad\qquad\qquad\qquad \dots \\
&= \sum_{k=1}^{n} \frac{(-1)^{k-1}}{k!}\Gamma^{(1)}(\tau_s) + \sum_{i=1}^{n-1} R_i - O(h_s^2),
\end{aligned}
\tag{D.13}
$$

where

$$
\begin{aligned}
R_1 &= \sum_{k=2}^{n} \frac{h_s^{k-2}}{k!}\Gamma^{(k-1)}(\tau_t) \\
R_2 &= -\sum_{k=3}^{n} \frac{h_s^{k-3}}{k!}\Gamma^{(k-2)}(\tau_t) \\
R_3 &= \sum_{k=4}^{n} \frac{h_s^{k-4}}{k!}\Gamma^{(k-3)}(\tau_t) \\
&\qquad \dots \\
R_{n-1} &= (-1)^n \sum_{k=n}^{n} \frac{h_s^{k-n}}{k!}\Gamma^{(k-n+1)}(\tau_t).
\end{aligned}
\tag{D.14}
$$

Hence adding the first term and second term of $R_i, i = 1, 2, \dots, n-1$ separately, we also have

$$
\begin{aligned}
\sum_{i=1}^{n-1} R_i &= \sum_{k=2}^{n} \frac{(-1)^k}{k!}\Gamma^{(1)}(\tau_t) + \sum_{k=3}^{n} \frac{(-1)^{k+1}h_s}{k!}\Gamma^{(2)}(\tau_t) + \sum_{i=3}^{n-1}\sum_{k=i+1}^{n} \frac{(-1)^{k+i-1}h_s^{i-1}}{k!}\Gamma^{(i)}(\tau_t) \\
&= \sum_{k=2}^{n} \frac{(-1)^k}{k!}\Gamma^{(1)}(\tau_t) + \sum_{k=3}^{n} \frac{(-1)^{k+1}h_s}{k!}\Gamma^{(2)}(\tau_t) + O(h_s^2).
\end{aligned}
\tag{D.15}
$$

Denote $\phi_1(n) = \sum_{k=1}^{n} \frac{(-1)^{k-1}}{k!}$, $\phi_2(n) = \sum_{k=2}^{n} \frac{(-1)^k}{k!}$, and $\phi_3(n) = \sum_{k=3}^{n} \frac{(-1)^{k+1}}{k!}$. Combining (D.13) and (D.15), we can easily derive $\Gamma(\tau_s, 1)$ by

$$
\Gamma^{(1)}(\tau_s) = \frac{1}{\phi_1(n)}\frac{\Gamma^{(0)}(\tau_t) - \Gamma^{(0)}(\tau_s)}{h_s} - \frac{\phi_2(n)}{\phi_1(n)}\Gamma^{(1)}(\tau_t) - \frac{\phi_3(n)h_s}{\phi_1(n)}\Gamma^{(2)}(\tau_t) + O(h_s^2).
\tag{D.16}
$$

Similarly, while using $\tilde{\mathbf{x}}$ to approximate $\mathbf{x}$ and replacing the terms like $\Gamma^{(k)}(\tau)$ in Eq. (D.16) with terms like $\epsilon_\theta^{(k)}\left(\mathbf{x}_{\psi(\tau)}, \psi(\tau)\right)$, we can get the results shown in Corollary 3.1. $\qquad\square$

# E    ANALYSIS OF RECURSIVE DIFFERENCE

In this Section, we provide some analysis of recursive difference (RD) method. Taking practical considerations into account for the iterative algorithm, we first introduce a simplified truncation of the RD method. Subsequently, we conduct a three-tier analysis of this simplified truncation.

## E.1    THE MOST SIMPLIFIED TRUNCATION OF RECURSIVE DIFFERENCE

We review that the $n$-th order Taylor expansion of $\epsilon_\theta(\mathbf{x}_{\psi(\tau_s)}, \psi(\tau_s))$ w.r.t. $\tau$ at $\tau_t$ is

$$\epsilon_\theta\left(\mathbf{x}_{\psi(\tau_s)}, \psi(\tau_s)\right) = \sum_{k=0}^{n} \frac{h_t^k}{k!} \epsilon_\theta^{(k)}\left(\mathbf{x}_{\psi(\tau_t)}, \psi(\tau_t)\right) + O(h_t^{n+1}). \tag{E.1}$$

After substituting this Taylor expansion into the score-integrand form presented in Eq. (3.2) for the diffusion ODEs, we derive

$$\mathbf{x}_s = \frac{\alpha_s}{\alpha_t}\mathbf{x}_t + \alpha_s \sum_{k=0}^{n} \frac{h_t^{k+1}}{(k+1)!} \epsilon_\theta^{(k)}\left(\mathbf{x}_{\psi(\tau_t)}, \psi(\tau_t)\right) + O(h_t^{n+2}). \tag{E.2}$$

When $n = 2$, we have then the following truncation formula:

$$\tilde{\mathbf{x}}_s = \frac{\alpha_s}{\alpha_t}\mathbf{x}_t + \alpha_s\left(h_t\epsilon_\theta(\mathbf{x}_{\psi(\tau_t)}, \psi(\tau_t)) + \frac{h_t^2}{2}\epsilon_\theta^{(1)}(\mathbf{x}_{\psi(\tau_t)}, \psi(\tau_t)) + \frac{h_t^3}{6}\epsilon_\theta^{(2)}(\mathbf{x}_{\psi(\tau_t)}, \psi(\tau_t)) + O(h_t^4)\right), \tag{E.3}$$

where $\tilde{\mathbf{x}}_s$ represents the approximate value of $\mathbf{x}_s$.

By the RD method in Corollary 3.1, we have

$$\begin{aligned}\epsilon_\theta^{(1)}\left(\mathbf{x}_{\psi(\tau_t)}, \psi(\tau_t)\right) &= \frac{1}{\phi_1(m)}\frac{\epsilon_\theta\left(\tilde{\mathbf{x}}_{\psi(\tau_{s1})}, \psi(\tau_{s1})\right) - \epsilon_\theta\left(\mathbf{x}_{\psi(\tau_t)}, \psi(\tau_t)\right)}{r_1 h_t} \\ &\quad - \frac{\phi_2(m)}{\phi_1(m)}\epsilon_\theta^{(1)}\left(\mathbf{x}_{\psi(\tau_{s1})}, \psi(\tau_{s1})\right) - \frac{\phi_3(m)r_1 h_t}{\phi_1(m)}\epsilon_\theta^{(2)}\left(\mathbf{x}_{\psi(\tau_{s1})}, \psi(\tau_{s1})\right) + O(h_t^2).\end{aligned} \tag{E.4}$$

where $\tau_{s1} - \tau_t = r_1 h_t$. Combining (E.3) with Eq. (E.4), we have

$$\begin{aligned}\tilde{\mathbf{x}}_s &= \frac{\alpha_s}{\alpha_t}\mathbf{x}_t + \alpha_s h_t\epsilon_\theta\left(\mathbf{x}_{\psi(\tau_t)}, \psi(\tau_t)\right) + \alpha_s\frac{h_t^2}{2}\frac{\epsilon_\theta\left(\tilde{\mathbf{x}}_{\psi(\tau_{s1})}, \psi(\tau_{s1})\right) - \epsilon_\theta\left(\mathbf{x}_{\psi(\tau_t)}, \psi(\tau_t)\right)}{\phi_1(m)r_1 h_t} \\ &\quad - \alpha_s\frac{h_t^2}{2}\frac{\phi_2(m)}{\phi_1(m)}\epsilon_\theta^{(1)}\left(\mathbf{x}_{\psi(\tau_{s1})}, \psi(\tau_{s1})\right) + \alpha_s h_t^3\left(\frac{1}{6} - \frac{\phi_3(m)r_1}{2\phi_1(m)}\right)\epsilon_\theta^{(2)}\left(\mathbf{x}_{\psi(\tau_{s1})}, \psi(\tau_{s1})\right) \\ &\quad + O(h_t^4).\end{aligned} \tag{E.5}$$

By truncating the term containing $h_t^2$ in Eq. (E.5), we obtain the algorithm shown in Algorithm 1:

$$\tilde{\mathbf{x}}_s \leftarrow \frac{\alpha_s}{\alpha_t}\mathbf{x}_t + \alpha_s h_t\epsilon_\theta\left(\mathbf{x}_{\psi(\tau_t)}, \psi(\tau_t)\right) + \alpha_s\frac{h_t}{2}\frac{\epsilon_\theta\left(\tilde{\mathbf{x}}_{\psi(\tau_{s1})}, \psi(\tau_{s1})\right) - \epsilon_\theta\left(\mathbf{x}_{\psi(\tau_t)}, \psi(\tau_t)\right)}{\phi_1(m)r_1}. \tag{E.6}$$

According to Eq. (E.5), it appears that we can easily conclude from Eq. (E.6) that the preliminary result has a local truncation error of $O(h_t^2)$. Nevertheless, our numerical experiments conducted on different datasets demonstrate that the algorithm 1 derived through this truncation method can generate high-quality samples with a restricted number of score function evaluations (NFE). Further details can be found in Appendix H. After careful observation, in fact, this technique partially eliminates the dependence on derivatives while still containing derivative-related information (such as the difference between two score function evaluations), thereby mitigating the error propagation caused by derivative estimation to a certain extent. By repeatedly using this technique, we can derive our SciRE-V1-3 in Algorithm 2.

Note that the truncation method in Eq. (E.6) is just a simplified truncation for RD method. In fact, with an even more accurate estimate for $\epsilon_\theta^{(1)}\left(\mathbf{x}_{\psi(\tau_{s1})}, \psi(\tau_{s1})\right)$, we can further truncate the term containing $h_t^3$, leading to a generalized SciRE-Solver. We leave it for future study.

E.2    ANALYSIS OF THE MOST SIMPLIFIED TRUNCATION IN RD

Generally, the expression for the first-order derivative can be written as follows by utilizing Eq. (E.1):

$$\Gamma^{(1)}(\tau_{t_i}) = \frac{\Gamma(\tau_{t_{i-1}}) - \Gamma(\tau_{t_i})}{h_{t_i}} - \sum_{k=2}^{n} \frac{h_{t_i}^{k-1}}{k!} \Gamma^{(k)}(\tau_{t_i}) + O(h_{t_i}^n), \tag{E.7}$$

where $\Gamma^{(k)}(\tau)$ is used to denote $\epsilon_\theta^{(k)}\left(\mathbf{x}_{\psi(\tau)}, \psi(\tau)\right)$ for simplicity. By simply truncating all higher-order terms, the FD approximation can be obtained:

$$\Gamma^{(1)}(\tau_{t_i}) \approx \frac{\Gamma(\tau_{t_{i-1}}) - \Gamma(\tau_{t_i})}{h_{t_i}}, \tag{E.8}$$

where we refer to *the coefficient $\frac{1}{h_{t_i}}$ of the term $\Gamma(\tau_{t_{i-1}}) - \Gamma(\tau_{t_i})$ as the coefficient of FD* for ease of description in the rest of this section. Clearly, the FD method exhibits an approximate error of $O(h_{t_i})$. In the preceding section, we elucidated the simplified truncation form of the RD method, which approximates the first-order derivative in the following manner:

$$\Gamma^{(1)}(\tau_{t_i}) \approx \frac{1}{\phi_1(m)} \frac{\Gamma(\tau_{t_{i-1}}) - \Gamma(\tau_{t_i})}{h_{t_i}}. \tag{E.9}$$

In the following, we will conduct a three-tier analysis of this simplified truncation form.

Firstly, we observe that it is common to replace $(e^h - h - 1)/h^2$ with $(e^h - 1)/h$ as the new FD coefficient within the framework of exponential integrators. We demonstrate that the coefficient of the simplified RD method exhibits numerical trends similar to the coefficient $(e^h - 1)/h$ employed in the exponential integrator. Clearly, $(e^h - 1)/h = (he^h - h)/h^2$. Due to the fundamental difference between their numerators, we only need to consider the numerical behavior between these numerators. Let $\hat{g}(h) = (he^h - h) - (e^h - h - 1)$. Clearly, $\hat{g}(h) = he^h - e^h + 1$ and $\hat{g}^{(1)}(h) = he^h$. Then, $\hat{g}(h)$ reaches its minimum value at $h = 0$, i.e., $\hat{g}(0) = 0$. Then $\hat{g}(h) > 0$ and $(e^h - 1)/h > (e^h - h - 1)/h^2$ when $h \neq 0$. This means that the essence of using $(e^h - 1)/h$ in the exponential integrator algorithm is to appropriately amplify the coefficients of the FD method. Besides, since $\phi_1(m) < 1$, the RD approximation in Eq. (E.9) also amplifies the coefficients of FD method. Therefore, the RD method and the equivalent infinitesimal substitution in the exponential integrator share the same numerical trend. This explains the numerical commonality between different coefficients in Eq. (C.5) and Eq. (C.6). In Figures 5 and 9, the sampling comparisons for DPM-2 and SciREI-2 vividly illustrate this numerical commonality.

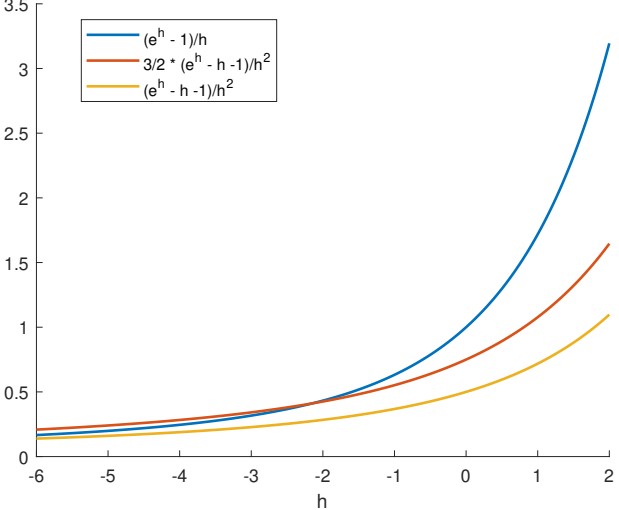

Figure 10: Comparison of coefficients between different difference methods. The red curve represents the recursive difference coefficient $\phi_1(m)$ at $m = 3$. The blue and yellow curves represent the coefficients used in the exponential integrator (DPM-Solver and DEIS) and finite difference, respectively.

Secondly, we present the numerical performance of coefficients among different difference methods, as shown in Figure 10. Clearly, RD amplifies the coefficients of FD, but overall maintains a similar trend to FD method. It can be seen as a numerical stretching of FD by a recursive factor. When $h$ is large, the coefficients based on the exponential integrator exhibit a more pronounced increasing trend in the positive direction of $h$ compared to RD. In the negative direction of $h$, two scenarios arise: on the left side of the intersection point between the RD coefficient function and the coefficient function employed by the exponential integrator, the RD coefficient value is slightly greater than the value employed by the exponential integrator; on the right side of the intersection point, the coefficient value employed by the exponential integrator is noticeably greater than the RD coefficient value. This difference explain why the algorithm based on the coefficient of the exponential integrator exhibits over-rendering in a small number of steps, as depicted in Figure 5. Another notable difference is that, as $h$ approaches 0, both the coefficient and its rate of change for the difference terms using the exponential integrator surpass those for the RD counterparts. This difference clarifies why algorithms based on the coefficients of the exponential integrator fail to achieve optimal sample quality in the case of a larger number of sampling steps, because FD can effectively approximate derivatives when $h \to 0$ under the assumption of model continuity and differentiability. This is evidenced in Table 3.

Lastly, we analyze this simplified RD method purely from the perspective of derivative estimation. Compared to FD method, specifically, such RD method incorporates additional information

$$\frac{1 - \phi_1(m)}{\phi_1(m)} \frac{\Gamma(\tau_{t_{i-1}}) - \Gamma(\tau_{t_i})}{h_{t_i}} \tag{E.10}$$

from other higher-order derivative terms, which may counterbalance with these higher-order terms to a certain level. The consideration of this counterbalance is mainly based on two factors.

The first piece of evidence is its origin; clearly, as demonstrated in Eq. (3.7) and Figure 2, this additional information arises from recursive operations on other higher-order derivative terms. Therefore, this additional information encompasses partial information from approximations of higher-order derivatives, providing a more comprehensive consideration of the complexity involved in derivative estimation. We believe that this integrated source of information has the potential to offset, to some extent, the trends in higher-order terms in Eq. (E.7), making such RD method potentially more accurate and reliable in estimating derivatives for complex functions.

The second piece of evidence is grounded in the idea that, under certain assumptions, the RD method can achieve a truncation error of $O(h_{t_i}^2)$. For example, when considering $m = 3$, one can subtract Eq. (E.9) from Eq. (E.7), resulting in:

$$\Delta\left(\Gamma^{(1)}(\tau_{t_i})\right) = \frac{1}{2} \frac{\Gamma(\tau_{t_{i-1}}) - \Gamma(\tau_{t_i})}{h_{t_i}} + \frac{1}{2} h_{t_i} \Gamma^{(2)}(\tau_{t_i}) + \frac{1}{3!} h_{t_i}^2 \Gamma^{(3)}(\tau_{t_i}) + O(h_{t_i}^3). \tag{E.11}$$

By employing Eq. (E.1) in Eq. (E.11), we obtain:

$$\begin{aligned}
\Delta\left(\Gamma^{(1)}(\tau_{t_i})\right) &= \frac{1}{2}\Gamma^{(1)}(\tau_{t_i}) + \left(\frac{1}{2} + 1\right)\frac{1}{2}h_{t_i}\Gamma^{(2)}(\tau_{t_i}) + \left(\frac{1}{2} + 1\right)\frac{1}{3!}h_{t_i}^2\Gamma^{(3)}(\tau_{t_i}) + O(h_{t_i}^3) \\
&= \frac{3}{4}\left(\frac{2}{3}\Gamma^{(1)}(\tau_{t_i}) + h_{t_i}\Gamma^{(2)}(\tau_{t_i})\right) + \left(\frac{1}{2} + 1\right)\frac{1}{3!}h_{t_i}^2\Gamma^{(3)}(\tau_{t_i}) + O(h_{t_i}^3).
\end{aligned} \tag{E.12}$$

According to Lagrange mean value theorem, there exist $\tau_1$ and $\tau_2$ satisfying $\tau_{t_i} \in (\tau_1, \tau_2)$ and $h_{t_i} = \tau_2 - \tau_1$, such that

$$\Gamma^{(2)}(\tau_{t_i})(\tau_2 - \tau_1) = \Gamma^{(1)}(\tau_2) - \Gamma^{(1)}(\tau_1).$$

The most direct assumption is that if $\Gamma^{(1)}(\tau_1) = \Gamma^{(2)}(\tau_2) + \frac{2}{3}\Gamma^{(1)}(\tau_{t_i})$, then

$$\Delta\left(\Gamma^{(1)}(\tau_{t_i})\right) = O(h_{t_i}^2).$$

Certainly, the condition can be further relaxed to if $\Gamma^{(1)}(\tau_1) = \Gamma^{(2)}(\tau_2) + \frac{2}{3}\Gamma^{(1)}(\tau_{t_i}) + O(h_{t_i}^2)$, then

$$\Delta\left(\Gamma^{(1)}(\tau_{t_i})\right) = O(h_{t_i}^2).$$

The above analysis of derivative estimation suggests from a simple perspective that, for certain functions, such RD method can achieve better truncation errors.

Our numerical experiments confirm the applicability of this recursive approach to accelerating diffusion sampling. Additionally, it is worth noting that the effectiveness of this recursive method may be associated with specific properties of the diffusion model. We leave the exploration of these specific attributes for future research.

# F  PROOF OF THEOREM 3.2

Under reasonable assumptions, SciRE-V1-$k$ is a $k$-th order solver.

## F.1  PRELIMINARIES

**Assumption F.1** *The function $\epsilon_\theta(\mathbf{x}_t, t)$ is set to be in $\mathbb{C}^m$, meaning that $\epsilon_\theta(\mathbf{x}_t, t)$ is $m$ times continuously differentiable. Specifically, $m \geq 3$ in this paper.*

**Assumption F.2** *$\epsilon_\theta(\mathbf{x}_t, t)$ is a Lipschitz continuous function w.r.t. $\mathbf{x}_t$.*

**Assumption F.3** *For $\forall t$, there exists $\delta > 0$ such that $\exists \rho \in \mathbb{U}(t, \delta), (l + \frac{1}{2^n})\epsilon_\theta^{(1)}(\mathbf{x}_t, t) = l(1 + \frac{1}{2^n l})\epsilon_\theta^{(1)}(\mathbf{x}_t, t) = l\epsilon_\theta^{(1)}(\mathbf{x}_\rho, \rho)$ if $n \in \mathbb{N}$ is large enough. Here $\mathbb{U}$ denotes the neighbourhood of $t$.*

As in Appendix D, we denote $\tau := \mathrm{NSR}(t)$, $\psi(\tau) := \mathrm{rNSR}(\tau)$ and $h_s := \tau_t - \tau_s$. In Appendix B, we transform the form of the solution to diffusion ODEs as in Eq. (B.3) to the form as in (B.4) by using the change of variable formula. The resulting solution can be formed as:

$$\mathbf{x}_t = \frac{\alpha_t}{\alpha_s}\mathbf{x}_s + \alpha_t \int_{\tau_s}^{\tau_t} \epsilon_\theta\left(\mathbf{x}_{\psi(\tau)}, \psi(\tau)\right) d\tau. \tag{F.1}$$

Then by substituting the $n$-th order Taylor expansion of $\epsilon_\theta\left(\mathbf{x}_{\psi(\tau_t)}, \psi(\tau_t)\right)$ w.r.t. $\tau$ at $\tau_s$

$$\epsilon_\theta\left(\mathbf{x}_{\psi(\tau_t)}, \psi(\tau_t)\right) = \sum_{k=0}^{n} \frac{h_s^k}{k!}\epsilon_\theta^{(k)}\left(\mathbf{x}_{\psi(\tau_s)}, \psi(\tau_s)\right) + O(h_s^{n+1}), \tag{F.2}$$

for the $\epsilon_\theta\left(\mathbf{x}_{\psi(\tau_t)}, \psi(\tau_t)\right)$ in Eq. (F.1), we can derive the exact solution of $\mathbf{x}_t$ in Eq. (F.1) as follows:

$$
\begin{aligned}
\mathbf{x}_t &= \frac{\alpha_t}{\alpha_s}\mathbf{x}_s + \alpha_t \sum_{k=0}^{n} \frac{h_s^{k+1}}{(k+1)!}\epsilon_\theta^{(k)}\left(\mathbf{x}_{\psi(\tau_s)}, \psi(\tau_s)\right) + O(h_s^{n+2}) \\
&= \frac{\alpha_t}{\alpha_s}\mathbf{x}_s + \alpha_t \sum_{k=0}^{n} \frac{h_s^{k+1}}{(k+1)!}\epsilon_\theta^{(k)}(\mathbf{x}_s, s) + O(h_s^{n+2}).
\end{aligned}
\tag{F.3}
$$

## F.2  PROOF OF THEOREM 3.2 WHEN $k = 2$

In this subsection, we prove the global convergence order of SciRE-V1-2 is no less than 1 and is 2 under reasonable assumptions.

*Proof.* For each iteration step, we first update according to:

$$h_s = \tau_t - \tau_s, \tag{F.4}$$

$$s_1 = \psi(\tau_s + r_1 h_s), \tag{F.5}$$

$$\mathbf{u}_1 = \frac{\alpha_{s_1}}{\alpha_s}\mathbf{x}_s + \alpha_{s_1} r_1 h_s \epsilon_\theta(\mathbf{x}_s, s), \tag{F.6}$$

$$\tilde{\mathbf{x}}_t = \frac{\alpha_t}{\alpha_s}\mathbf{x}_s + \alpha_t h_s \epsilon_\theta(\mathbf{x}_s, s) + \alpha_t \frac{h_s^2}{2\phi_1(m)r_1 h_s}\left(\epsilon_\theta(\mathbf{u}_1, s_1) - \epsilon_\theta(\mathbf{x}_s, s)\right), \tag{F.7}$$

where $\tilde{\mathbf{x}}_t$ denotes the approximate solution of $\mathbf{x}_t$ computed by SciRE-V1-2, $r_1 \in (0, 1)$ is a hyperparameter so that $s_1 \in (t, s)$.

Next, taking $n = 1$ in Eq. (F.3), we can get the exact solution of $\mathbf{x}_t$ as follows:

$$\mathbf{x}_t = \frac{\alpha_t}{\alpha_s}\mathbf{x}_s + \alpha_t h_s \epsilon_\theta(\mathbf{x}_s, s) + \alpha_t \frac{h_s^2}{2}\epsilon_\theta^{(1)}(\mathbf{x}_s, s) + O(h_s^3) \tag{F.8}$$

Then by subtracting the last equation from Eq. (F.7) and using $\mathbf{u}_1 - \mathbf{x}_{s_1} = O(h_s^2)$, we have

$$
\begin{aligned}
\frac{\mathbf{x}_t - \tilde{\mathbf{x}}_t}{\alpha_t} &= \frac{h_s^2}{2}\epsilon_\theta^{(1)}(\mathbf{x}_s, s) - \frac{h_s^2}{2\phi_1(m)r_1 h_s}\left(\epsilon_\theta(\mathbf{u}_1, s_1) - \epsilon_\theta(\mathbf{x}_s, s)\right) + O(h_s^4) \\
&= \frac{h_s^2}{2}\left(\epsilon_\theta^{(1)}(\mathbf{x}_s, s) - \frac{\epsilon_\theta(\mathbf{x}_{s_1}, s_1) - \epsilon_\theta(\mathbf{x}_s, s)}{\phi_1(m)r_1 h_s}\right) + O(h_s^3),
\end{aligned}
\tag{F.9}
$$

where $\|\epsilon_\theta(\mathbf{u}_1, s_1) - \epsilon_\theta(\mathbf{x}_{s_1}, s_1)\| = O(\|\mathbf{u}_1 - \mathbf{x}_{s_1}\|) = O(h_s^2)$ under Assumption F.2.

By Assumption F.2 and Lagrange's mean value theorem, we find that

$$\|\epsilon_\theta(\mathbf{x}_{s_1}, s_1) - \epsilon_\theta(\mathbf{x}_s, s)\| \le L\|\mathbf{x}_{s_1} - \mathbf{x}_s\| = L\|\mathbf{x}'_\eta(s_1 - s)\| = L\|\mathbf{x}'_\eta \psi'(\tau_\xi)(\tau_{s_1} - \tau_s)\|, \tag{F.10}$$

where $\eta \in (\psi(\tau_{s_1}), \psi(\tau_s))$, $\tau_\xi \in (\tau_{s_1}, \tau_s)$ and $L$ is the Lipschitz constant. Since $\|\tau_{s_1} - \tau_s\| \le \|\tau_t - \tau_s\| = O(h_s)$, the r.h.s. of the above inequation is $O(h_s)$.

Besides, by Assumption F.1, we also have

$$\epsilon_\theta^{(1)}(\mathbf{x}_s, s) - \frac{\epsilon_\theta(\mathbf{x}_{s_1}, s_1) - \epsilon_\theta(\mathbf{x}_s, s)}{\phi_1(m)r_1h_s} = O(1) - \frac{O(h_s)}{h_s} = O(1). \tag{F.11}$$

Hence $\mathbf{x}_t - \tilde{\mathbf{x}}_t = O(h_s^2)$. We prove that SciRE-V1-2 is **at least** a first order solver.

Furthermore, we will prove that **SciRE-V1-2 is a second order solver under mild condition.**

Specifically, while $\phi_1(m) = \phi_1(3) = 2/3$, by the Lagrange's mean value theorem, there exists $\xi \in (s_1, s)$ such that

$$\frac{\epsilon_\theta(\mathbf{x}_{s_1}, s_1) - \epsilon_\theta(\mathbf{x}_\xi, \xi)}{\phi_1(m)r_1h_s} = \epsilon_\theta^{(1)}(\mathbf{x}_\xi, \xi) + O(h_s). \tag{F.12}$$

Note that $\phi_1(m)r_1h_s = \tau_{s1} - \tau_\xi$ and $(1 - \phi_1(m))r_1h_s = \tau_\xi - \tau_s$, hence

$$\begin{aligned}
\frac{\epsilon_\theta(\mathbf{x}_{s_1}, s_1) - \epsilon_\theta(\mathbf{x}_s, s)}{\phi_1(m)r_1h_s} &= \frac{\epsilon_\theta(\mathbf{x}_{s_1}, s_1) - \epsilon_\theta(\mathbf{x}_\xi, \xi)}{\phi_1(m)r_1h_s} + \frac{\epsilon_\theta(\mathbf{x}_\xi, \xi) - \epsilon_\theta(\mathbf{x}_s, s)}{\phi_1(m)r_1h_s} \\
&= \epsilon_\theta^{(1)}(\mathbf{x}_\xi, \xi) + O(h_s) + \frac{1}{2}\frac{\epsilon_\theta(\mathbf{x}_\xi, \xi) - \epsilon_\theta(\mathbf{x}_s, s)}{(1 - \phi_1(m))r_1h_s} \\
&= \epsilon_\theta^{(1)}(\mathbf{x}_\xi, \xi) + \frac{1}{2}\epsilon_\theta^{(1)}(\mathbf{x}_s, s) + O(h_s).
\end{aligned} \tag{F.13}$$

Combining the above equation with Eq. (F.9), we have

$$\begin{aligned}
\frac{\mathbf{x}_t - \tilde{\mathbf{x}}_t}{\alpha_t} &= \frac{h_s^2}{2}\epsilon_\theta^{(1)}(\mathbf{x}_s, s) - \frac{h_s^2}{2}\frac{\epsilon_\theta(\mathbf{x}_{s_1}, s_1) - \epsilon_\theta(\mathbf{x}_s, s)}{\phi_1(m)r_1h_s} + O(h_s^3) \\
&= \frac{h_s^2}{2}\epsilon_\theta^{(1)}(\mathbf{x}_s, s) - \frac{h_s^2}{2}\left(\epsilon_\theta^{(1)}(\mathbf{x}_\xi, \xi) + \frac{1}{2}\epsilon_\theta^{(1)}(\mathbf{x}_s, s)\right) + O(h_s^3) \\
&= \frac{h_s^2}{4}\left(\epsilon_\theta^{(1)}(\mathbf{x}_s, s) - 2\epsilon_\theta^{(1)}(\mathbf{x}_\xi, \xi)\right) + O(h_s^3),
\end{aligned} \tag{F.14}$$

where $\xi \in (s_1, s)$. By Assumption F.1, $\epsilon_\theta^{(2)}(\mathbf{x}_s, s)$ is bounded hence the first term in the r.h.s. of above equation is $O(h_s^3)$. While for the second term, we find that

$$\begin{aligned}
&\epsilon_\theta^{(1)}(\mathbf{x}_s, s) - 2\epsilon_\theta^{(1)}(\mathbf{x}_\xi, \xi) \\
&= \left[\epsilon_\theta^{(1)}(\mathbf{x}_s, s) - (1 + \frac{1}{2})\epsilon_\theta^{(1)}(\mathbf{x}_{s_1}, s_1)\right] + \left[(1 + \frac{1}{2})\epsilon_\theta^{(1)}(\mathbf{x}_{s_1}, s_1) - 2\epsilon_\theta^{(1)}(\mathbf{x}_\xi, \xi)\right] \\
&= \left[\epsilon_\theta^{(1)}(\mathbf{x}_s, s) - (1 + \frac{1}{4})\epsilon_\theta^{(1)}(\mathbf{x}_\xi, \xi)\right] + \left[(1 + \frac{1}{4})\epsilon_\theta^{(1)}(\mathbf{x}_\xi, \xi) - (1 + \frac{1}{2})\epsilon_\theta^{(1)}(\mathbf{x}_{s_1}, s_1)\right] \\
&\quad + \left[(1 + \frac{1}{2})\epsilon_\theta^{(1)}(\mathbf{x}_{s_1}, s_1) - (1 + \frac{1}{2} + \frac{1}{4})\epsilon_\theta^{(1)}(\mathbf{x}_s, s)\right] + \left[(1 + \frac{1}{2} + \frac{1}{4})\epsilon_\theta^{(1)}(\mathbf{x}_s, s) - 2\epsilon_\theta^{(1)}(\mathbf{x}_\xi, \xi)\right] \\
&\quad \cdots
\end{aligned} \tag{F.15}$$

We now define *group*, for example, $\epsilon_\theta^{(1)}(\mathbf{x}_s, s) - (1 + \frac{1}{2})\epsilon_\theta^{(1)}(\mathbf{x}_{s_1}, s_1)$, which is grouped by "[]". We also find that for each group, if the coefficient of the first term is $l$, then the coefficient of the second term is $l + \frac{1}{2^n}$ after using dichotomy for $n$ times. Note that $l \in [1, 2)$ such that $l + \frac{1}{2^n} \in (1, 2], \forall n \in \mathbb{N}$. Besides, $t$ takes two different values in $\{s, s_1, \xi\}$. By Eq. (F.10) and Assumption F.3, if $n$ is large enough, each group is $O(h_s)$, for example,

$$l\epsilon_\theta^{(1)}(\mathbf{x}_{s_1}, s_1) - \left(l + \frac{1}{2^n}\right)\epsilon_\theta^{(1)}(\mathbf{x}_s, s) = l\epsilon_\theta^{(1)}(\mathbf{x}_{s_1}, s_1) - l\epsilon_\theta^{(1)}(\mathbf{x}_\rho, \rho) = O(h_s). \tag{F.16}$$

Hence $\epsilon_\theta^{(1)}(\mathbf{x}_s, s) - 2\epsilon_\theta^{(1)}(\mathbf{x}_\xi, \xi)$ is $O(2^n h_s)$. In practice, $n$ is finite, meaning that $2^n$ is bounded and $O(2^n h_s) = O(h_s)$. Subquently, the proof is completed by

$$\frac{\mathbf{x}_t - \tilde{\mathbf{x}}_t}{\alpha_t} = O(h_s^3) + \frac{h_s^2}{4}O(h_s) + O(h_s^3) = O(h_s^3). \tag{F.17}$$

$\square$

### F.3 Proof of Theorem 3.2 when $k = 3$

In this subsection, we prove the global convergence order of SciRE-V1-3 is no less than 2.

*Proof.* For each iteration step, we first update according to:

$$h_s = \tau_t - \tau_s, \tag{F.18}$$

$$s_1 = \psi(\tau_s + r_1 h_s), \tag{F.19}$$

$$s_2 = \psi(\tau_s + r_2 h_s), \tag{F.20}$$

$$\mathbf{u}_1 = \frac{\alpha_{s_1}}{\alpha_s}\mathbf{x}_s + \alpha_{s_1} r_1 h_s \epsilon_\theta(\mathbf{x}_s, s), \tag{F.21}$$

$$\mathbf{u}_2 = \frac{\alpha_{s_2}}{\alpha_s}\mathbf{x}_s + \alpha_{s_2} r_2 h_s \epsilon_\theta(\mathbf{x}_s, s) + \alpha_{s_2}\frac{h_s}{\phi_1(m)}(\epsilon_\theta(\mathbf{u}_1, s_1) - \epsilon_\theta(\mathbf{x}_s, s)), \tag{F.22}$$

$$\tilde{\mathbf{x}}_t = \frac{\alpha_t}{\alpha_s}\mathbf{x}_s + \alpha_t h_s \epsilon_\theta(\mathbf{x}_s, s) + \alpha_t\frac{h_s^2}{2\phi_1(m)r_2 h_s}(\epsilon_\theta(\mathbf{u}_2, s_2) - \epsilon_\theta(\mathbf{x}_s, s)), \tag{F.23}$$

where $\tilde{\mathbf{x}}_t$ denotes the approximate solution of $\mathbf{x}_t$ computed by SciRE-V1-3, $r_1 \in (0, 1)$ and $r_2 = 1 - r_1$ are hyperparameters so that $s_1, s_2 \in (t, s)$.

We firstly prove that $\mathbf{u}_2 - \mathbf{x}_{s_2} = O(h_s^3)$:

$$\mathbf{u}_2 = \frac{\alpha_{s_2}}{\alpha_s}\mathbf{x}_s + \alpha_{s_2} r_2 h_s \epsilon_\theta(\mathbf{x}_s, s) + \alpha_{s_2}\frac{h_s}{\phi_1(m)}(\epsilon_\theta(\mathbf{u}_1, s_1) - \epsilon_\theta(\mathbf{x}_s, s))$$

$$= \underbrace{\frac{\alpha_{s_2}}{\alpha_s}\mathbf{x}_s + \alpha_{s_2} r_2 h_s \epsilon_\theta(\mathbf{x}_s, s) + \alpha_{s_2}\frac{h_s}{\phi_1(m)}(\epsilon_\theta(\mathbf{x}_{s_1}, s_1) - \epsilon_\theta(\mathbf{x}_s, s))}_{\mathbf{x}_{s_2}} + O(h_s^3), \tag{F.24}$$

by $\mathbf{u}_1 - \mathbf{x}_{s_1} = O(h_s^2)$ and Assumption F.2.

Next, taking $n = 2$ in Eq. (F.3), we can get the exact solution of $\mathbf{x}_t$ as follows:

$$\mathbf{x}_t = \frac{\alpha_t}{\alpha_s}\mathbf{x}_s + \alpha_t h_s \epsilon_\theta(\mathbf{x}_s, s) + \alpha_t \sum_{k=1}^2 \frac{h_s^{k+1}}{(k+1)!}\epsilon_\theta^{(k)}(\mathbf{x}_s, s) + O(h_s^4) \tag{F.25}$$

Then by subtracting the last equation from Eq. (F.23) and using $\mathbf{u}_1 - \mathbf{x}_{s_1} = O(h_s^2)$, we have

$$\frac{\mathbf{x}_t - \tilde{\mathbf{x}}_t}{\alpha_t} = \sum_{k=1}^2 \frac{h_s^{k+1}}{(k+1)!}\epsilon_\theta^{(k)}(\mathbf{x}_s, s) - \frac{h_s^2}{2\phi_1(m)r_2 h_s}(\epsilon_\theta(\mathbf{u}_2, s_2) - \epsilon_\theta(\mathbf{x}_s, s)) + O(h_s^4)$$

$$= \frac{h_s^2}{2}\epsilon_\theta^{(1)}(\mathbf{x}_s, s) + \frac{h_s^3}{3!}\epsilon_\theta^{(2)}(\mathbf{x}_s, s) - \frac{h_s^2}{2}\frac{\epsilon_\theta(\mathbf{x}_{s_2}, s_2) - \epsilon_\theta(\mathbf{x}_s, s)}{\phi_1(m)r_2 h_s} + O(h_s^4), \tag{F.26}$$

where $\|\epsilon_\theta(\mathbf{u}_2, s_2) - \epsilon_\theta(\mathbf{x}_{s_2}, s_2)\| = O(\|\mathbf{u}_2 - \mathbf{x}_{s_2}\|) = O(h_s^3)$. Similar to the proof in F.2, we have

$$\frac{\mathbf{x}_t - \tilde{\mathbf{x}}_t}{\alpha_t} = \sum_{k=1}^2 \frac{h_s^{k+1}}{(k+1)!}\epsilon_\theta^{(k)}(\mathbf{x}_s, s) - \frac{h_s^2}{2}\frac{\epsilon_\theta(\mathbf{x}_{s_2}, s_2) - \epsilon_\theta(\mathbf{x}_s, s)}{\phi_1(m)r_2 h_s} + O(h_s^4)$$

$$= \frac{h_s^3}{3!}\epsilon_\theta^{(2)}(\mathbf{x}_s, s) + \frac{h_s^2}{2}\epsilon_\theta^{(1)}(\mathbf{x}_s, s) - \frac{h_s^2}{2}\left(\epsilon_\theta^{(1)}(\mathbf{x}_\xi, \xi) + \frac{1}{2}\epsilon_\theta^{(1)}(\mathbf{x}_s, s)\right) + O(h_s^3) \tag{F.27}$$

$$= \frac{h_s^3}{3!}\epsilon_\theta^{(2)}(\mathbf{x}_s, s) + \frac{h_s^2}{4}\left(\epsilon_\theta^{(1)}(\mathbf{x}_s, s) - 2\epsilon_\theta^{(1)}(\mathbf{x}_\xi, \xi)\right) + O(h_s^3)$$

$$= O(h_s^3) + 2^n O(h_s^3) + O(h_s^3) = O(h_s^3),$$

where $\xi \in (s_2, s)$ and $n \in \mathbb{N}$ is finite. Hence we prove that the global convergence order of SciRE-V1-3 is no less than 2. $\square$

# G ALGORITHMS OF SCIRE-SOLVER

In Section 3.2, we propose the *recursive difference* (RD) method for approximating derivatives, the conclusion as shown in Theorem 3.1 and Corollary 3.1. In this section, we discuss in detail the SciRE-Solver based on RD method.

## G.1 SCIRE-V1 WITH BACKWARD DIFFERENCE

Here, we introduce a simple variant based on SciRE-V1-2, which utilizes RD method and backward difference estimation for the estimation of derivative of the score function. The specific details are as described in Algorithm 4.

---

**Algorithm 4** SciRE-V1-2m

---

**Require:** initial value $\mathbf{x}_T$, time trajectory $\{t_i\}_{i=0}^N$, model $\epsilon_\theta, m \geq 3$, empty cache $R$.
1: Denote $h_{i-1} := \mathrm{NSR}(t_{i-1}) - \mathrm{NSR}(t_i)$, where $i = 1, \ldots, N$.
2: $\tilde{\mathbf{x}}_{t_N} \leftarrow \mathbf{x}_T, R \overset{cache}{\leftarrow} \epsilon_\theta\left(\tilde{\mathbf{x}}_{t_N}, t_N\right)$
3: $\tilde{\mathbf{x}}_{t_{N-1}} \leftarrow \frac{\alpha_{t_{N-1}}}{\alpha_{t_N}}\tilde{\mathbf{x}}_{t_N} + \alpha_{t_{N-1}}h_N\epsilon_\theta\left(\tilde{\mathbf{x}}_{t_N}, t_N\right)$
4: $R \overset{cache}{\leftarrow} \epsilon_\theta\left(\tilde{\mathbf{x}}_{t_{N-1}}, t_{N-1}\right)$
5: **for** $i \leftarrow N - 2$ to 1 **do**
6:   $r_i \leftarrow \phi_1(m)\frac{h_i}{h_{i-1}}, D_{i+1} \leftarrow \epsilon_\theta(\tilde{\mathbf{x}}_{t_{i+1}}, t_{i+1}) - \epsilon_\theta(\tilde{\mathbf{x}}_{t_{i+2}}, t_{i+2})$
7:   $\tilde{\mathbf{x}}_{t_i} \leftarrow \frac{\alpha_{t_i}}{\alpha_{t_{i+1}}}\tilde{\mathbf{x}}_{t_{i+1}} + \alpha_{t_i}h_i\epsilon_\theta\left(\tilde{\mathbf{x}}_{t_{i+1}}, t_{i+1}\right) + \alpha_{t_i}\frac{h_i}{2\phi_1(m)r_i}D_{i+1}$
8:   $R \overset{cache}{\leftarrow} \epsilon_\theta\left(\tilde{\mathbf{x}}_{t_i}, t_i\right)$
9: **end for**
**Return:** $\tilde{\mathbf{x}}_0$.

---

## G.2 ALGORITHMS OF SCIRE-V2

In this section, we introduce *SciRE-V2*, which is based on ODE Eq. (3.3) and the RD method. The derivation principle and convergence order of the SciRE-V2 algorithms remain consistent with SciRE-V1, while the specific iteration details are shown in Algorithms 5, 6, and 7.

---

**Algorithm 5** SciRE-V2-2

---

**Require:** initial value $\mathbf{x}_T$, time trajectory $\{t_i\}_{i=0}^N$, model $\boldsymbol{x}_\theta, m \geq 3$.
1: $\tilde{\mathbf{x}}_{t_N} \leftarrow \mathbf{x}_T, r_1 \leftarrow \frac{1}{2}$
2: **for** $i \leftarrow N$ to 1 **do**
3:   $h_i \leftarrow 1/\mathrm{NSR}(t_{i-1}) - 1/\mathrm{NSR}(t_i)$
4:   $s_i \leftarrow \mathrm{rNSR}\left(1/(1/\mathrm{NSR}(t_i) + r_1h_i)\right)$
5:   $\tilde{\mathbf{x}}_{s_i} \leftarrow \frac{\sigma_{s_i}}{\sigma_{t_i}}\tilde{\mathbf{x}}_{t_i} + \sigma_{s_i}r_1h_i\boldsymbol{x}_\theta\left(\tilde{\mathbf{x}}_{t_i}, t_i\right)$
6:   $\tilde{\mathbf{x}}_{t_{i-1}} \leftarrow \frac{\sigma_{t_{i-1}}}{\sigma_{t_i}}\tilde{\mathbf{x}}_{t_i} + \sigma_{t_{i-1}}h_i\boldsymbol{x}_\theta\left(\tilde{\mathbf{x}}_{t_i}, t_i\right) + \sigma_{t_{i-1}}\frac{h_i}{2\phi_1(m)r_1}\left(\boldsymbol{x}_\theta\left(\tilde{\mathbf{x}}_{s_i}, s_i\right) - \boldsymbol{x}_\theta\left(\tilde{\mathbf{x}}_{t_i}, t_i\right)\right)$
7: **end for**
**Return:** $\tilde{\mathbf{x}}_0$.

---

## G.3 ANALYTICAL FORMULATION OF THE FUNCTION rNSR($\cdot$)

The computational costs associated with computing rNSR($\cdot$) are negligible. This is due to the fact that for the noise schedules of $\alpha_t$ and $\sigma_t$ employed in previous DMs (referred to as "linear" and "cosine" in (Ho et al., 2020; Nichol & Dhariwal, 2021)), both rNSR($\cdot$) and its inverse function NSR($t$) have analytic formulations. We mainly consider the variance preserving type here, since it is the most widely-used type. The functions for other types (variance exploding and sub-variance preserving type) can be similarly derived.

**Linear Noise Schedule (Ho et al., 2020).** In fact,

$$\alpha_t = \exp\left(-\frac{(\beta_1 - \beta_0)}{4}t^2 - \frac{\beta_0}{2}t\right),$$

where $\beta_0 = 0.1$ and $\beta_1 = 20$, following (Song et al., 2021c; Lu et al., 2022b). As $\sigma_t = \sqrt{1 - \alpha_t^2}$, we can compute NSR(t) analytically. Specifically, the inverse function is

$$t = \text{rNSR}(\tau) = \frac{1}{\beta_1 - \beta_0}\left(\sqrt{\beta_0^2 + 2(\beta_1 - \beta_0)\log(1 + \tau^2)} - \beta_0\right),$$

where $\tau = \text{NSR}(t)$. In order to mitigate the influence of numerical issues, we have the option of calculating the value of $t$ using the alternative expression provided below:

$$t = \text{rNSR}(\tau) = \frac{2\log\left(1 + \tau^2\right)}{\sqrt{\beta_0^2 + 2(\beta_1 - \beta_0)\log(1 + \tau^2)} + \beta_0}.$$

---

**Algorithm 6** SciRE-V2-3

---

**Require:** initial value $\mathbf{x}_T$, time trajectory $\{t_i\}_{i=0}^N$, model $\boldsymbol{x}_\theta, m \geq 3$.
1: $\tilde{\mathbf{x}}_{t_N} \leftarrow \mathbf{x}_T, r_1 \leftarrow \frac{1}{3}, r_2 \leftarrow \frac{2}{3}$
2: **for** $i \leftarrow N$ to $1$ **do**
3: $\quad h_i \leftarrow 1/\text{NSR}(t_{i-1}) - 1/\text{NSR}(t_i)$
4: $\quad s_{i_1}, s_{i_2} \leftarrow \text{rNSR}\left(1/(1/\text{NSR}(t_i) + r_1 h_i)\right), \text{rNSR}\left(1/(1/\text{NSR}(t_i) + r_2 h_i)\right)$
5: $\quad \tilde{\mathbf{x}}_{s_{i_1}} \leftarrow \frac{\sigma_{s_{i_1}}}{\sigma_{t_i}}\tilde{\mathbf{x}}_{t_i} + \sigma_{s_{i_1}} r_1 h_i \boldsymbol{x}_\theta\left(\tilde{\mathbf{x}}_{t_i}, t_i\right)$
6: $\quad \tilde{\mathbf{x}}_{s_{i_2}} \leftarrow \frac{\sigma_{s_{i_2}}}{\sigma_{t_i}}\tilde{\mathbf{x}}_{t_i} + \sigma_{s_{i_2}} r_2 h_i \boldsymbol{x}_\theta\left(\tilde{\mathbf{x}}_{t_i}, t_i\right) + \sigma_{s_{i_2}}\frac{r_2 h_i}{2\phi_1(m)r_1}\left(\boldsymbol{x}_\theta\left(\tilde{\mathbf{x}}_{s_{i_1}}, s_{i_1}\right) - \boldsymbol{x}_\theta\left(\tilde{\mathbf{x}}_{t_i}, t_i\right)\right)$
7: $\quad \tilde{\mathbf{x}}_{t_{i-1}} \leftarrow \frac{\sigma_{t_{i-1}}}{\sigma_{t_i}}\tilde{\mathbf{x}}_{t_i} + \sigma_{t_{i-1}} h_i \boldsymbol{x}_\theta\left(\tilde{\mathbf{x}}_{t_i}, t_i\right) + \sigma_{t_{i-1}}\frac{h_i}{2\phi_1(m)r_2}\left(\boldsymbol{x}_\theta\left(\tilde{\mathbf{x}}_{s_{i_2}}, s_{i_2}\right) - \boldsymbol{x}_\theta\left(\tilde{\mathbf{x}}_{t_i}, t_i\right)\right)$
8: **end for**
**Return:** $\tilde{\mathbf{x}}_0$.

---

**Algorithm 7** SciRE-V2-2m

---

**Require:** initial value $\mathbf{x}_T$, time trajectory $\{t_i\}_{i=0}^N$, model $\boldsymbol{x}_\theta, m \geq 3$, empty cache $R$.
1: Denote $h_{i-1} := 1/\text{NSR}(t_{i-1}) - 1/\text{NSR}(t_i)$, $\hat{h}_{i-1} := \text{NSR}(t_{i-1}) - \text{NSR}(t_i)$, where $i = 1, \dots, N$.
2: $\tilde{\mathbf{x}}_{t_N} \leftarrow \mathbf{x}_T, R \overset{cache}{\leftarrow} \boldsymbol{x}_\theta\left(\tilde{\mathbf{x}}_{t_N}, t_N\right)$
3: $\tilde{\mathbf{x}}_{t_{N-1}} \leftarrow \frac{\sigma_{t_{N-1}}}{\sigma_{t_N}}\tilde{\mathbf{x}}_{t_N} + \sigma_{t_{N-1}} h_N \boldsymbol{x}_\theta\left(\tilde{\mathbf{x}}_{t_N}, t_N\right)$
4: $R \overset{cache}{\leftarrow} \boldsymbol{x}_\theta\left(\tilde{\mathbf{x}}_{t_{N-1}}, t_{N-1}\right)$
5: **for** $i \leftarrow N - 2$ to $1$ **do**
6: $\quad r_i \leftarrow \phi_1(m)\frac{h_i}{\hat{h}_{i-1}}, D_{i+1} \leftarrow \boldsymbol{x}_\theta(\tilde{\mathbf{x}}_{t_{i+1}}, t_{i+1}) - \boldsymbol{x}_\theta(\tilde{\mathbf{x}}_{t_{i+2}}, t_{i+2})$
7: $\quad \tilde{\mathbf{x}}_{t_i} \leftarrow \frac{\sigma_{t_i}}{\sigma_{t_{i+1}}}\tilde{\mathbf{x}}_{t_{i+1}} + \sigma_{t_i} h_i \boldsymbol{x}_\theta\left(\tilde{\mathbf{x}}_{t_{i+1}}, t_{i+1}\right) + \sigma_{t_i}\frac{h_i}{2\phi_1(m)r_i} D_{i+1}$
8: $\quad R \overset{cache}{\leftarrow} \boldsymbol{x}_\theta\left(\tilde{\mathbf{x}}_{t_i}, t_i\right)$
9: **end for**
**Return:** $\tilde{\mathbf{x}}_0$.

---

Moreover, we solve diffusion ODEs within the interval $[\epsilon, T]$, where $T$ is set to 1.

**Cosine Noise Schedule (Nichol & Dhariwal, 2021).** Denote

$$\alpha_t = \cos\left(\frac{\pi}{2} \cdot \frac{t + s}{1 + s}\middle/\frac{s}{1 + s}\right),$$

where $s = 0.008$, following (Nichol & Dhariwal, 2021). As $\sigma_t = \sqrt{1 - \alpha_t^2}$, we can compute NSR(t) analytically. Denote $\tau = \text{NSR}(t)$, and define the function $\varphi(\tau)$ as follows:

$$\varphi(\tau) = -\frac{1}{2}\log\left(1 + \tau^2\right).$$

Then the inverse function is

$$t = \text{rNSR}(\tau) = \frac{2(1+s)}{\pi} \arccos\left(e^{\varphi(\tau) + \log\cos\left(\frac{\pi s}{2(1+s)}\right)}\right) - s.$$

Moreover, we solve diffusion ODEs within the interval $[\epsilon, T]$, where $T$ is set to 0.9946, following (Lu et al., 2022b).

### G.4 SciRE-V1-agile

In order to facilitate the exploration of more possibilities of the SciRE-V1 our proposed and to fully utilize the given number of score function evaluations (NFE), we defined a simple combinatorial version based on SciRE-V1-$k$ and named as SciRE-V1-agile. This version is based on whether the given NFE is divisible by $k$. If it is not divisible, solver-$k$ is used as much as possible first, and then smaller order SciRE-V1 or DDIM are used to supplement.

To achieve this, when given a fixed budget $N$ for the number of score function evaluations, we evenly divide the given interval into $M = (\lfloor N/3 \rfloor + 1)$ segments. Subsequently, we carry out $M$ sampling steps, adjusting based on the remainder $R$ when dividing $N$ by 3 to ensure a precise total of $N$ evaluations.

When $R = 0$, we initiate $M - 2$ SciRE-V1-3 steps, succeeded by 1 SciRE-V1-2 step and 1 DDIM step. This results in a total of $3 \cdot \left(\frac{N}{3} - 1\right) + 2 + 1 = N$ evaluations.

In the case of $R = 1$, we begin with $M - 1$ SciRE–V1-3 steps, followed by 1 DDIM step. This yields a total of $3 \cdot \left(\frac{N-1}{3}\right) + 1 = N$ evaluations.

Lastly, when $R = 2$, we conduct $M - 1$ SciRE–V1-3 steps, succeeded by 1 SciRE–V1-2 step. This leads to a cumulative count of $3 \cdot \left(\frac{N-2}{3}\right) + 2 = N$ score function evaluations.

Our empirical observations show that using this time step design can enhance the quality of image generation. With the implementation of the SciRE–V1 algorithm, high-quality samples can be generated in just 20 steps, such as achieving a 2.42 FID result on CIFAR-10 with just 20 NFE.

### G.5 Sampling from Discrete-Time DMs

SciRE-V1 aims to solve continuous-time diffusion ODEs. For DMs trained on discrete-time labels, we need to firstly wrap the model function to a noise prediction model that accepts the continuous time as the input. In the subsequent discussion, we examine the broader scenario of discrete-time DMs, specifically focusing on two variants: the 1000-step DMs (Ho et al., 2020) and the 4000-step DMs (Nichol & Dhariwal, 2021). Discrete-time DMs (Ho et al., 2020) train the noise prediction model at $N$ fixed time steps $\{t_n\}_{n=1}^{N}$, and the value of $N$ is typically set to either 1000 or 4000 in practice. The implementation of the 4000-step DMs (Nichol & Dhariwal, 2021) entails mapping the time steps of the 4000-step DMs to the range of the 1000-step DMs. Specifically, the noise prediction model is parameterized as $\tilde{\epsilon}_\theta\left(\mathbf{x}_n, \frac{1000n}{N}\right)$, where $\mathbf{x}_n$ is corresponding to the value at time $t_{n+1}$, and $n$ ranges from 0 to $N - 1$. In practice, these discrete-time DMs commonly employ uniform time steps between $[0, T]$, then $t_n = \frac{nT}{N}$, for $n = 1, \ldots, N$.

As sated by Lu et al. (2022b), the discrete-time noise prediction model is limited in predicting noise levels for times less than the smallest time $t_1$. Given that $t_1 = \frac{T}{N}$ and the corresponding discrete-time noise prediction model at time $t_1$ is $\tilde{\epsilon}_\theta(\mathbf{x}_0, 0)$, it is necessary to "scale" the discrete time steps from $[t_1, t_N] = \left[\frac{T}{N}, T\right]$ to the continuous time range $[\epsilon, T]$. However, the question of which scaling approach would be beneficial to the corresponding sampling algorithm remains an open problem.

In our codebase, we employ two types of scaling recommended by Lu et al. (2022b) as follows.

**Discrete-1.** Let $\epsilon_\theta(\cdot, t) = \epsilon_\theta\left(\cdot, \frac{T}{N}\right)$ for $t \in \left[\epsilon, \frac{T}{N}\right]$, and scale the discrete time steps $[t_1, t_N] = \left[\frac{T}{N}, T\right]$ to the continuous time range $\left[\frac{T}{N}, T\right]$. Then, the continuous-time noise prediction model is defined by

$$\epsilon_\theta(\mathbf{x}, t) = \tilde{\epsilon}_\theta\left(\mathbf{x}, 1000 \cdot \max\left(t - \frac{T}{N}, 0\right)\right),$$

where the continuous time $t \in \left[\epsilon, \frac{T}{N}\right]$ maps to the discrete input $0$, and the continuous time $T$ maps to the discrete input $\frac{1000(N-1)}{N}$.

**Discrete-2.** Scale the discrete time steps $[t_1, t_N] = \left[\frac{T}{N}, T\right]$ to the continuous time range $[0, T]$. In this case, the continuous-time noise prediction model is defined by

$$\epsilon_\theta(\mathbf{x}, t) = \tilde{\epsilon}_\theta\left(\mathbf{x}, 1000 \cdot \frac{(N-1)t}{NT}\right),$$

where the continuous time $0$ maps to the discrete input $0$, and the continuous time $T$ maps to the discrete input $\frac{1000(N-1)}{N}$.

By such reparameterization, the noise prediction model can adopt the continuous-time steps as input, which enables SciRE-V1 to perform sampling not only for continuous-time DMs but also for discrete-time DMs.

### G.6 CONDITIONAL SAMPLING BY SCIRE-SOLVER

With a simple modification, following the settings provided by Lu et al. (2022b), SciRE-Solver can be used for conditional sampling. The conditional generation requires sampling from a conditional diffusion ODE, as stated in (Song et al., 2021c; Dhariwal & Nichol, 2021). Specifically, by following the classifier guidance method (Dhariwal & Nichol, 2021), the conditional noise prediction model can defied as $\epsilon_\theta(\mathbf{x}_t, t, y) := \epsilon_\theta(\mathbf{x}_t, t) - s \cdot \sigma_t \nabla_{\mathbf{x}} \log p_t(y \mid \mathbf{x}_t; \theta)$. Here, $p_t(y \mid \mathbf{x}_t; \theta)$ represents a pre-trained classifier, and $s$ denotes the classifier guidance scale. Thus, one can utilize SciRE-Solver to solve this diffusion ODE for fast conditional sampling.

### G.7 SUPPORTED MODELS

SciRE-Solver support four types of diffusion probabilistic models, including the noise prediction model $\epsilon_\theta$ (Ho et al., 2020; Rombach et al., 2021), the data prediction model $x_\theta$ (Ramesh et al., 2022), the velocity prediction model $\mathbf{v}_\theta$ (Ho et al., 2022) and marginal score function $\mathbf{s}_\theta$ (Song et al., 2021c). Here, we follow the configurations provided by Lu et al. (2022b;c).

## H EXPERIMENT DETAILS

In this section, we provide more details on SciRE-V1 and further demonstrate the performance of SciRE-V1 on both discrete-time DMs and continuous-time DMs. Specifically, we consider the 1000-step DMs (Ho et al., 2020) and the 4000-step DMs (Nichol & Dhariwal, 2021), and consider the end time $\epsilon$ and time trajectory for sampling. We test our method for sampling the most widely-used variance-preserving (VP) type DMs (Sohl-Dickstein et al., 2015; Song et al., 2021c). In this case, we have $\alpha_t^2 + \sigma_t^2 = 1$ for all $t \in [0, T]$. In spite of this, our method and theoretical results are general and independent of the choice of the noise schedule $\alpha_t$ and $\sigma_t$. In all experiments, the number of NFE represents the sampling steps. For early experiments, we evaluate SciRE-V1 on NVIDIA TITAN X GPUs. For each experiments, we draw 50K samples and assess sample quality using the widely adopted FID score (Heusel et al., 2017), where lower FID generally indicate better sample quality. In order to facilitate the exploration of more possibilities of the SciRE-V1 our proposed and to fully utilize the given number of score function evaluations (NFE), we defined a simple combinatorial version based on SciRE-V1-$k$ and named as SciRE–V1-agile, as detailed in Appendix H.

### H.1 END TIME OF SAMPLING

Theoretically, we need to solve diffusion ODEs from time $T$ to time $0$ to generate samples. Practically, the training and evaluation for the noise prediction model $\epsilon_\theta(\mathbf{x}_t, t)$ usually start from time $T$ to time $\epsilon$ to avoid numerical issues for t near to $0$, where $\epsilon \geq 0$ is a hyperparameter (Song et al., 2021c). In contrast to the sampling methods based on diffusion SDEs (Ho et al., 2020; Song et al., 2021c), we, like DPM-Solver (Lu et al., 2022b), do not incorporate the "denoising" trick (i.e., setting the noise variance to zero) in the final step at time $\epsilon$. Instead, we solely solve diffusion ODEs from T to $\epsilon$ using the SciRE-V1.

### H.2 TIME TRAJECTORIES

In SciRE-V1, it is necessary to specify a time trajectory. Although SciRE-V1 can generate high-quality samples in a few steps using existing quadratic and uniform time trajectories, it has been demonstrated in experiments in (Lu et al., 2022b) and (Zhang & Chen, 2023) that the optimal time trajectory can further improve the sampling efficiency. Here, we present two parametrizable alternative methods for the NSR function to compute the time trajectory, named as *NSR*-type and *Sigmoid*-type time trajectories. Let $\{t_i\}_{i=0}^N$ be the time trajectory of diffusion probabilistic models, where $t_N = T$ and $t_0 = \epsilon \geq 0$. In the context of fast sampling, it is always desirable for the number $N$ of time points in the time trajectory to be as small as possible. However, the selection of the optimal time trajectory remains an open problem for the few-step sampling regime of diffusion probabilistic models. In this work, we hypothesize that selecting a time trajectory with sparser time points in the middle and relatively denser time points at the two ends would be beneficial for improving the quality of sample generation. To validate this hypothesis, inspired by the logarithmic and sigmoid functions, we propose two parametrizable alternative methods for the NSR function to compute the time trajectory, named as *NSR*-type and *Sigmoid*-type time trajectories, respectively.

**NSR-type:** For a given starting time $t_T$ and ending time $t_0$ of the sampling, the time values at the intermediate endpoints $t_i$ of *NSR*-type time trajectory are obtained as follows:

1. $\text{trans}_T = -\log(\text{NSR}(t_T) + k \cdot \text{NSR}(t_0))$,
2. $\text{trans}_0 = -\log(\text{NSR}(t_0) + k \cdot \text{NSR}(t_0))$,
3. $\text{trans}_i = \text{trans}_T + i \cdot \frac{\text{trans}_0 - \text{trans}_T}{N}$,
4. $t_i = \text{rNSR}(e^{-\text{trans}_i} - k \cdot \text{NSR}(t_0))$,

where $k$ is a hyperparameter that controls the flexibility of *NSR*-type time trajectory.

In our experiments, we found that relatively good results can be obtained when $k \in [2, 7]$. This means that when using this kind of time trajectory, one can consider setting the value of $k$ within this range.

**Sigmoid-type:** For a given starting time $t_T$ and ending time $t_0$ of the sampling, the time values at the intermediate endpoints $t_i$ of *Sigmoid*-type time trajectory are obtained as follows:

1. $\text{trans}_T = -\log(\text{NSR}(t_T))$, $\text{trans}_0 = -\log(\text{NSR}(t_0))$,
2. $\text{central} = k \cdot \text{trans}_T + (1 - k) \cdot \text{trans}_0$,
3. $\text{shift}_T = \text{trans}_T - \text{central}$, $\text{shift}_0 = \text{trans}_0 - \text{central}$,
4. $\text{scale} = \text{shift}_T + \text{shift}_0$,
5. $\text{sigm}_T = sigmoid\left(\frac{\text{shift}_T}{\text{scale}}\right)$, $\text{sigm}_0 = sigmoid\left(\frac{\text{shift}_0}{\text{scale}}\right)$,
6. $\text{sigm}_i = \text{sigm}_T + i \cdot \frac{\text{sigm}_0 - \text{sigm}_T}{N}$,
7. $\text{trans}_i = \text{scale} \cdot logistic(\text{sigm}_i) + \text{central}$,
8. $t_i = \text{rNSR}(e^{-\text{trans}_i})$,

where $k$ is a hyperparameter that controls the flexibility of *Sigmoid*-type time trajectory.

Empirically, we suggest using the *NSR*-type time trajectory. However, when NFE is less than or equal to 15, it is recommended to try using the *Sigmoid*-type time trajectory. The generation quality measured by FID of *NSR*-type time trajectory and *Sigmoid*-type time trajectory are shown in Table (9) and Table (8), respectively. Besides, we also demonstrate the efficiency of our proposed algorithms by using conventional time-quadratic trajectory in Table (6). In these experimental results, *NSR*-type time trajectory is better than time-quadratic trajectory.

### H.3 COMPARING SAMPLE QUALITY WITH DIFFERENT SAMPLERS

We show the detailed FID results of different sampling methods for DMs on CIFAR-10 and CelebA 64×64 with discrete-time or continuous-time pre-trained models in Table 1. We utilize the code and checkpoint provided in (Ho et al., 2020; Song et al., 2021c; Nichol & Dhariwal, 2021). Specifically,

Table 4: Generation quality measured by FID ↓ of different sampling methods for DMs on the pre-trained discrete-time models (Ho et al., 2020; Song et al., 2021a) of CIFAR-10 and CelebA 64×64.

| Trajectory | Initial Time | Sampling method \NFE | 12 | 15 | 20 | 50 | 100 |
|---|---|---|---|---|---|---|---|
| CIFAR-10 (discrete-time model (Ho et al., 2020), linear noise schedule) | | | | | | | |
| logSNR | $\epsilon = 10^{-3}$ | DDIM | 16.08 | 12.43 | 9.28 | 5.36 | 4.55 |
| | | DPM-Solver-2 | **5.18** | †**4.42** | 4.05 | 3.97 | 3.97 |
| | | DPM-Solver-3 | 7.39 | 4.60 | †**4.33** | †3.98 | †3.97 |
| | | SciRE-V1-2 (ours) | 5.48 | †4.55 | 3.96 | 3.66 | 3.71 |
| | | SciRE-V1-3 (ours) | 8.53 | 5.00 | †4.34 | †**3.66** | †**3.62** |
| logSNR | $\epsilon = 10^{-4}$ | DDIM | 17.40 | 13.12 | 9.54 | 5.03 | 4.13 |
| | | DPM-Solver-2 | 6.40 | †5.26 | 4.02 | 3.56 | 3.51 |
| | | DPM-Solver-3 | 9.52 | 5.17 | †**3.80** | †3.53 | †3.50 |
| | | SciRE-V1-2 (ours) | 6.48 | †5.35 | 4.01 | 3.34 | 3.27 |
| | | SciRE-V1-3 (ours) | 11.71 | 5.99 | †4.15 | †**3.30** | †**3.163** |
| NSR ($k = 2$) | $\epsilon = 10^{-3}$ | DDIM | 13.58 | 10.63 | 8.12 | 5.03 | 4.40 |
| | | DPM-Solver-2 | 4.91 | †4.51 | 4.19 | 4.00 | 3.96 |
| | | DPM-Solver-3 | 7.33 | 4.97 | †4.56 | †4.00 | †3.96 |
| | | SciRE-V1-2 (ours) | **4.49** | †4.12 | 3.74 | 3.70 | 3.76 |
| | | SciRE-V1-3 (ours) | 5.29 | 4.19 | †**3.94** | †**3.76** | †3.71 |
| NSR ($k = 2$) | $\epsilon = 10^{-4}$ | DDIM | 15.51 | 11.86 | 8.77 | 4.86 | 4.07 |
| | | DPM-Solver-2 | 5.38 | †4.46 | 3.78 | 3.53 | 3.51 |
| | | DPM-Solver-3 | 7.29 | **4.03** | †**3.66** | †3.52 | †3.50 |
| | | SciRE-V1-2 (ours) | 5.91 | †4.76 | 3.88 | 3.30 | 3.28 |
| | | SciRE-V1-3 (ours) | 9.10 | 4.52 | †4.07 | †**3.24** | †**3.167** |
| CelebA 64×64 (discrete-time model (Song et al., 2021a), linear noise schedule) | | | | | | | |
| logSNR | $\epsilon = 10^{-3}$ | DDIM | 14.37 | 11.91 | 9.66 | 6.13 | 5.15 |
| | | DPM-Solver-2 | 3.952 | †3.953 | 4.05 | 4.21 | 4.24 |
| | | DPM-Solver-3 | 3.79 | 3.91 | †4.05 | †4.26 | †4.25 |
| | | SciRE-V1-2 (ours) | 5.39 | †4.51 | 3.76 | 3.49 | 3.71 |
| | | SciRE-V1-3 (ours) | 4.91 | 3.65 | †**3.29** | †**3.09** | †3.41 |
| logSNR | $\epsilon = 10^{-4}$ | DDIM | 12.81 | 10.28 | 7.98 | 4.52 | 3.59 |
| | | DPM-Solver-2 | **3.26** | †3.14 | 2.92 | 2.82 | 2.82 |
| | | DPM-Solver-3 | 3.93 | **2.91** | †2.85 | †2.82 | †2.81 |
| | | SciRE-V1-2 (ours) | 4.29 | †3.70 | 2.87 | 2.37 | 2.43 |
| | | SciRE-V1-3 (ours) | 5.04 | 3.43 | †**2.58** | †**2.06** | †**2.20** |
| NSR ($k = 2$) | $\epsilon = 10^{-3}$ | DDIM | 13.08 | 10.99 | 8.96 | 5.88 | 4.40 |
| | | DPM-Solver-2 | 5.39 | †4.93 | 4.37 | 4.24 | 4.236 |
| | | DPM-Solver-3 | 6.14 | 4.77 | †4.41 | †4.24 | †4.24 |
| | | SciRE-V1-2 (ours) | 4.61 | †4.20 | 3.58 | 3.56 | 3.76 |
| | | SciRE-V1-3 (ours) | 4.75 | 3.33 | †**3.04** | †**3.15** | †3.51 |
| NSR ($k = 2$) | $\epsilon = 10^{-4}$ | DDIM | 11.88 | 9.59 | 7.53 | 4.38 | 3.54 |
| | | DPM-Solver-2 | 3.11 | †2.91 | 2.88 | 2.79 | 2.81 |
| | | DPM-Solver-3 | **2.94** | 2.88 | †2.87 | †2.80 | †2.81 |
| | | SciRE-V1-2 (ours) | 3.95 | †3.39 | 2.61 | 2.31 | 2.43 |
| | | SciRE-V1-3 (ours) | 4.47 | **2.68** | †**2.29** | †**2.03** | †**2.20** |

we employ their *checkpoint_8* of the "VP deep" type. In this table, we compare the FID achieved by our proposed SciRE-V1 with the best FID reported in existing literature at the same NFE. We consistently use the NSR-type time trajectory with parameter $k = 3.1$ for SciRE-V1 on the discrete models of CIFAR-10 and CelebA 64×64 datasets. For continuous models on the CIFAR-10 dataset, we use a Sigmoid-type time trajectory with parameter $k = 0.65$ for the SciRE-Solver when the NFE is less than 15. When NFE is greater than or equal to 15, we consistently use an NSR-type time trajectory with $k = 3.1$. In order to objectively compare the quality of generated samples for the CelebA 64×64 dataset, given the presence of different FID statistical data, we utilized the FID stats

employed by Liu et al. (2022b) in Tables 1, 6 and 9, and utilized the FID stats employed by Lu et al. (2022b) in Tables 4, 3 and 2. Figure 7 illustrates the FIDs achieved by different samplers at various NFE levels. Moreover, in Table 5, we also evaluate SciRE-V1, DPM-Solver and DDIM with the same settings on the pre-trained model of high-resolution ImageNet 128×128 dataset (Dhariwal & Nichol, 2021), refer to Figures 11 and 12 for the comparisons of generated samples. In all tables, the results $^{\dagger}$ means the actual NFE is smaller than the given NFE.

In Table 4, in order to ensure fairness, we compare the generation performance of SciRE-V1 with DPM-Solver and DDIM on discrete models (Ho et al., 2020; Song et al., 2021a) of CIFAR-10 and CelebA 64×64 datasets using the same trajectories, settings and codebase. In this experiment, we employ different time trajectories to evaluate the sampling performance of each sampling algorithm, such as the NSR trajectory and the logNSR trajectory (Lu et al., 2022b). Unlike in Table 1 with parameter $k = 3.1$, we consistently use parameter $k = 2$ for the NSR time trajectory in Table 4, in order to showcase the impact of different $k$ values on the samplers. Meanwhile, we also compare the performance of generative samples for these three samplers at different sampling endpoints, such as $1e − 3$ and $1e − 4$.

Tables 1 and 4 demonstrate that the SciRE-V1 attains SOTA sampling performance with limited NFE on both discrete-time and continuous-time DMs in comparison to existing training-free sampling algorithms. Such as, in Table 1, we achieve 3.48 FID with 12 NFE and 2.42 FID with 20 NFE for continuous-time DMs on CIFAR10, respectively. Furthermore, with fewer NFE, SciRE-V1 surpass the benchmark values demonstrated in the original paper of the proposed pre-trained model. For example, we reach SOTA value of 2.40 FID with no more than 100 NFE for continuous-time DMs and of 3.15 FID with 84 NFE for discrete-time DMs on CIFAR-10, as well as of 2.17 FID with 18 NFE for discrete-time DMs on CelebA 64×64. Moreover, SciRE-V1 can also achieve SOTA sampling performance within 100 NFE for both the NSR time trajectory with different parameter values $k$ and the logSNR time trajectory, as shown in Tables 1 and 4. Especially, in Table 4, DPM-Solver is more likely to achieve better sampling performance within 15 NFE for the logSNR time trajectory and the NSR time trajectory with $k = 2$. However, when NFE exceeds 15, ScrRE-Solver becomes more advantageous. Moreover, when the endpoint of the sampling is set at $1e − 4$, both with the logSNR time trajectory and NSR time trajectory ($k = 2$), SciRE-V1 can achieve SOTA sampling performance between 50 NFE and 100 NFE.

In Table 5, we also evaluate SciRE-V1, DPM-Solver and DDIM on the high-resolution ImageNet 128×128 dataset (Dhariwal & Nichol, 2021). For the sake of fairness, we use the same uniform time trajectory, the same codebase, and the same settings to evaluate SciRE-V1-2, DPM-Solver-2, and DDIM for 10, 12, 15, 20, and 50 NFEs. The numerical experiment results report that SciRE-V1-2 achieved 5.58 FID with 10 NFE and 3.67 FID with 20 NFE, respectively, while DMP-Solver-2 only achieved 4.17 FID with 50 NFE. In all these different NFEs, SciRE-V1-2 outperforms DPM-Solver-2.

In summary, within 20 NFE, SciRE-V1 with NSR trajectory ($k = 3.1$) achieves better FID than existing training-free solvers (Bao et al., 2022; Song et al., 2021a; Zhang & Chen, 2023; Lu et al., 2022b; Liu et al., 2022b; Li et al., 2023) for CIFAR-10 and CelebA 64×64 datasets, as shown Table 1. Meanwhile, within 100 NFE (or even 1000 NFE), existing solvers in the context of discrete models on CIFAR-10 dataset are hardly able to achieve an FID below 3.45, as shown in Table 1. On the other hand, SciRE-V1, with different time trajectories such as logNSR trajectory and NSR trajectory, can achieve an FID below 3.17, and even surpass the 3.16 FID obtained by DDPM at 1000 NFE, as shown in Tables 1 and 4. For the continuous VP-type model on CIFAR-10, SciRE-V1 also surpasses the 2.41 FID obtained by Song et al. (2021c) using SDE solver with 1000 NFE. In Table 4, under the time trajectories, settings and the same codebase, SciRE-V1 outperforms the DPM-Solver (Lu et al., 2022b) widely used in stable diffusion (Rombach et al., 2021). Specifically, SciRE-V1 achieves an FID of 3.16 and 2.03 within 100 NFE on CIFAR-10 and CelebA 64×64 datasets respectively, whereas DPM-Solver struggles to achieve FID values lower than 3.50 and 2.79 respectively on the same datasets. Furthermore, the FID comparison on the high-resolution 128×128 dataset presented in Table 5 suggests that SciRE-V1 also possesses advantages in sample generation tasks involving high-resolution image datasets. For more random sampling sample comparisons on different high-resolution (≥128×128) image datasets, please refer to Figures 11, 12, 13, 17, 18, 19, and 20.

Table 5: Generation quality measured by FID ↓ of different sampling methods with the same codebase for DMs on the pre-trained discrete-time model of Imagenet 128×128 (Dhariwal & Nichol, 2021).

| Trajectory | Initial Time | Sampling method \NFE | 10 | 12 | 15 | 20 | 50 |
|---|---|---|---|---|---|---|---|
| Imagenet 128×128 (with classifier guidance: scale=1.25, under the same codebase) | | | | | | | |
| Uniform time | $\epsilon = 10^{-3}$ | DDIM | 9.33 | 7.55 | 6.07 | 4.91 | 3.51 |
| | | DPM-Solver-2 | 10.17 | 7.78 | †6.68 | 5.46 | 4.17 |
| | | SciRE-V1-2 (our) | 5.58 | 4.73 | †4.27 | 3.67 | 3.36 |

Table 6: SciRE-V1 with time-quadratic trajectory

| Initial time | Sampling method \ NFE | 12 | 15 | 20 | 50 | 100 |
|---|---|---|---|---|---|---|
| CIFAR-10 (discrete-time model (Ho et al., 2020)) | | | | | | |
| $\epsilon = 10^{-3}$ | SciRE-V1-2 | **4.86** | †**4.10** | **3.56** | 3.74 | 3.83 |
| | SciRE-V1-3 | 19.37 | 11.18 | †**7.48** | †3.94 | †3.85 |
| $\epsilon = 10^{-4}$ | SciRE-V1-2 | 6.13 | †5.12 | 3.83 | **3.31** | 3.27 |
| | SciRE-V1-3 | 22.39 | 13.09 | †8.54 | †3.45 | †**3.22** |
| CIFAR-10 (VP deep continuous-time model (Song et al., 2021c)) | | | | | | |
| $\epsilon = 10^{-4}$ | SciRE-V1-2 | **5.00** | †4.24 | 3.23 | 2.59 | 2.53 |
| | SciRE-V1-3 | 12.53 | 7.33 | †5.43 | †2.64 | †**2.50** |
| | SciRE-V1-agile | 5.03 | **4.24** | **3.21** | **2.59** | 2.51 |
| Initial time | Sampling method \ NFE | 12 | 15 | 20 | 30 | 50 |
| CelebA 64×64 (discrete-time model (Song et al., 2021a)) | | | | | | |
| $\epsilon = 10^{-3}$ | SciRE-V1-2 | 5.83 | †4.67 | 3.92 | 3.77 | 3.86 |
| | SciRE-V1-3 | 8.72 | 5.06 | †3.81 | †3.31 | †3.56 |
| $\epsilon = 10^{-4}$ | SciRE-V1-2 | **4.24** | †**3.27** | **2.46** | 2.23 | 2.20 |
| | SciRE-V1-3 | 11.08 | 5.62 | †3.53 | †**2.13** | †**2.03** |

### H.4 ABLATIONS STUDY

#### H.4.1 DIFFERENT ORDERS AND STARTING TIMES

**Order**  We compare the sample quality with different orders of SciRE-V1-2,3. However, in practice, the actual NFE may be smaller than the given NFE, for example, given the NFE=15, the actucal NFE of SciRE-V1-2 is 14. To mitigate this problem, we propose the SciRE-V1-agile method for continuous models. We compare the results of models with different orders on CIFAR-10 and CelebA 64×64 datasets. Our results indicate that if NFE is less than 20, SciRE-V1-2 outperforms SciRE-V1-3, or the latter variant is superior – depending on the specific use case.

**Starting time**  We also compare SciRE-V1-2,3 with different starting times $\epsilon = 10^{-3}$ and $\epsilon = 10^{-4}$. Corresponding results are placed in Tables 6 and 9. We use time-quadratic trajectory and *NSR*-type

Table 7: SciRE-V1-agile with NSR trajectory and starting time $1e - 4$.

| $k$ | $\phi_1(m)$\ NFE | 12 | 15 | 20 | 50 | 100 |
|---|---|---|---|---|---|---|
| CIFAR-10 (VP deep continuous-time model (Song et al., 2021c)) | | | | | | |
| $k = 3.1$ | $\phi_1(m) = \phi_1(3)$ | 6.93 | 3.73 | **2.42** | 2.52 | 2.48 |
| | $\phi_1(m) = \frac{e-1}{e}$ | 6.79 | **2.57** | 2.48 | 2.61 | 2.41 |
| $k = 2.2$ | $\phi_1(m) = \phi_1(3)$ | 4.06 | 3.34 | 2.54 | **2.51** | 2.42 |
| | $\phi_1(m) = \frac{e-1}{e}$ | 6.15 | 3.39 | 2.57 | 2.61 | **2.40** |

Table 8: Comparison between different time trajectories, starting time is $1e-3$.

| CIFAR-10 (VP deep continuous-time model (Song et al., 2021c)) | | | |
| --- | --- | --- | --- |
| Sampling method | Sampling method\ NFE | 12 | 15 |
| SciRE-V1-3 | *NSR*-type($k = 3.2 - 0.005 \cdot$NFE) | 4.41 | **3.06** |
| | *Sigmoid*-type ($k = 0.65$) | **3.48** | 3.47 |

Table 9: SciRE-V1 with NSR trajectory ($k = 3.1$).

| Initial time | Sampling method \ NFE | 12 | 15 | 20 | 50 | 100 |
| --- | --- | --- | --- | --- | --- | --- |
| CIFAR-10 (discrete-time model (Ho et al., 2020)) | | | | | | |
| $\epsilon = 10^{-3}$ | SciRE-V1-2 | **4.41** | [†]4.09 | **3.67** | 3.70 | 3.80 |
| | SciRE-V1-3 | 4.68 | **4.00** | [†]3.72 | [†]3.84 | [†]3.77 |
| $\epsilon = 10^{-4}$ | SciRE-V1-2 | 5.86 | [†]4.77 | 3.87 | 3.28 | 3.27 |
| | SciRE-V1-3 | 8.28 | 4.51 | [†]3.96 | [†]**3.23** | [†]**3.17** |
| Initial time | Sampling method \ NFE | 12 | 15 | 20 | 30 | 50 |
| CIFAR-10 (VP deep continuous-time model (Song et al., 2021c)) | | | | | | |
| | SciRE-V1-2 | **5.49** | [†]4.19 | 3.02 | 2.55 | 2.47 |
| $\epsilon = 10^{-4}$ | SciRE-V1-3 | 6.29 | **3.39** | [†]2.68 | [†]2.56 | [†]**2.44** |
| | SciRE-V1-agile | 6.93 | 3.73 | **2.42** | **2.52** | 2.48 |
| CelebA 64×64 (discrete-time model (Song et al., 2021a)) | | | | | | |
| $\epsilon = 10^{-3}$ | SciRE-V1-2 | 4.79 | [†]4.28 | 3.86 | 3.69 | 3.82 |
| | SciRE-V1-3 | 5.01 | 3.32 | [†]3.12 | [†]3.09 | [†]3.40 |
| $\epsilon = 10^{-4}$ | SciRE-V1-2 | **3.91** | [†]3.38 | 2.56 | 2.41 | 2.30 |
| | SciRE-V1-3 | 4.07 | **2.53** | [†]**2.17** | [†]**2.03** | [†]**2.02** |

time trajectory for both SciRE-V1-2 and SciRE-V1-3 on CIFAR-10 and CelebA 64 ×64 datasets. In our study on the CIFAR-10 dataset, we have observed that employing a sampling method with $\epsilon = 10^{-3}$ results in superior sample quality for both continuous and discrete models when NFE is restricted to either 12 or 15. However, for NFE values greater than 15, we recommend opting for $\epsilon = 10^{-4}$ to ensure the generation of high-quality samples. Moreover, in our analysis of the CelebA 64 ×64 dataset, we have found that $\epsilon = 10^{-4}$ consistently yields better results than $\epsilon = 10^{-3}$ across different orders and NFEs. It is noteworthy that for NFE=20, SciRE-V1-2,3 show promising results that are on par with the former.

### H.4.2  $\phi_1(m) = \phi_1(3)$ or $\phi_1(m) = \frac{e-1}{e}$

When running our proposed SciRE-V1-$k$ in Algorithm 1 and Algorithm 2, it is necessary to assign a value $m$ to $\phi_1(m)$. As stated in Corollary 3.1, when assigning $m$, we need to ensure that $m \geq 3$. Considering that the limit of $\phi_1(m)$ is $\frac{e-1}{e}$, i.e., $\lim_{m \to \infty} \phi_1(m) = \lim_{m \to \infty} \sum_{k=1}^{m} \frac{(-1)^{k-1}}{k!} = \frac{e-1}{e}$, then our experiments only consider these two extreme cases, i.e., we only choose to allocate $m$ as 3 or directly set $\phi_1(m) = \frac{e-1}{e}$. We provide ablation experiments for these two cases in Table 7. In case of $\phi_1(m) = \frac{e-1}{e}$, we reach 2.40 FID SOTA value with 100 NFE on CIFAR-10 dataset.

## I  COMPARISONS OF SAMPLES GENERATED

In this section, we provide sample comparisons of random sampling using SciRE-V1, DPM-Solver, and DDIM with the same codebase on different datasets, as depicted in Figures 11 to 20. Additionally, we present some generated samples on CIFAR-10, CelebA 64×64, Imagenet 256×256 and Imagenet 512×512, which reported in Figures 21 to 28.

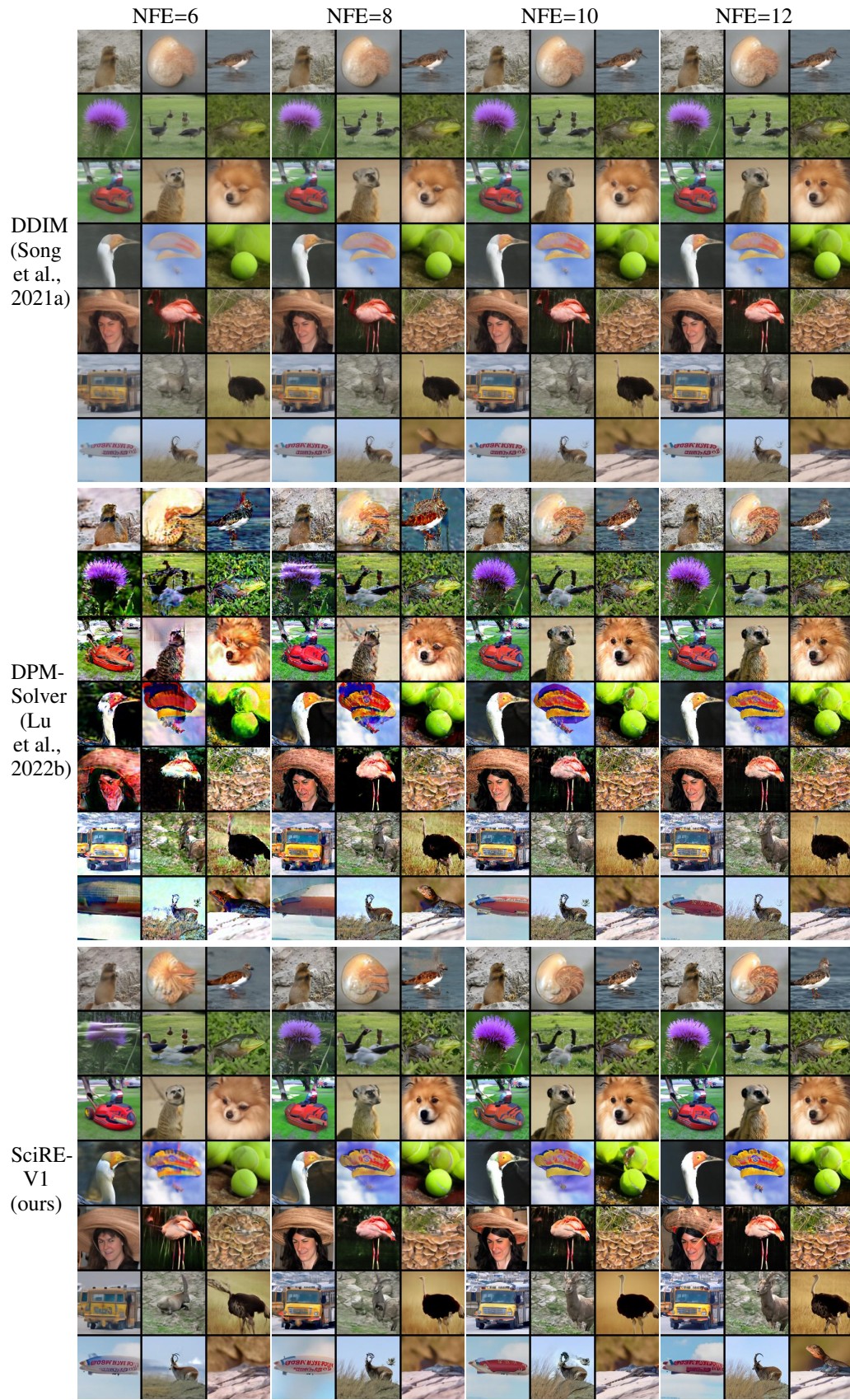

Figure 11: Generated samples of the pre-trained DM on ImageNet 128×128 (classifier scale: 1.25) using 6-12 sampling steps from different sampling methods with the same settings and codebase.

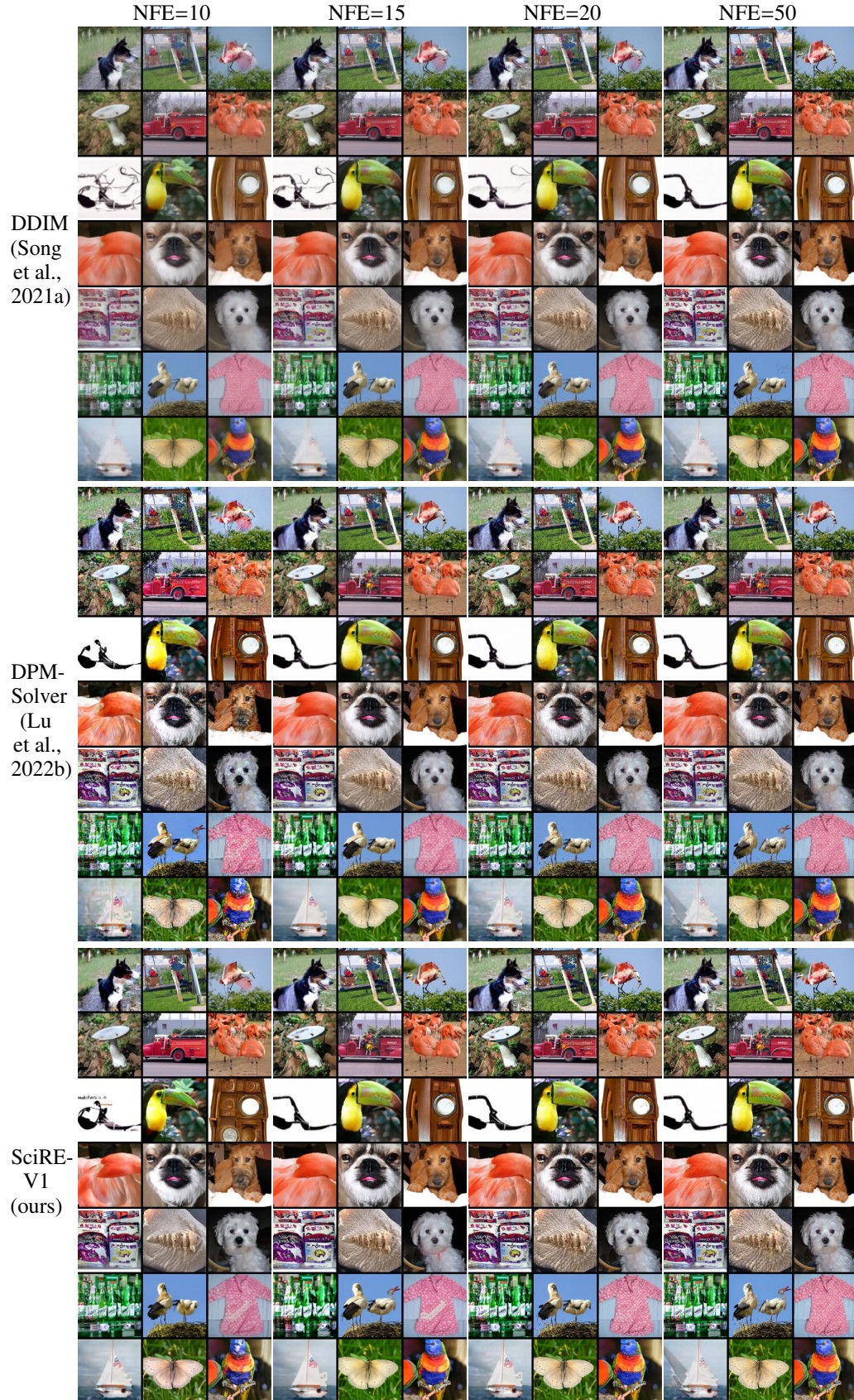

Figure 12: Generated samples of the pre-trained DM on ImageNet 128×128 (classifier scale: 1.25) using 10-50 sampling steps from different sampling methods with the same settings and codebase.

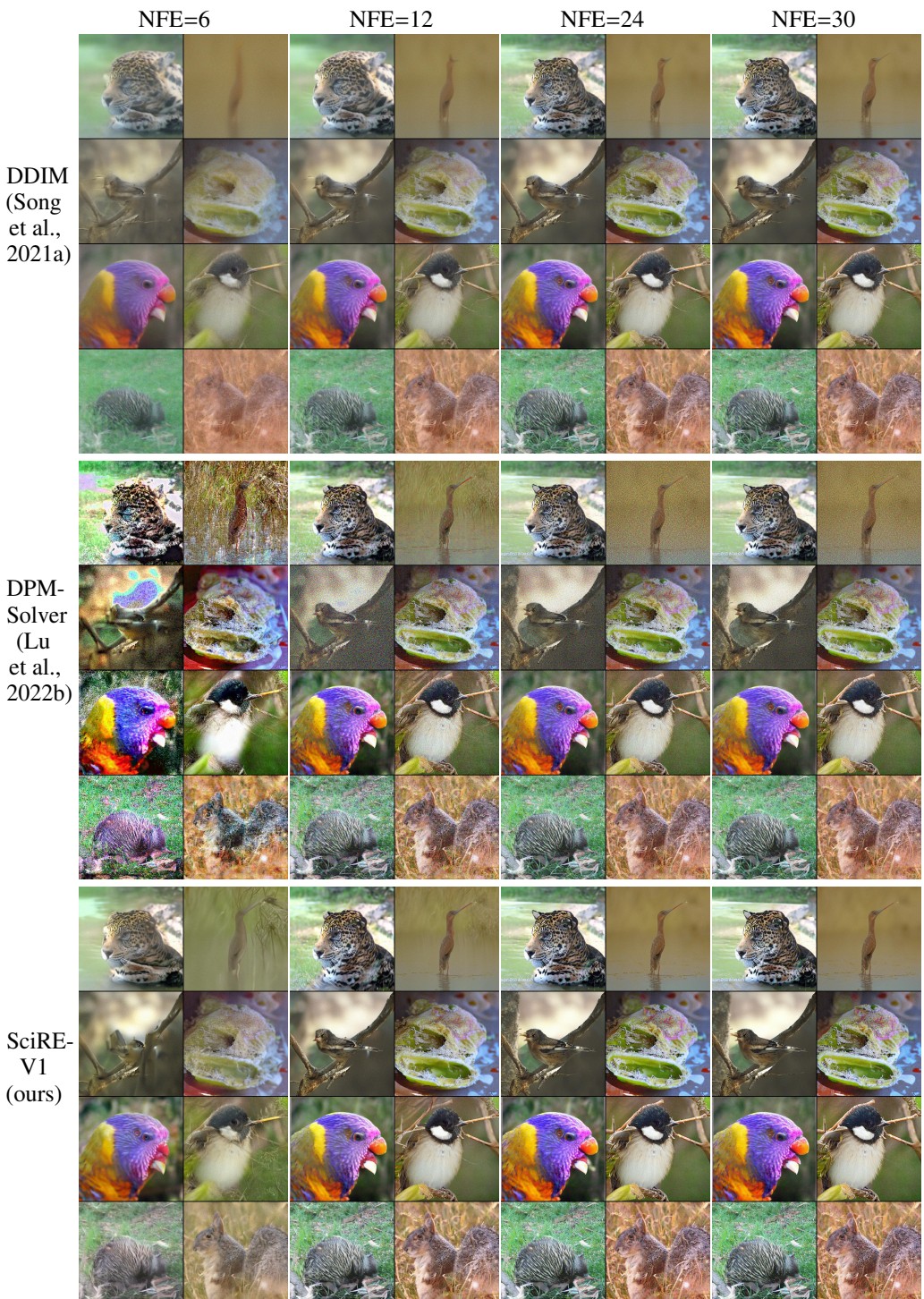

Figure 13: Generated samples of the pre-trained DM on ImageNet 512×512 (classifier scale: 1) using 6-30 sampling steps from different sampling methods with the same settings and codebase.

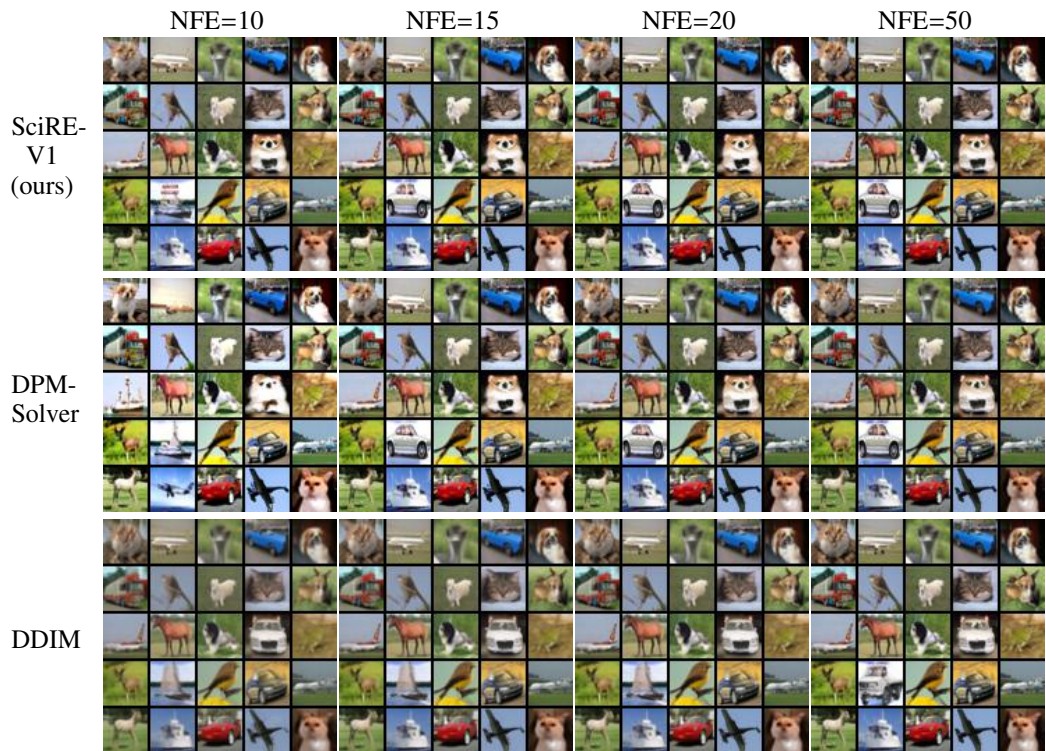

Figure 14: Random samples with the same random seed were generated by DDIM (Song et al., 2021a) (uniform time steps), DPM-Solver (Lu et al., 2022b) (logSNR time steps), and SciRE-V1 (NSR time steps, $k = 3.1$), employing the pre-trained discrete-time DM (Ho et al., 2020) on CIFAR-10.

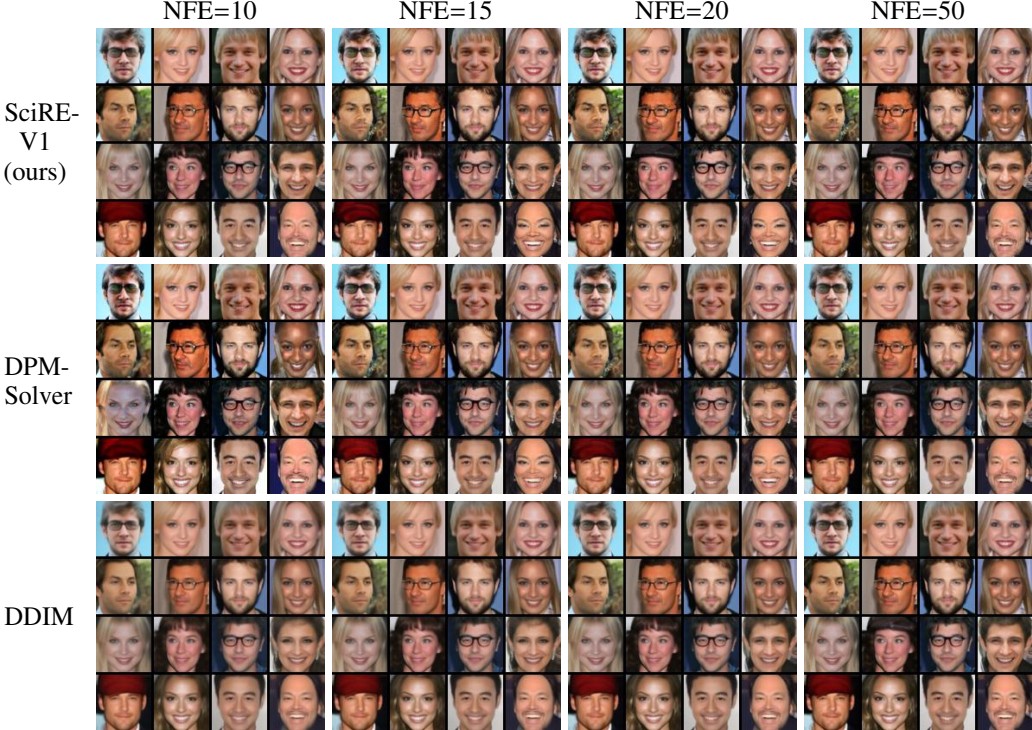

Figure 15: Random samples with the same random seed were generated by DDIM (uniform time steps), DPM-Solver (logSNR time steps), and SciRE-V1 (NSR time steps, $k = 3.1$), employing the pre-trained discrete-time DM (Song et al., 2021a) on CelebA 64×64.

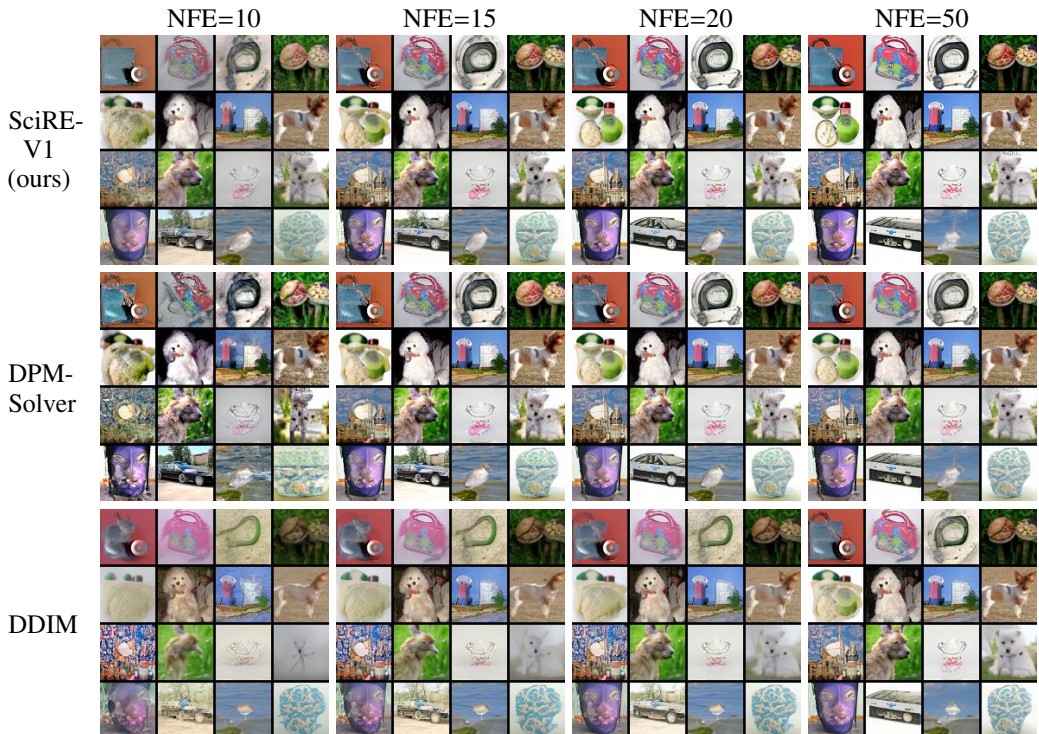

Figure 16: Random samples with the same random seed were generated by DDIM (uniform time steps), DPM-Solver (logSNR time steps), and SciRE-V1 (NSR time steps, $k = 3.1$), employing the pre-trained discrete-time DM (Nichol & Dhariwal, 2021) on ImageNet 64×64.

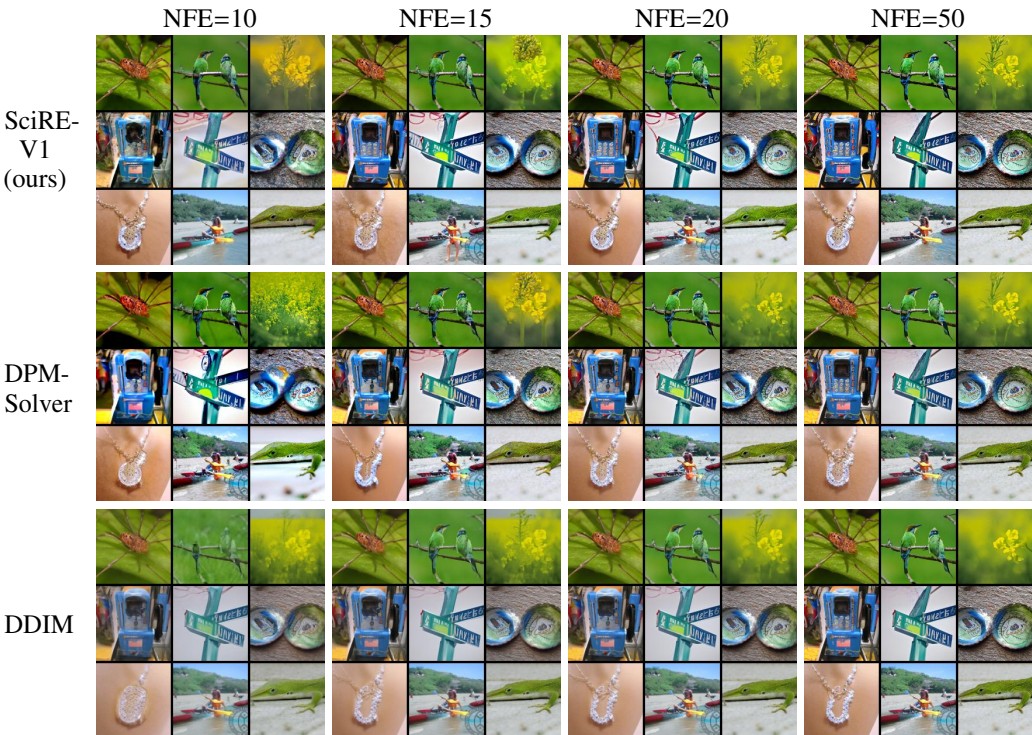

Figure 17: Random samples with the same random seed were generated by DDIM (uniform time steps), DPM-Solver (logSNR time steps), and SciRE-V1 (NSR time steps, $k = 3.1$), employing the pre-trained DM (Dhariwal & Nichol, 2021) on ImageNet 128×128 (classifier scale: 1.25).

NFE=10          NFE=15          NFE=20          NFE=50

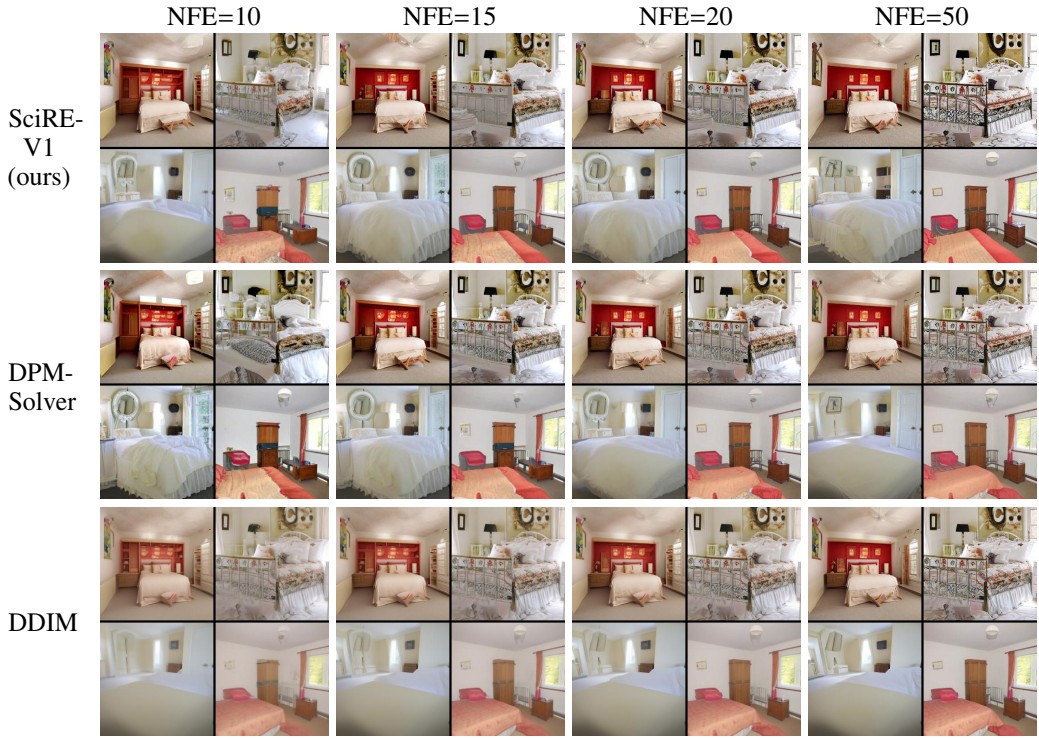

Figure 18: Random samples with the same random seed were generated by DDIM (uniform time steps), DPM-Solver (logSNR time steps), and SciRE-V1 (NSR time steps, $k = 3.1$), employing the pre-trained discrete-time DM (Dhariwal & Nichol, 2021) on LSUN bedroom 256×256.

NFE=10          NFE=15          NFE=20          NFE=50

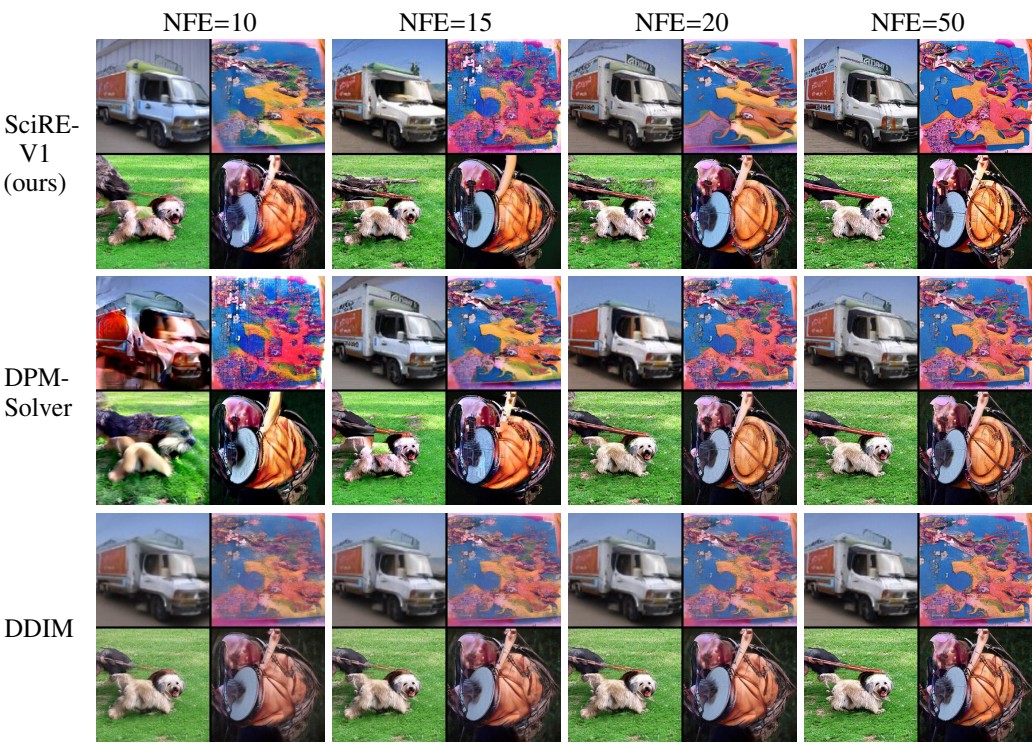

Figure 19: Random samples with the same random seed were generated by DDIM (uniform time steps), DPM-Solver (logSNR time steps), and SciRE-V1 (NSR time steps, $k = 3.1$), employing the pre-trained DPM (Dhariwal & Nichol, 2021) on ImageNet 256×256 (classifier scale: 2.5).

NFE=10      NFE=15      NFE=20      NFE=50

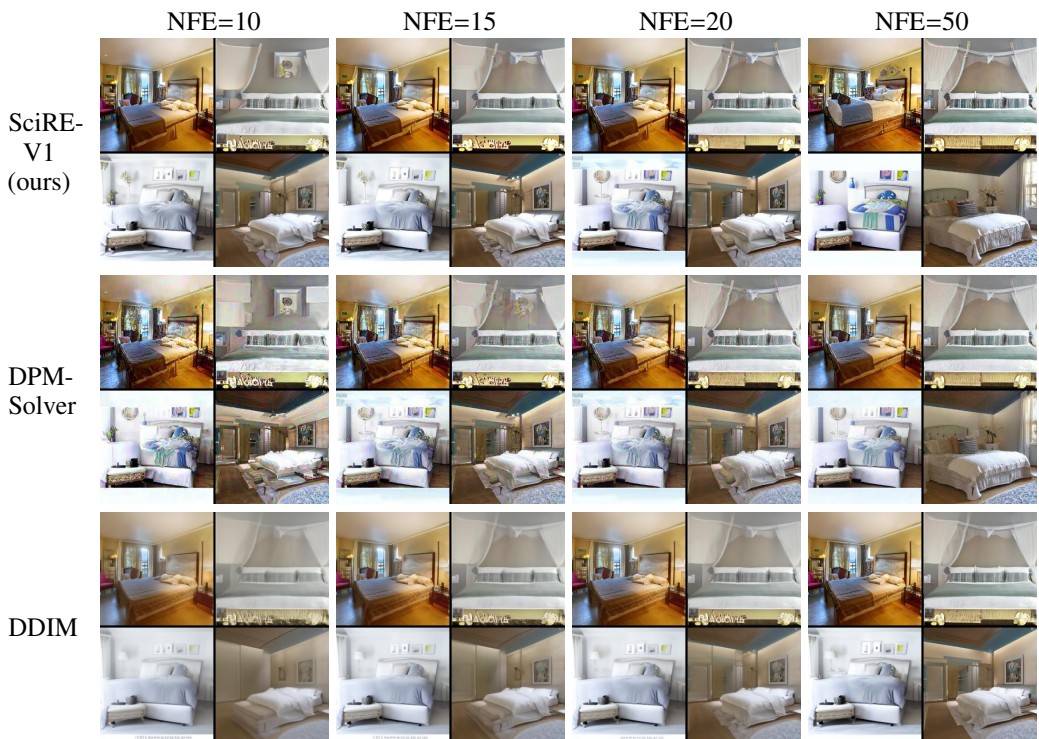

Figure 20: Random samples with the same random seed were generated by SciRE-V1, DPM-Solver (Lu et al., 2022b), and DDIM (Song et al., 2021a) with the consistently uniform time steps, employing the pre-trained discrete-time DM (Dhariwal & Nichol, 2021) on LSUN bedroom 256×256.

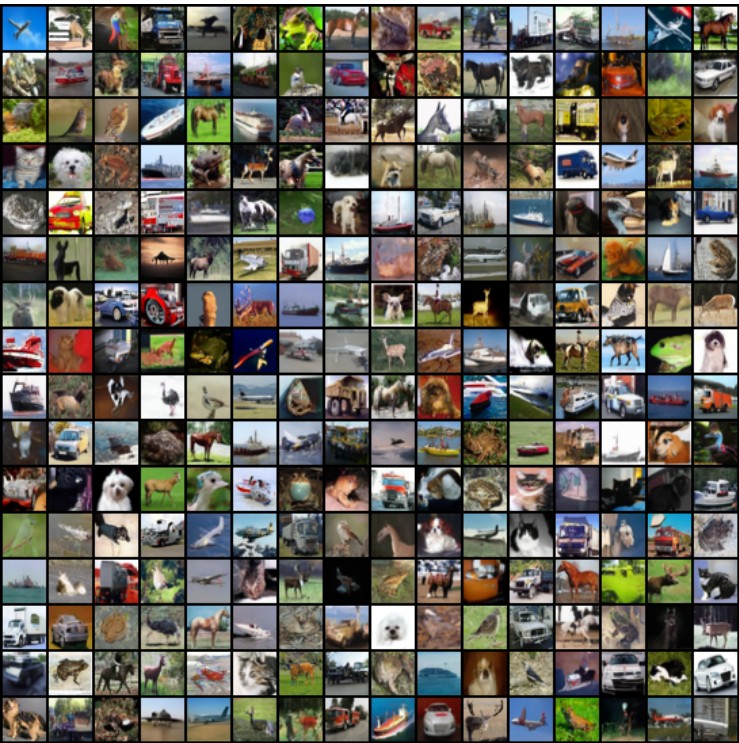

Figure 21: Random samples were generated by SciRE-V1 with 12 NFE, employing the pre-trained discrete-time DM (Song et al., 2021c) on continious-time CIFAR-10. We achieve an 3.48 FID by using the *Sigmoid*-type time trajectory with $k = 0.65$, and setting the initial time as $\epsilon = 10^{-3}$.

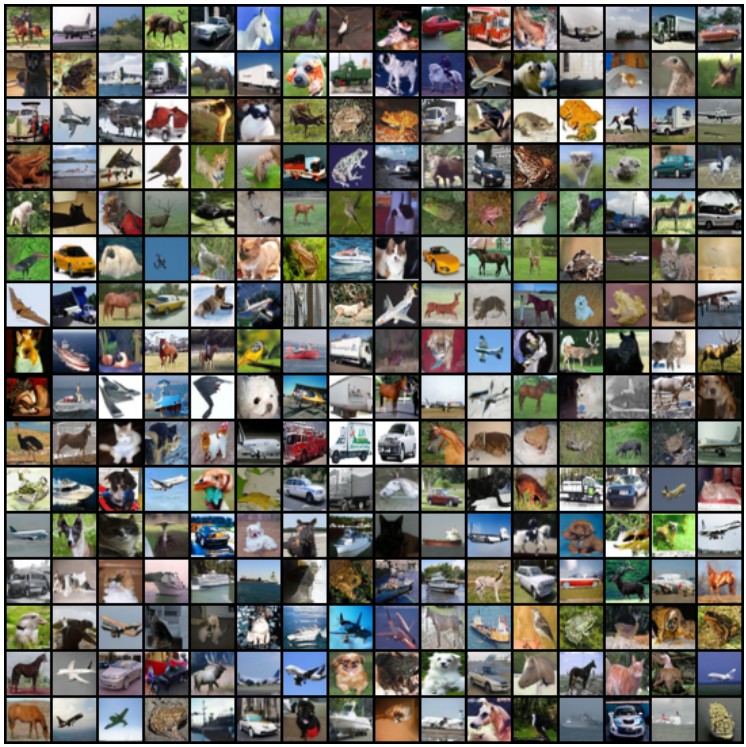

Figure 22: Random samples were generated by SciRE-V1 with 20 NFE, employing the pre-trained discrete-time DM (Song et al., 2021c) on continious-time CIFAR-10. We achieve an 2.42 FID by using the *NSR*-type time trajectory with $k = 3.10$, and setting the initial time as $\epsilon = 10^{-4}$.

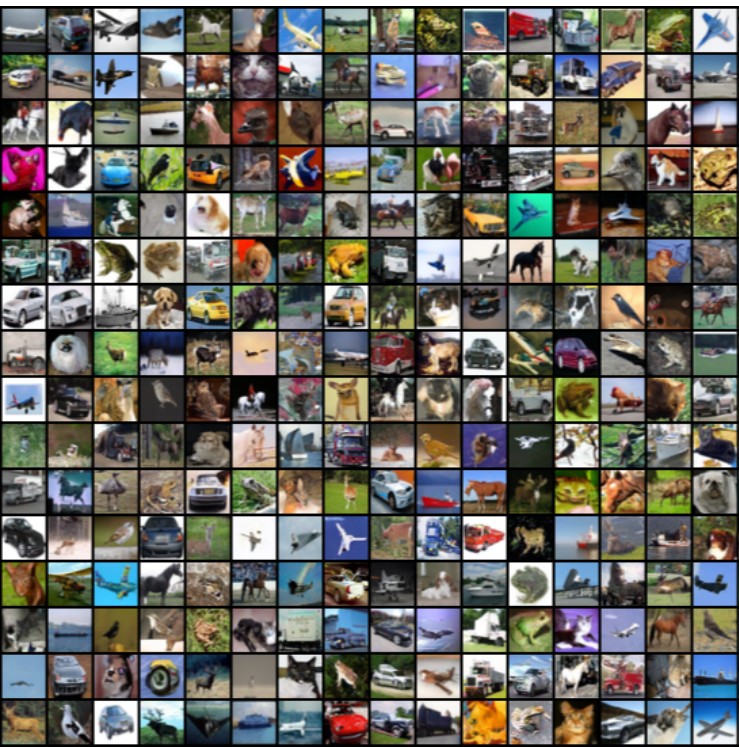

Figure 23: Random samples were generated by SciRE-V1 with 100 NFE, employing the pre-trained discrete-time DM (Song et al., 2021c) on continious-time CIFAR-10. We achieve an 2.40 FID by using the *NSR*-type time trajectory with $k = 3.10$, and setting the initial time as $\epsilon = 10^{-4}$.

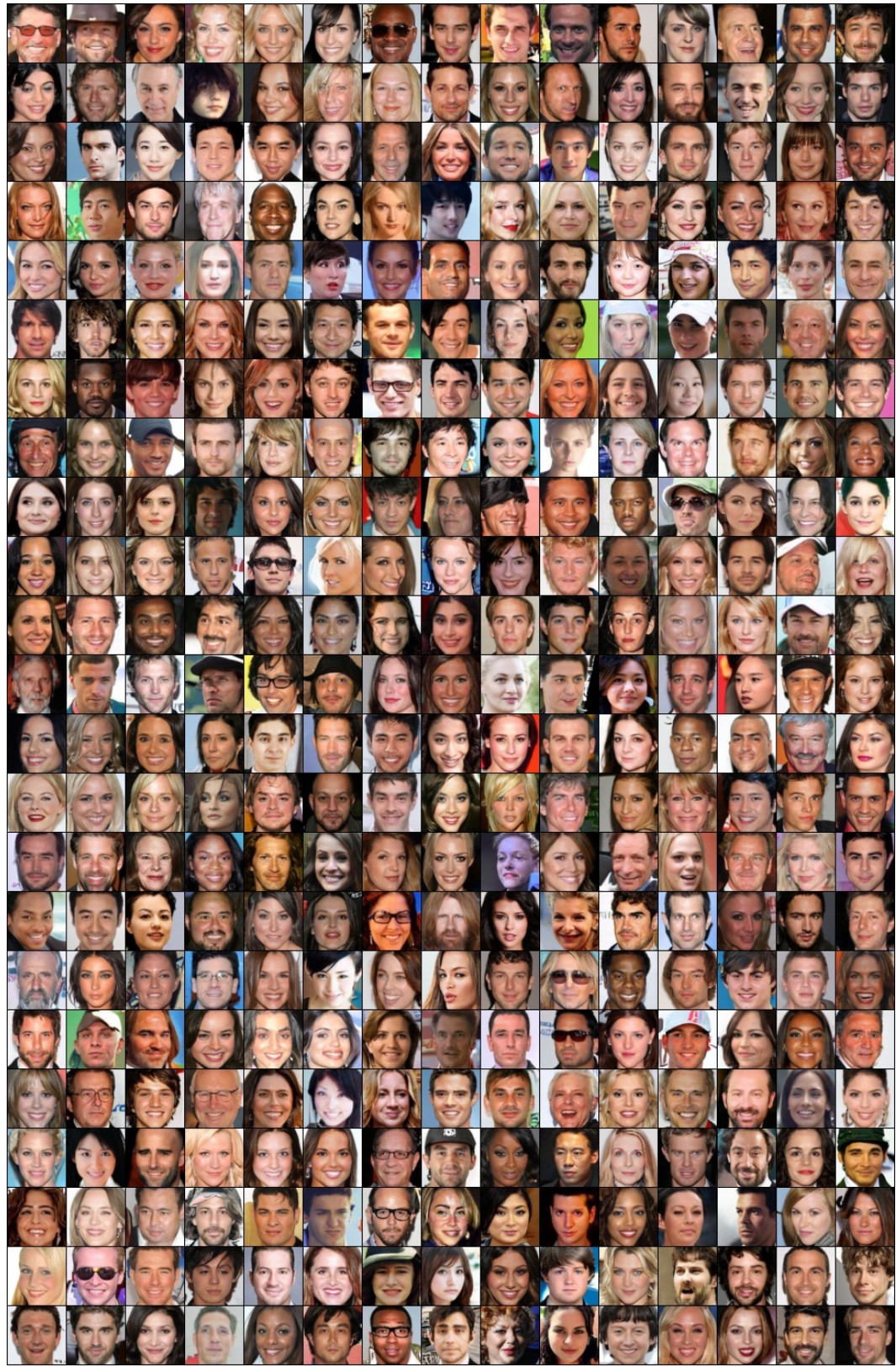

Figure 24: Random samples were generated by SciRE-V1 with 18 NFE, employing the pre-trained discrete-time DM (Song et al., 2021a) on CelebA 64×64. We achieve an 2.17 FID by using the *NSR*-type time trajectory with $k = 3.10$, and setting the initial time as $\epsilon = 10^{-4}$.

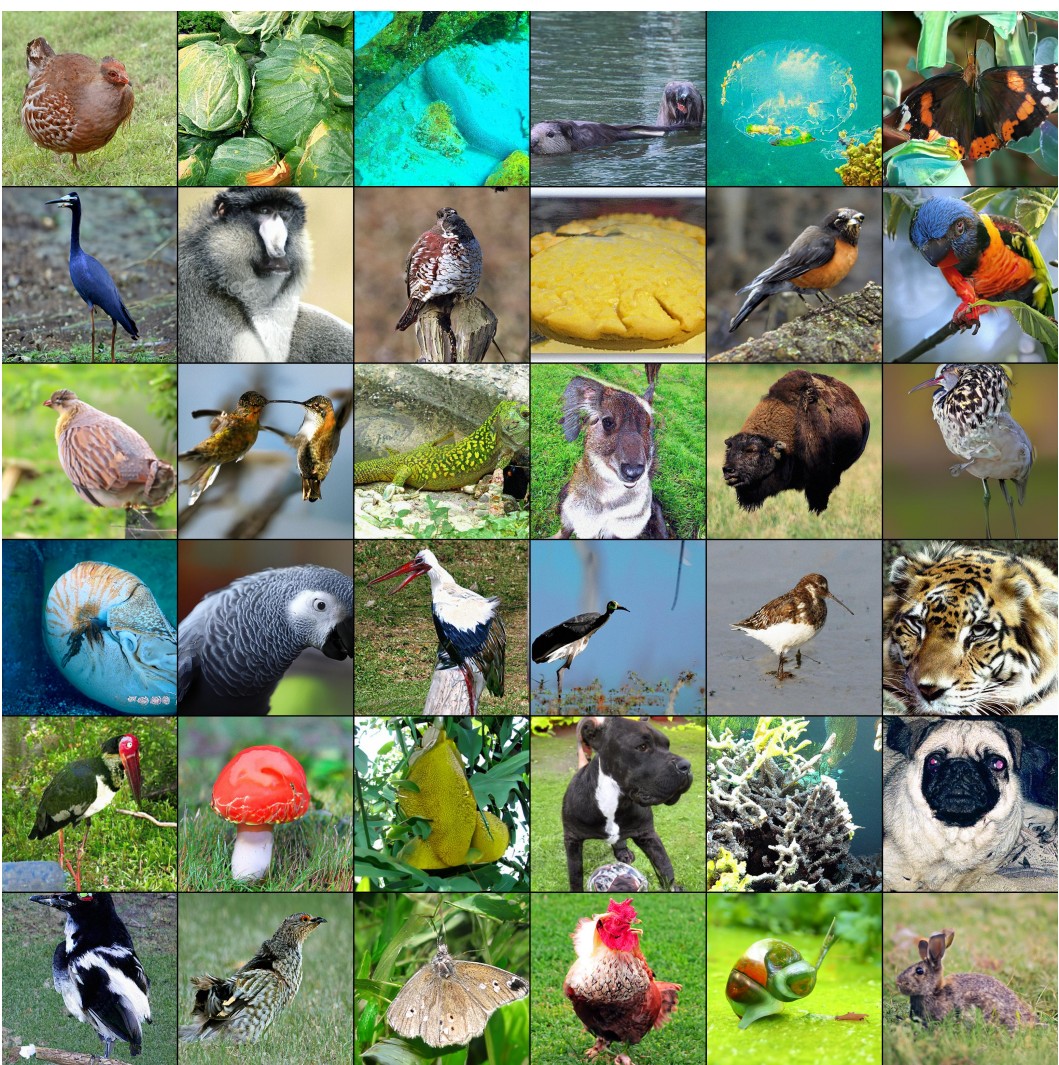

Figure 25: Random samples were generated by SciRE-V1-2 with 20 NFE and the uniform time steps, using the pre-trained discrete-time DM (Dhariwal & Nichol, 2021) on Imagenet 512×512 (classifier scale: 4).

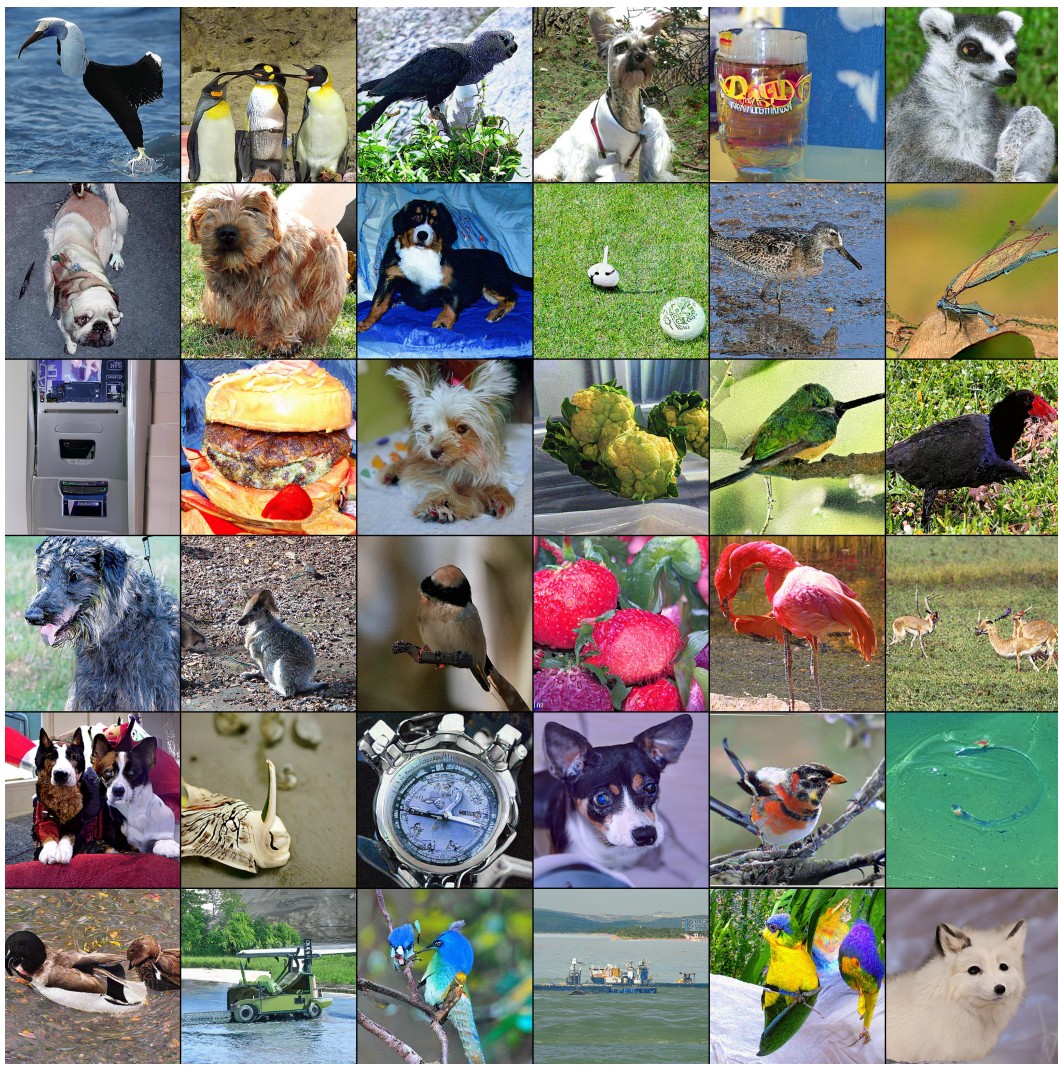

Figure 26: Random samples were generated by SciRE-V1-3 with 18 NFE and the uniform time steps, using the pre-trained discrete-time DM (Dhariwal & Nichol, 2021) on Imagenet 512×512 (classifier scale: 1).

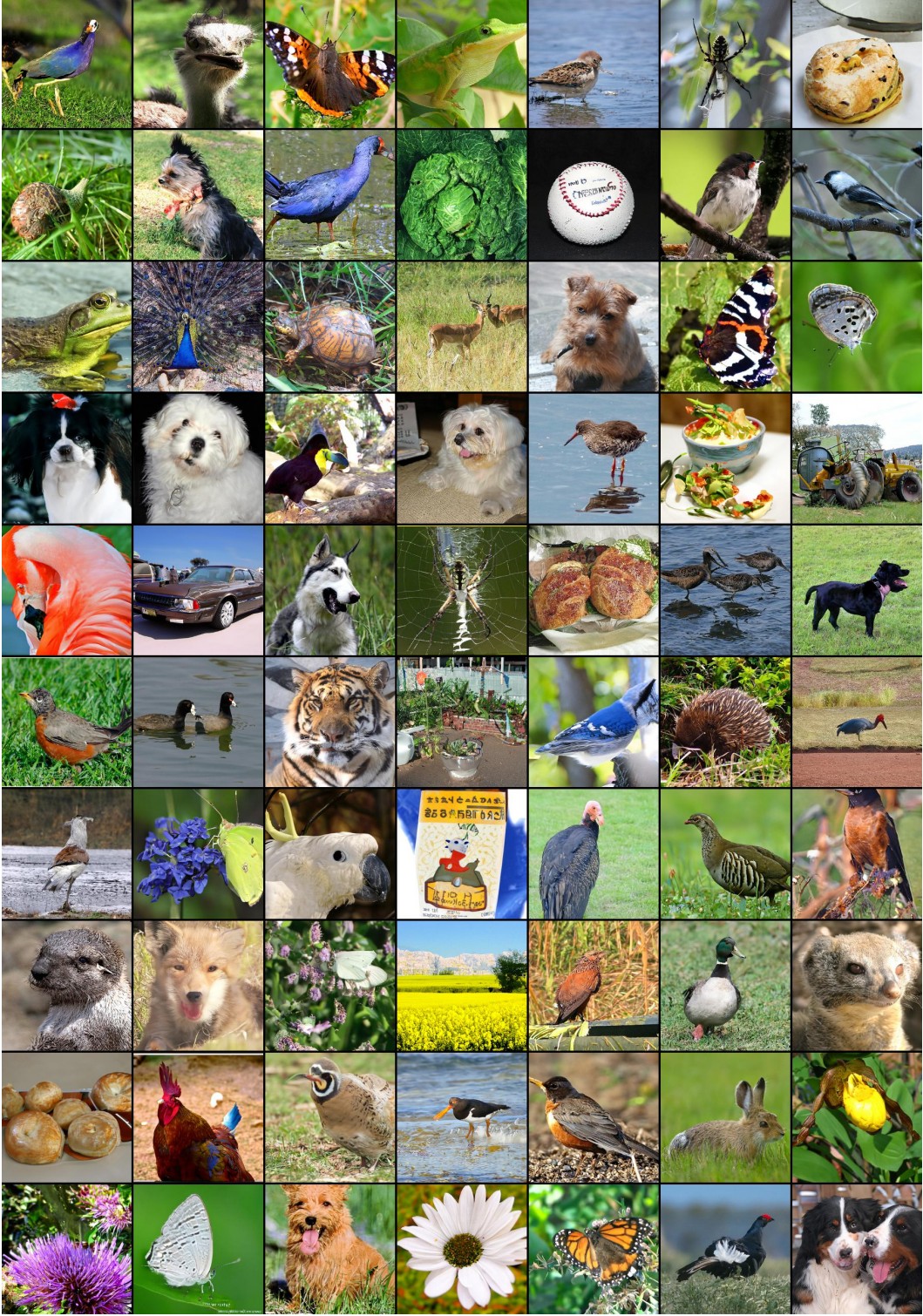

Figure 27: Random samples were generated by SciRE-V1-2 with 20 NFE and the uniform time steps, using the pre-trained discrete-time DM (Dhariwal & Nichol, 2021) on Imagenet 256×256 (classifier scale: 1).

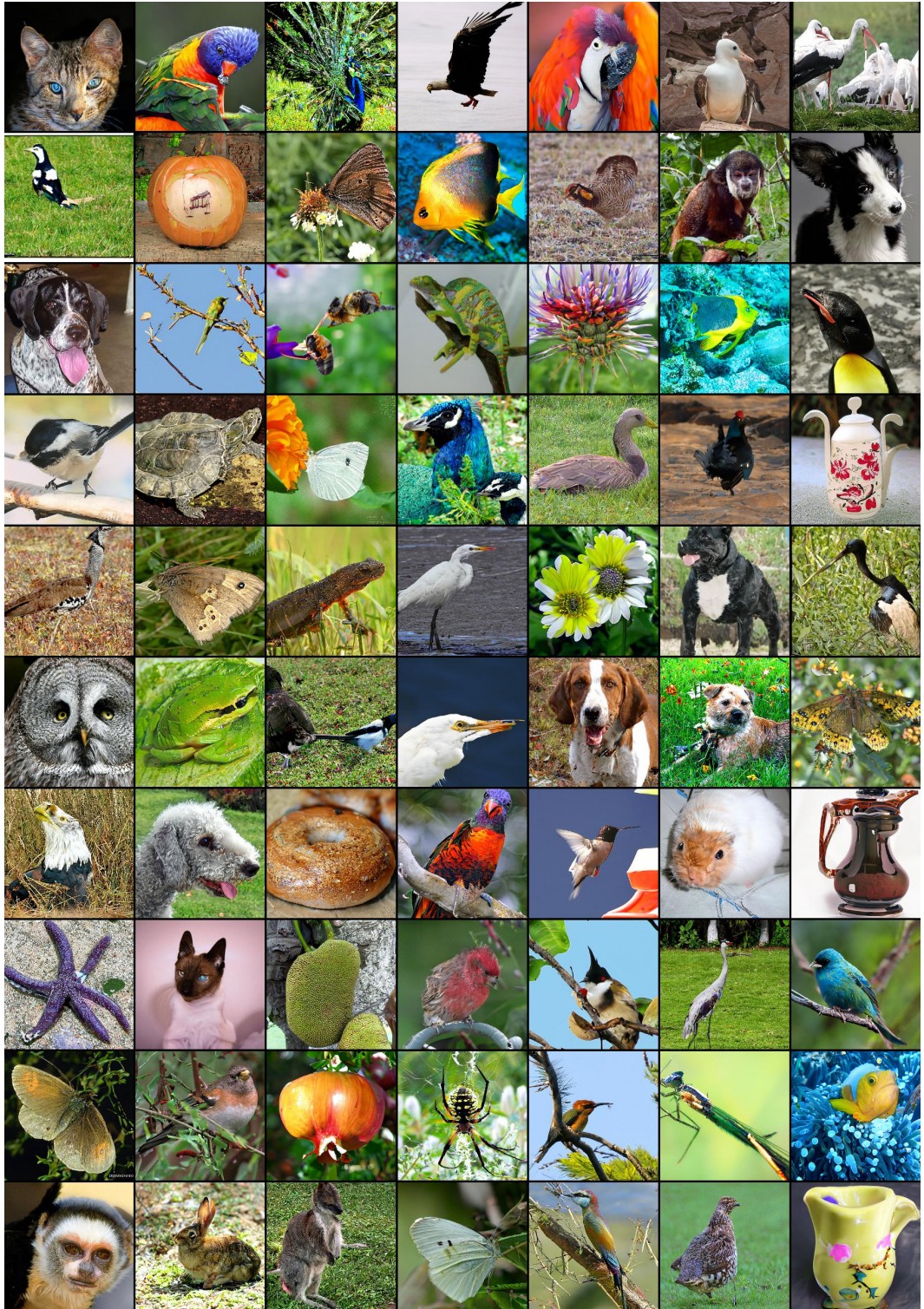

Figure 28: Random samples were generated by SciRE-V1-3 with 18 NFE and the uniform time steps, using the pre-trained discrete-time DM (Dhariwal & Nichol, 2021) on Imagenet 256×256 (classifier scale: 1).

