# OpenReview forum: "SciRE-Solver: Accelerating  Diffusion Models Sampling by Score-integrand Solver with Recursive Difference"
_ICLR.cc/2024/Conference — Submitted to ICLR 2024_

### Official Review · Reviewer_zqdo · 2023-10-30

**Soundness:** 3 good
**Presentation:** 1 poor
**Contribution:** 2 fair
**Rating:** 5
**Confidence:** 4

**Summary:**

The work investigates accelerating diffusion model sampling. Compared with existing works, the authors propose a new gradient estimation method,  named recursive difference, and show improvement compared with existing works.

**Strengths:**

1. The experiments are very comprehensive for image experiments in terms of FID.
2. Based on experiments in the main paper and appendix, the proposed new method shows improvements.

**Weaknesses:**

1. After reading the main paper, I have difficulties in understanding why the proposed recursive difference works better. The authors claim " This method recursively extracts the hidden lower-order derivative information of the higher-order derivative terms in the Taylor
expansion of the score-integrand at the required point, as illustrated in Figure 2." The recursive difference trick is key contribution of the work, authors should consider rewriting the above high-density sentence into an easy-to-understand paragraph and highlight why it works, presenting more analysis.

2. Can the author show some comparison in terms of numerical accuracy (MSE against ground truth solution) besides FID? How fast of various methods converge to ground truth solution?

3. After reading the main paper and appendix, it is unclear to me why the chosen coefficient C6 is better than C5. Besides the empirical experiments, do authors have more principled math analysis for them?

4. Similar to Q2, can authors present evidence that the proposed method can better estimate score gradients?

5. A recent work investigates a similar problem and shares a similar algorithm. Can authors comment on the connection and difference[1]?

[1] Zhang et al. Improved order analysis and design of exponential integrator for diffusion models sampling

**Questions:**

See above

---

> ### Author Response · Authors · 2023-11-19
>
> Dear Reviewer zqdo,
>
> Thank you for the detailed review and valuable feedback. Your comments are helping improve our work, many thanks. Below, we address specific questions you are concerned about.
>
> ### W1: ... The recursive difference trick is key contribution of the work, authors should consider rewriting the above high-density sentence... and highlight why it works, presenting more analysis.
>
> Thanks for the detailed review and valuable suggestions. We highly value your suggestions and have rephrased the relevant portion in main paper to provide a detailed explanation and context. For example, we indicate that in algorithms based on the exponential integrator (such as DPM-Solver [1] and DEIS [2]), in order to achieve fast sampling, the original coefficient $\frac{e^h-h-1}{h^2}$ of the difference term is replaced by $\frac{e^h-1}{h}$. Although this substitution is based on the concept of equivalent infinitesimals, it simultaneously reveals that after the replacement of the original coefficients, the acceleration effect becomes more pronounced. Therefore, we speculate that utilizing the conventional FD method directly to evaluate the derivative of the score function may be a suboptimal choice. Then, based on Taylor expansion, we recursively applied finite differences to handle higher-order terms and derived a new equivalent infinitesimal. We refer to this new equivalent infinitesimal as the recursive difference technique. For specific details, please refer to pages 4-5 of the main paper and Appendix E.
>
> [1] Lu et al., Dpm-solver: A fast ode solver for diffusion probabilistic model sampling in around 10 steps, NeurIPS 2022.
>
> [2] Zhang et al., Fast sampling of diffusion models with exponential integrator, ICLR 2023.
>
> ### W2-W4 ...why the chosen coefficient C6 is better than C5... How fast of various methods converge to ground truth solution?  ... can authors present evidence that the proposed method can better estimate score gradients?
>
> Thanks for the thoughtful questions. In fact, these issues may be equivalent. In fact, the main characteristic of the RD method is that this novel estimation incorporates low-order derivative information hidden in the higher-order derivative terms of the Taylor expansion. Refer to the demonstration in Eq. $(3.5)$ and the summary in Figure 2 for details. Compared to FD method, the RD method incorporates additional information $\frac{1-\phi_1(m)}{\phi_1(m)}\frac{\Gamma (\tau_{t_{i-1}}) - \Gamma (\tau_{t_i})}{h_{t_i}}$ from other higher-order derivative terms. Naturally, we have reason to believe that this additional term may counterbalance to a certain level with these higher-order terms from the Taylor expansion. This explanation may address your concerns. The details are included in the newly added Appendix E, especially in E2.
>
> In E2, to address the concerns you raised, we have analyze RD method from three perspectives as comprehensively as possible.
>
> - Firstly, we indicate that a commonality among the coefficients of RD method and the coefficients used in the exponential integrator is the amplification of the original coefficients of FD.
>
> - Secondly, we compare the coefficients of EI and RD method from the perspective of analyzing these three different coefficient functions (FD, EI, RD).
>
> - Finally, we attempt to theoretically substantiate our viewpoint regarding the offsetting of higher-order terms. We demonstrate that under certain conditions, RD methods for derivative estimation can achieve better truncation errors. We feel that these conditions may be relaxed, but after several days of derivation, we have not yet found a suitable approach. Nevertheless, these conditions all affirm the observed phenomenon.
>
>
> ### W5: A recent work investigates a similar problem and shares a similar algorithm. Can authors comment on the connection and difference[3]?
>
> Thanks for the interesting question. In fact, we believe that the exponential integrator fundamentally utilizes an equivalent infinitesimal substitution, amplifying the ratio of difference terms (as indicated in the previous answer). Upon your prompt, we noticed that paper [3] is also under review at ICLR 2024. Wishing them good luck. Based on my reading and learning, it appears that paper [3] also involves the adjustment of coefficients for difference terms. The difference lies in the fact that in reference [3], a method similar to the Butcher tableau is employed to adjust the coefficients of the original EI within the context of EI.
>
> [3] Zhang et al., Improved order analysis and design of exponential integrator for diffusion models sampling.

---

> > ### Comment · Reviewer_zqdo · 2023-11-22
> >
> > Thanks for addressing my questions. I will maintain my rating.
> >
> >
> > Regarding "present evidence that the proposed method can better estimate score gradients",
> >
> > Can the author provide empirical results based on MSE between various solvers and ground truth solutions, which can be obtained with a large NFE solution?

---

> ### Author Response · Authors · 2023-11-23
>
> Thanks for your response.
>
> We have conducted a comparison of MSE on the bedroom dataset under the same basic settings.
> Specifically, we used the uniform-time trajectory with the termination time set at $1e-3$.
> Due to time constraints, we have included these results in the attachment. Please refer to the report file "report_MSE.txt".  We sincerely hope that our endeavors garner your support.
>
> Thanks again.

---

### Official Review · Reviewer_MTm2 · 2023-10-31

**Soundness:** 3 good
**Presentation:** 3 good
**Contribution:** 3 good
**Rating:** 6
**Confidence:** 4

**Summary:**

This work introduces a new method, the recursive difference method, to improve the speed of Diffusion models by efficiently calculating score function derivatives. Their SciRESolver technique significantly accelerates DM sampling, achieving state-of-the-art FID scores with fewer NFEs. The method demonstrates remarkable performance on tasks like text-to-image generation, requiring few NFEs for high-quality samples.

**Strengths:**

In diffusion models, the NFE required for sampling has always been the main computational overhead, and reducing NFEs is very important for efficiency.

**Weaknesses:**

The improvement is not consistent, which makes the interpretation a little challenging.

**Questions:**

Given the current literature, as the distillation of score-based models can reduce NFE significantly. How can your algorithm be combined with distillation?

How can you explain the behavior of SciRE-V1-2 and SciRE-V1-3 from a theoretical perspective?

---

> ### Author Response · Authors · 2023-11-19
>
> Dear Reviewer MTm2,
>
> Thank you for the detailed review and thoughtful feedback. Your comments are helping improve our work, many thanks. Below, we address specific questions you are concerned about.
>
> ### W: The improvement is not consistent, which makes the interpretation a little challenging.
>
> Thanks for the detailed review. We carefully examined our entire paper, and perhaps the inconsistency you mentioned is reflected in the performance of the SciRE-V1-3-agile on the pre-trained model of continuous-time CIFAR-10. After closely inspecting all the procedures on continuous-time CIFAR-10, we found a lack of setting for random seeds in the codebase we downloaded. Upon fixing the random seed, we observed that the experimental results no longer exhibit fluctuations like the blue dashed line in sub-figure b of Figure 7.
>
> ### Q: Given the current literature, as the distillation of score-based models can reduce NFE significantly. How can your algorithm be combined with distillation? How can you explain the behavior of SciRE-V1-2 and SciRE-V1-3 from a theoretical perspective?
>
> Thanks for the thoughtful question. The SciRE-Solver sets up numerical algorithms from the perspective of numerical solutions to differential equations, eliminating the need for any additional training and optimization and offering strong flexibility.
>
> From my current perspective, while the distillation method of score-based can significantly reduce the NFE, the quality of the generated samples only achieves "a comparable level", falling short of the optimal performance (FID) of its base model (undistilled).
>
> We believe that training-free algorithms, can not only accelerate but also enhance the generative capabilities of models. SciRE-Solver has validated this perspective. Regarding how to further reduce NFE, research will become very challenging, but SciRE-Solver at least gives us a glimpse of such hope.
>
> The response regarding how to integrate with distillation and the theoretical analysis of SciRE-V1 is as follows:
>
> - The combination of an ODE Solver and distillation is exemplified in the Consistency Distillation algorithm for consistency models (see [1], [2]). As for how to integrate them more effectively, I don't have a definitive answer.
>
> - In Appendix F, we provide the convergence (convergence order) proofs for SciRE-V1-2 and SciRE-V1-3. Additionally, in Appendix E, we further analyze the recursive difference method.
>
>
> [1] Song et al., Consistency Models, ICML 2023.
>
> [2] Song et al., Improved Techniques for Training Consistency Models.

---

### Official Review · Reviewer_eocv · 2023-11-01

**Soundness:** 3 good
**Presentation:** 3 good
**Contribution:** 3 good
**Rating:** 6
**Confidence:** 4

**Summary:**

This paper proposes a new training-free sampling algorithm for diffusion models, called SciRE-solver. SciRE-solver is based on Taylor expansion and uses the proposed Recursive Difference (RD) method to estimate the derivative of the score model. The authors conduct extensive experiments on various datasets such as CIFAR10, CelebA, ImageNet, LSUN, showing that the proposed SciRE-solver consistently outperforms existing numerical samplers.

**Strengths:**

1. The paper is well-written with sufficient details and comprehensive ablation studies.
2. The paper clearly explains its relationship to and differences from the related work to be well-placed in the literature.
3. SOTA performance compared to existing numerical sampling algorithms on various datasets and diffusion models.

**Weaknesses:**

1. The paragraph above Figure 2 needs revision for clarity. The score approximation error is inevitable. How can the proposed sampling method mitigate this issue? The third sentence is also vague without explaining what the "additional variables" means. It is not clear how these considerations lead to the hypothesis either.
2. While the authors demonstrate generated samples of pre-trained DMs on high-resolution datasets such as ImageNet, LSUN-bedroom, and stable diffusion model, there is lack of quantitative results on these datasets except for ImageNet128. Can you also add quantitative results on ImageNet256 and LSUN-bedroom?
3. In Table 1, SciRE-solver uses pre-trained model from EDM. But the results of the original EDM sampler is missing from the comparison.
4. Minor: in the abstract, "Experiments demonstrate also that demonstrate that" -> "Experiments also demonstrate that".

**Questions:**

1. While SciRE-solver outperforms its counterpart DPM-solver in the experimental results, can you elaborate more on why it is better than DPM-solver numerically? Does SciRE-solver provide more accurate higher-order derivative estimation than DPM-solver theoretically?

---

> ### Author Response · Authors · 2023-11-19
>
> Dear Reviewer eocv,
>
> Thank you for the detailed review and thoughtful feedback. Your comments are helping improve our work, many thanks. Below, we address specific questions you are concerned about.
>
> ### W1: The paragraph above Figure 2 needs revision for clarity. The score approximation error is inevitable ...
>
> Thanks for the valuable feedback. Taking into full consideration this feedback, as well as the inevitability of approximation errors, we have revised the expression of this paragraph from the perspective of the characteristics of existing literature. Specifically, in order to accelerate sampling, existing literature has replaced the coefficients of finite difference terms with the concept of equivalent infinitesimals during algorithm design. We believe that such replacement fundamentally alters the way derivatives are estimated, and it also potentially indicates that using finite difference to estimate derivatives may not lead to further acceleration effects. This analysis supports our hypothesis.
>
> ### W2: ...demonstrate generated samples of pre-trained DMs ... such as ImageNet, LSUN-bedroom, and stable diffusion model, there is lack of quantitative results on these datasets except for ImageNet128. Can you also add quantitative results on ImageNet256 and LSUN-bedroom?
>
> Thank you for your suggestions. Due to time and server constraints, computing FID on large datasets has become challenging for us. Nonetheless, we have provided the main file of our algorithm, "scire_solver.py", in the supplementary materials. We welcome any explorers to try our algorithm on these large datasets. Subsequently, we will also open source all of our code.
>
> ### W3: In Table 1, SciRE-Solver uses pre-trained model from EDM. But the results of the original EDM sampler is missing from the comparison.
>
> Thanks for the detailed review. We have included the experiments for EDM in Table 1.  When invoking the Heun iteration algorithm for EDM, we utilized the trajectory with the EDM type.
>
> ### W3: Minor: in the abstract, "Experiments demonstrate also that demonstrate that" -> "Experiments also demonstrate that".
>
> Thanks for the detailed review. We have thoroughly examined our paper and made the necessary revisions. Many Thanks.
>
> ### Q: While SciRE-Solver outperforms its counterpart DPM-Solver in the experimental results, can you elaborate more on why it is better than DPM-solver numerically? Does SciRE-Solver provide more accurate higher-order derivative estimation than DPM-solver theoretically?
>
> Thanks for the thoughtful review. This question has prompted deep contemplation for me. I have attempted to theoretically derive some insights, but it has proven exceptionally challenging (so far). Due to time constraints, we discuss and analyze the commonalities and differences among the difference coefficients used in various estimation methods in the context of the exponential integrator. The details refer to Appendix E, esp. E2.

---

### Official Review · Reviewer_gjHa · 2023-11-04

**Soundness:** 2 fair
**Presentation:** 2 fair
**Contribution:** 3 good
**Rating:** 6
**Confidence:** 2

**Summary:**

This paper introduces the Recursive Difference (RD) method to calculate the derivative of the score function network. Based on the RD method and the truncated Taylor expansion of score-integrand, the authors propose SciRE-Solver to accelerate diffusion model sampling.
The core of their algorithm relies on evaluating higher-order derivatives of the score functions, which cannot be done by conventional finite difference methods, as errors can propagate easily. The RD method is proposed to tackle this problem. They provide extensive experiments on variant benchmark datasets to demonstrate the effectiveness of their approach.

**Strengths:**

The authors propose a fast sampling algorithm for diffusion models based on using an RD approach to evaluate higher-order derivatives of the score function. They clearly introduce the intuiotion and the background story. Extensive experiments on various datasets are conducted to support the use of their proposed algorithm. Compared to existing algorithms, the proposed SciRE-based algorithm in many cases achieve lower FID score with a fewer number of NFEs.

**Weaknesses:**

I have two major concerns:

1. Their main result, the RD procedure is not presented clearly enough. This algorithm is only described in words, and Figure 2 is hard to parse. From Equation (3.7) I see that to evaluate first order derivative at $s$, we need both the first and the second order derivatives at $t$. Then why the authors say in the caption of Figure 2 that we can evaluate the first order derivative at $s$ with only zero order derivative at $t$? I would suggest present the most general form algorithm in a pseudo-code format like Algorithm 1 and 2.

2. This paper might contain some critical typos that affect the entire proposal (see my second question).

I would love to increase my score if these issues are well-addressed.

**Questions:**

1. Is there any acceleration algorithm for diffusion SDE as well? If yes, I would love to see the authors providing a discussion. If no, could the authors elaborate a bit on why training-free acceleration is mostly for diffusion ODE?
2. I thought $\alpha_t = \prod_{i = 1}^t \beta_i$ is piecewise constant?  Then how do you define $f(t)$ as the derivative of $\log \alpha_t$ with respect to $t$? It is unclear whether the authors use $t$ as a index for discrete time step or continuous time.
3. In Eq (3.1), $h(r)$ should be $h_1(r)$.
4. I understand that $NSR$ is monotone, but why is it strictly monotone? Namely, how to guaratee the existence of its inverse function.
5. Why the authors say in Figure 1 that the proposed algorithm outperforms DDIM and DPM solver?
6. Where is $t_i$ defined?
7. Maybe this is a dumb question. Why can we assume the neural network is differentiable? I would imagine this is not the case when the activation function is ReLU.
8. This sentence is hard to parse. "This method recursively extracts the hidden lower-order derivative information of the higher-order derivative terms in the Taylor expansion of the score-integrand at the required point, as illustrated in Figure 2". I would suggest the authors present their most general form algorithm in the format of Algorithm 1 and 2.
9. Could the authors elaborate on what Figure 2 is trying to illustrate? In particular, why some blocks are colored in red and the others in blue? Why do you call the first row Taylor series and the second row Weighted sum?
10. In Theorem 3.2, $m$ has to be larger than 2 or 3?
11. The legend in Figure 3 is a bit misleading. I assume the first four ones are for dashed lines, but it is not apparent at first glance.

---

> ### Author Response · Authors · 2023-11-19
>
> Dear Reviewer gjHa,
>
> Thank you for the detailed review and valuable feedback. Your comments are helping improve our work, many thanks. Below, we address specific questions you are concerned about.
>
> ### Q1: Is there any acceleration algorithm for diffusion SDE as well? ... to see the authors providing a discussion.
>
> Thanks for providing such interesting topic. We observed that 'dpm-solver++' has introduced a dedicated accelerated iterative scheme specifically designed for diffusion SDE.
>
> The most significant difference between diffusion SDE and diffusion ODE lies in the presence of stochastic term. In diffusion SDE, once the coefficient function of the stochastic term is determined, the remaining integral term is equivalent to solving an ODE, because the integration of stochastic term can be estimated using the mean formula and variance formula of Itô integration. Therefore, there are two key aspects for diffusion SDEs.
> - The first involves finding a balance between the coefficient function of the stochastic term and the coefficient function of the Score function term.
> - The second aspect pertains to devising an algorithm capable of accommodating diffusion ODEs with deterministic variance perturbations.
>
> [1] Lu et al., DPM-Solver++: Fast Solver for Guided Sampling of Diffusion Probabilistic Models.
>
> ### Q2: how do you define $f(t)$ ..., It is unclear...use $t$ as a index for discrete time or continuous time.
>
> Thanks for the detailed review. We have corrected the expression in the Background and provided detailed explanations in Appendix A.1. We clarify that $\alpha_t=\prod_{i=1}^t \beta_i$ serves as the schedule for DDPM. From a discrete perspective, your thought is correct. For $f(t)$, we follow the generalized schedule defined by Kingma et al.; the details refer to Appendix A.1. Our theoretical framework is established in continuous time. Nevertheless, our algorithms can also be applied to discrete-time diffusion models, thanks to the groundwork laid by DPM-Solver. Details of the conversion from continuous time to discrete time are provided in Appendix G.
>
> [2] Kingma et al., Variational Diffusion Models, NeurIPS 2021.
>
> [3] Lu et al., Dpm-solver: A fast ode solver for diffusion probabilistic model sampling in around 10 steps, NeurIPS 2022.
>
> ### Q3: In Eq (3.1), $h(r)$ should be $h_1(r)$.
>
> Thanks for the very detailed review. We carefully examined the entire text and corrected it.
>
> ### Q4: ...why $NSR$ is it strictly monotone?
>
> Under the variance-preserving setting, $\sigma_t^2 +\alpha_t^2 = 1$. Since $\sigma_t$ is strictly monotonic, $NSR(t)=\frac{\sigma_t}{\alpha_t} $ is also strictly monotonic. The inverse of $NSR(t)=\frac{\sigma_t}{\alpha_t}$ is provided in Appendix G3.
>
> ### Q5: Why ... in Figure 1 that the proposed algorithm outperforms DDIM and DPM solver?
>
> Thanks for the detailed question. For a single image, we often lack a reasonable method to compare its generation quality, as each algorithm's generated image may not necessarily be the optimal image under that specific noise. In such cases, we can only evaluate based on the details and clarity of the generated images. Therefore, in Figure 1, we can assess based on the details and clarity of the generated samples. For instance, the samples produced by our algorithm at 15 steps are clearer than those generated by DDIM at 15 and 20 steps, and are even comparable to those produced by DDIM at 50 steps.
>
> ### Q6: Where is $t_i$ defined?
>
> $t_i \in[0,1]$ represents the time at the $i$-th moment.
>
> ### Q7: Why can we assume the neural network is differentiable? ... is not the case when the activation function is ReLU.
>
> Thanks for the thoughtful question. The differentiability allows us to compute the gradient of the loss function with respect to the network parameters, enabling parameter updates. ReLU is indeed non-differentiable. We have provided a review study that we hope will be helpful to you for more details.
>
> [4] Dubey et al., Activation functions in deep learning: A comprehensive survey and benchmark, Neurocomputing (2022).
>
> ### Q8-Q9: This sentence is hard to parse. "...".  I would suggest ...
>
> Thanks for the valuable suggestion. In order to facilitate reader comprehension, we have rephrased the sentence into a paragraph and provided a detailed explanation. In the revised version, we illustrate this with an example using Eq. (3.7), where the colors annotated in the equation correspond to the colors in Figure 2. For the general form of the algorithm, we showcase it in the attached scire-solver.py.
>
> ### Q10: In Theorem 3.2, has to be larger than 2 or 3?
>
> In fact, $m-1$ represents the number of recursive operations performed on how many higher-order derivatives. We recommend performing at least two recursive operations, so m should be greater than 2.
>
> ### Q11: The legend in Figure 3 is a bit misleading. ... for dashed lines, but it is not apparent...
>
> Thanks for the detailed review. We have made appropriate adjustments to make it look more apparent.

---

> > ### Author Response · Authors · 2023-11-22
> >
> > Thank you again for your time and effort in reviewing our work.  We hope that we were able to address your concerns in our response.  If possible, please let us know if you have any further questions before the reviewer-author discussion period ends.  We would be delighted to address your further concerns.
> >
> > Thanks again.

---

> > ### Comment · Reviewer_gjHa · 2023-11-22
> >
> > I want to thank the authors for the detailed response. A good proportion of my concerns are addressed, hence I increase my rating from 5 to 6.
> >
> > However, I still feel hiding the general form of the algorithm in Python script causes a lot of confusion, maybe the authors should consider improving their manuscript by presenting a the algorithm instead in the main paper.

---

> > > ### Author Response · Authors · 2023-11-23
> > >
> > > Thanks again.
> > >
> > > Due to time constraints, if possible, we will do our best to consider this issue.

---

### Public Comment · ~Shigui_Li2 · 2025-07-03

**SciRE-Solver** is a training-free acceleration framework for diffusion model sampling, built on a novel numerical approach we call *recursive difference refinement*.

## 🔍 The Problem

Existing fast samplers like DPM-Solver rely on Signal-to-Noise Ratio (SNR = $\frac{\alpha_t^2}{\sigma_t^2}$) transformations during inference stage, but suffer from numerical instability in final denoising steps due to SNR divergence as $\sigma_t \to 0$. Even log-SNR ($\lambda$) space, while moderating SNR values, still faces the same issue where $\log\text{-SNR} \to \infty$ as $\sigma_t \to 0$.

## 🔧 Our Solution

We reformulate the problem using **Noise-to-Signal Ratio** (NSR = $\frac{\sigma_t^2}{\alpha_t^2}$), which naturally avoids numerical instability as $\sigma_t \to 0$, providing superior numerical conditioning. This NSR-based approach, combined with our recursive difference refinement of score function integrals, delivers improved generation quality and computational efficiency.

Our ablation studies demonstrate that the NSR space not only addresses these numerical challenges but also enhances both generation quality and computational performance.

---

### Meta-Review · Area_Chair_wCwD · 2023-12-03

**Metareview:**

This paper focuses on the numerical simulation of the backward process of diffusion generative model, for accelerated generation speed. A Recursive Difference method is proposed to approximate the derivative of the score function network. I, like several reviewers, consider this to be an interesting idea. However, expert reviewers also raised concerns about evidence that the proposed method can better estimate score gradients. Besides soundness, there are also shared concerns about the presentation. Although after careful consideration based on the reviews and the discussion, I cannot recommend acceptance at this stage, I encourage the authors to investigate more and strengthen the evidence for a future acceptance. In particular, one possibility is to provide separate demonstrations, one on RD's numerical differentiation capability, perhaps with both numerical and analytical results, and another on how this enables improves the generation. For the latter, numerical results should suffice.

**Justification For Why Not Higher Score:**

I agree with the reviews.

**Justification For Why Not Lower Score:**

N/A

---

### Decision · Program_Chairs · 2024-01-16

Reject